Manuscript prepared for Hydrol. Earth Syst. Sci.
with version 2014/09/16 7.15 Copernicus papers of the LaTeX class copernicus.cls.
Date: 31 January 2020

# Reconstructing the 2015 Salgar Flash Flood Using Radar Retrievals and a Conceptual Modeling Framework in an Ungauged Basin

Nicolás Velásquez [1,2], Carlos D. Hoyos [1,2], Jaime I. Vélez [1], and Esneider Zapata [2]

[1]Universidad Nacional de Colombia, Sede Medellín, Facultad de Minas, Departamento de Geociencias y Medio Ambiente
[2]Sistema de Alerta Temprana de Medellín y el Valle de Aburrá (SIATA), Área Metropolitana del Valle de Aburrá (AMVA)

*Correspondence to:* Nicolás Velásquez (nvelasqg@unal.edu.co)

**Abstract.** On May 18, 2015, a severe rainfall event triggered a flash flood in the municipality of Salgar, located in the northwestern Colombian Andes. This work aims to reconstruct the main hydrological features of the flash flood to better understand the processes modulating the occurrence of the event. Radar quantitative precipitation estimates (QPEs), satellite information, and post-event field

visits are used to reconstruct the Salgar flash flood, in an ungauged basin, addressing the relationship among rainfall spatiotemporal structure, soil moisture, and runoff generation during successive rainfall events by using a conceptual modeling framework including landslide and hydraulic submodels. The hydrological model includes virtual tracers to explore the role of runoff and subsurface flow and the relative importance of convective and stratiform precipitation in flash flood generation.

Despite potential shortcomings due to the lack of data, the modeling results allow an assessment of the impact of the interactions between runoff, subsurface flow, and convective-stratiform rainfall on the short-term hydrological mechanisms leading to the flash flood event. The overall methodology reproduces the magnitude and timing of the La Liboriana flash flood peak discharge considerably well, as well as the areas of landslide occurrence and flood spots, with limitations due to the spatial

resolution of the available digital elevation model. Simulation results indicate that the flash flood and regional landslide features were strongly influenced by the antecedent rainfall, which was associated with a northeasterly stratiform event. The latter recharged the gravitational and capillary storages within the mode, moistening the entire basin before the occurrence of the flash flood event and impacting the subsurface-runoff partitioning during the flash flood event. Evidence suggests that the

spatial structure of the rainfall is at least as important as the geomorphological features of the basin in regulating the occurrence of flash flood events.

## 1 Introduction

Flash floods are regarded as one of the most destructive hydrological hazards, resulting in considerable loss of human life and high costs due to infrastructure damage (Roux et al., 2011; Gruntfest and Handmer, 2001). Among all different types of floods, Jonkman (2005) shows that flash floods result in the highest average mortality rate per event (3.62%), almost ten times larger than the mortality rate for river floods. Flash floods are usually described as rapidly rising water level events occurring in steep streams and rivers, associated with short-term, very intense convective precipitation systems or orographically forced rainfall events over highly saturated land surfaces and steep terrains (Šálek et al., 2006; Llasat et al., 2016; Douinot et al., 2016). Convective precipitation episodes often feature high intensity, short duration, and relatively reduced spatial coverage (Houze, 2004).

Several authors have assessed the role of the geological and geomorphological features of the catchment, soil type, soil moisture conditions, and the spatiotemporal structure of rainfall on flash flood occurrence, identifying the leading causative mechanisms of this hazard (Merz and Blöschl, 2003). Adamovic et al. (2016) and Vannier et al. (2016) related the flash floods governing processes to the geological properties of the basins with mixed results. Wu and Sidle (1995) emphasized the role of the topography, ground cover, and groundwater in the occurrence of shallow landslides and associated debris flows. Many authors have assessed the influence of hills and stream slopes, suggesting the slopes of the hills are significantly more important for flash flood occurrence and magnitude than the slope of the stream (Šálek et al., 2006; Roux et al., 2011; Yatheendradas et al., 2008; Younis et al., 2008). Rodriguez-Blanco et al. (2012) analyzed flash flood episodes in Spain and determined that antecedent soil moisture conditions play a significant role in runoff production. Castillo et al. (2003), also suggested an significant correlation between flash flood magnitude and the antecedent moisture conditions. Aronica et al. (2012) used spatial and statistical analysis to reconstruct landslides and deposits, finding a connection between flash flood occurrence and soil moisture antecedent conditions.

The fact that small basins are more prone to flash floods (Wagener et al., 2007), makes difficult their measurement and, consequently, their understanding and their prediction (Hardy et al., 2016; Ruiz-Villanueva et al., 2013; Yamanaka and Ma, 2017; Borga et al., 2011; Marra et al., 2017). The local rainfall storm events related to flash floods require that high spatiotemporal resolution to be characterized (Norbiato et al., 2008). Some authors follow a climatological approximation to assess the recurrence of flash floods in particular regions, focusing on the atmospheric causative mechanisms. For example, Kahana et al. (2002) examined the extent to which floods in the Negev Desert are the outcome of climatological synoptic-scale features, finding that about 80% of the events can be linked to distinct synoptic conditions occurring days prior to the flood events. Schumacher and Johnson (2005) studied extreme rain events associated with flash flooding in the United States over a 3-yr period, using the national radar reflectivity composite data. They found that 65% of the total number of flah floods are associated with mesoscale convective systems (MCSs), with two recurrent

patterns of organization: the existence of training convective elements and the generation of quasi-stationary areas of convection with stratiform rainfall downstream. Fragoso et al. (2012) analyzed storm characteristics and rainfall conditions for flash flood occurrence at Madeira (Portugal), and their results suggest an essential role of global climate patterns (North Atlantic Oscillation -NAO-forcing) and local forcing (orographic features) in the triggering of such events. Implicitly, these studies and all the others available in the peer-reviewed literature point to the need for local and regional high-quality spatiotemporal rainfall data. Berne and Krajewski (2013) highlighted the need to incorporate high-resolution weather radar information, even with some limitations, in flash flood hydrology.

The topography of Colombia is characterized by three branches of the Andes crossing the country south-to-north, generating a mixture of landscapes from high snow-capped mountains, vast highland plateaus, deep canyons to wide valleys, making some regions highly prone to flash flood occurrence. The likelihood of flash flood occurrence in Colombia is also high due to the spatiotemporal behavior of the Intertropical Convergence Zone, and the direction of the near-surface moist air flow leading to orographic enhancement of convective cores (Poveda et al., 2007). In the last decade, there have been several widespread and localized flash flood events in Colombia associated with climatological features and the local intensification of rainfall events. According to estimates by the "Comisión Económica para América Latina y el Caribe", the 2010-2011 La Niña event alone triggered 1233 flooding events and 778 mass removal processes in Colombia, with more than 3 million people affected and damages estimated at more than 6.5 billion US dollars.

After the 2010-2011 widespread disaster, several isolated events have occurred in the country with devastating consequences. The present paper focuses on studying the processes triggering a flash flood in La Liboriana basin, a 56 km$^2$ basin located in the western range of the Colombian Andes, as a result of consecutive rainfall storms that took place between May 15 and May 18, 2015. The resulting flash flood dramatically affected the region, causing more than 100 casualties, affecting several buildings and critical infrastructure, and resulting in a total reconstruction cost estimated at 36,000 million Colombian pesos (about 12.5 million dollars considering the 2018 exchange rate), which corresponds to three times the annual income of the municipality. Figure 1 shows an example of infrastructure damage and changes in the basin's main channel as a result of the flash flood event, showing considerable river margin and bed erosion. Despite the data scarcity, including of discharge measurements, the analysis of the successive rainfall events triggering the Salgar flash flood provides an interesting case of study for assessing the mechanisms that depend on the soil moisture conditions and rainfall distribution.

La Liboriana is a typical case of an ungauged basin (Sivapalan et al., 2003; Seibert and Beven, 2009; Beven, 2007; Bonell et al., 2006; Yamanaka and Ma, 2017), without any detailed records of soils or land use, topographic maps or high-resolution digital elevation models (DEMs), and scarce hydro-meteorological data. According to Blöschl et al. (2012), there are three general strategies for

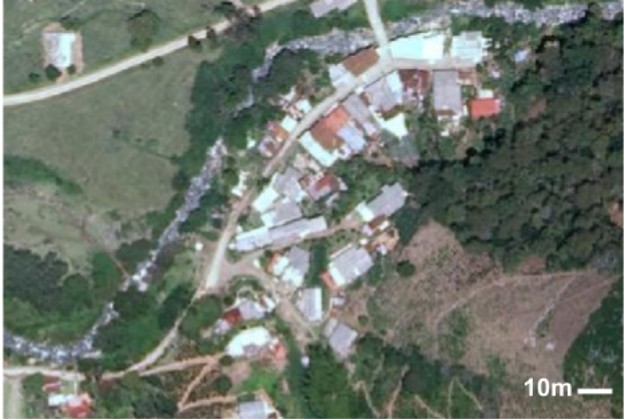

a) Aerial photograph before the event (2012).

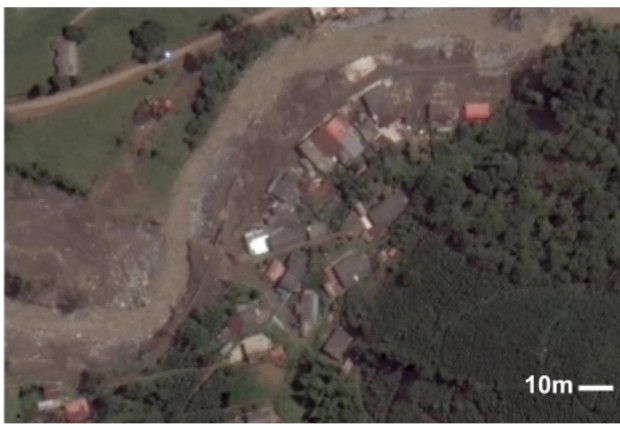

b) Aerial photograph taken after the event (2015-05).

**Figure 1.** Example of infrastructure damage as a result of La Liboriana flash flood event on May 18, 2015. a) Aerial photograph taken before the event (2012), during a mission of the Department of Antioquia's government, and b) satellite image after the event (courtesy of CNES/Airbus via Google Earth). The images show the destruction of most houses in that particular community, a bridge over La Liboriana, and the main road. All of the houses shown in the 2015 image had to be either demolished or structurally repaired. The images also show changes in the delineation of the main channel as well as considerable erosion in the river margins.

using models under these conditions. The first strategy is to obtain the required model parameters from the historical basin behavior and the morphological characteristics of the basin. This strategy often leads to low model performance (Duan et al., 2006). The second approach is to inherit the hydrological model calibration from a neighboring gauged watershed, which in this case does not exist. The third method is to parameterize the model based on proxy variables, such as hydraulic information obtained during field visits. In the case of the 2015 La Liboriana basin flash flood, there are no previous historical streamflow records, nor records from a neighboring watershed; thus, we followed the third approach. We use precipitation information derived from radar, satellite and aerial images, in addition to post-event field visits, to reconstruct the Salgar flash flood event. This

study addresses two broad hydrological issues. The first issue consists in exploring the relationship between rainfall spatiotemporal structure (Llasat et al., 2016; Fragoso et al., 2012), soil moisture and runoff generation (Penna et al., 2011; Tramblay et al., 2012; Garambois et al., 2013) during the successive rainfall events, and the second one in proposing a simplified hydrological modeling scheme, including landslide and hydraulic submodels, to assess the potential occurrence of flash flood events.

We use the WMF (Watershed Modeling Framework) which includes a variation of the TETIS hydrological model (Vélez, 2001; Francés et al., 2007), modified to include a shallow landslide submodel, and a floodplain submodel called *HydroFlash*. The TETIS model is a cell-distributed conceptual hydrological model that uses storage tanks and the kinematic wave approximation to simulate the most relevant processes in the basin. The landslides submodel is a stability model that classifies cells into unconditionally-stable, unconditionally-unstable, and conditionally stable depending on geomorphology; conditionally stable cells are further classified as stable or unstable based in their variable water content (Aristizábal et al., 2016). HydroFlash is a low-cost 1D model that estimates the cross-sectional filled area at all time steps on the basis of the liquid discharge and the sediment transport. In addition, the TETIS model was modified to include four virtual tracers to separately explore the role of runoff and subsurface flow, as well as the relative importance of convective and stratiform precipitation in flash flood generation. The assessment of the interactions between runoff, subsurface flow, and convective-stratiform rainfall allows a better understanding of the short-term hydrological mechanisms leading to the flash flood event.

The document is structured as follows. Section 2 describes in more detail the region of study, La Liboriana basin, including geomorphological and climatological characteristics of the basin, and the information sources used in this assessment. Section 3 presents a description of the overall methodology and the TETIS model, including flow separation, and the shallow landslide and HydroFlash submodels. Section 4 describes the main results of the study, including model validation and sensitivity analysis, and presents results from the landslide and HydroFlash submodels. Section 5 includes a discussion on the role of the rainfall structure in the flash flood reconstruction. Finally, the conclusions are presented in section 6.

## 2   Study site and data

### 2.1   Catchment description

The urban area of the municipality of Salgar is located near the outlet of La Liboriana basin, a small (56 km$^2$) tropical watershed located in the westernmost range of Colombia's Andes (Figure 2). By 2015, Salgar counted 17 400 inhabitants, including 8 800 residing in the urban area. La Liboriana basin joins the El Barroso river basin, and both drain to the Cauca River.

The availability of the ALOS-PALSAR DEM (ASF, 2011), with a resolution of 12.7 m, allow
to estimate the main geomorphological features of the basin. The average slope of La Liboriana is
57.6%, and the basin longitude and perimeter are 13.5 km and 57.8 km, respectively. The Strahler-
Horton order of the main stream is 5, and its longitude and slope are 18.1 km and 8.1%, respec-
tively. The highest elevation of the watershed (Cerro Plateado) reaches 3 609 meters above sea level
(m.a.s.l), while the outlet of the basin is at 1 316 m.a.s.l. The 99th slope percentile of order 1 streams
is 78%. For streams of order 2, 3, 4, and 5, the 99th slope percentiles are 61, 27, 18 and 11%,
respectively. Figure 2 shows the spatial distribution of the slopes in the watershed. These features
are typical of Andean mountainous basins. Geomorphologically, this kind of watershed tends to be
prone to the occurrence of flash floods (Lehmann and Or, 2012; Penna et al., 2011; Martín-Vide and
Llasat, 2018; Longoni et al., 2016; Ozturk et al., 2018; Khosravi et al., 2018; Marchi et al., 2016;
Bisht et al., 2018).

At the subbasin scale, La Liboriana exhibits a vast range of slopes and altitude differences. Figure
2 shows the height above the nearest drainage (HAND) model (Rennó et al., 2008) for La Liboriana.
The HAND calculates the relative height difference between cell $i$ and its nearest streamflow cell $j$.
La Liboriana HAND exhibits values between 500 and 800 m. Near the outlet of the basin, over the
banks, there are values close to 0 m. High HAND values at the upper region of the watershed often
denote areas of high potential energy, with increased sediment production and frequent shallow
landslide occurrence. Banks with low HAND values are more susceptible to flooding and tend to
correspond to areas prone to extensive damages caused by extreme events. The social challenges lie
in the high vulnerability of Salgar, given the location of the main urban settlement.

Vegetation and land use vary considerably within the basin. Figure 3 shows land use in different
regions of the watershed from a 2012 aerial image. In the upper La Liboriana basin, there is dense
vegetation (see Zoom 1 in Figure 3), with a high percentage of the area covered by tropical forests
and presence of grass and few crop fields. A portion of the upper watershed is considered a national
park. Hillslopes near the divide do not evidence significant anthropic intervention most likely due to
the steepness of this region. Down the hills and at the bottom of the valley, there are coffee plantations
(the primary economic activity of the region) and pastures. Downstream (Figure 3, Zoom 2), the
presence of crops is evident among forest and grass areas. Near the middle of the basin (Figure 3,
Zoom 3), the presence of crops is more obvious, and human settlements and roads start to appear.
The watershed exhibits grazing areas and urban development near the river banks. In Figure 3, the
Zoom 4 corresponds to the first affected urban area from upstream to downstream during the flash
flood. It is also possible to see a marked presence of crops and some patches of forest. Finally, Zoom
5 shows the main urban area of Salgar surrounded by crops, grass and an important loss of forest
coverage.

One of the challenges for hydrological modeling and risk management in the country is that soils
are not well mapped; the national soil cartography is usually available in a 1:400,000 scale. At this

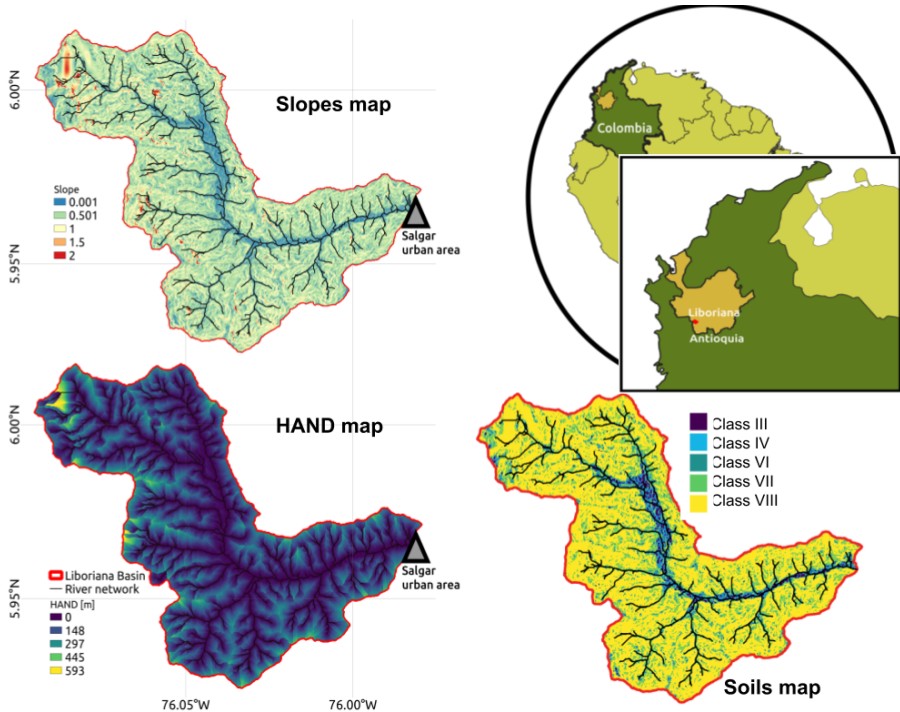

**Figure 2.** Geographical context of Liboriana basin, located in Colombia, in the Department of Antioquia. The panels include the map of slopes, the height above the nearest drainage (HAND), and the soil type map. The HAND values where estimated using a 12.7 m-resolution DEM. Low HAND values correspond to areas prone to flooding. Note that the soil type map is an extrapolation of the soil properties as a function of slope.

**Table 1.** Description of the soils in the region (Osorio, 2008).

| Type | Slope | Depth [m] | Retention | Permeability | Percentage |
|------|-------|-----------|-----------|--------------|------------|
| **Class III** | <12 | 0.6 | Low | High | 3.2 |
| **Class IV** | 12-25 | 0.6 | Mean | Mean | 8.3 |
| **Class VI** | 25-30 | 1.0 | Mean | Mean | 2.1 |
| **Class VII** | 30-50 | 0.3 | Too Low | Low | 25.5 |
| **Class VIII** | >50 | 0.2 | Too Low | Low | 60.0 |

scale, the municipality of Salgar, including La Liboriana basin, corresponds to only one category of soil texture. Osorio (2008), based on field campaign observations and laboratory tests, described La Liboriana soils as well-drained with poor retention capacity. Organic material is predominant in the first layer, and clay loam soil predominates within the second layer. The depth of the soil is hillslope dependent, varying from 20 cm to 1 m (Osorio, 2008). Table 1 provides a summary of soil characteristics for five different categories, all as a function of slope. Each soil category has a corresponding depth and a qualitative description of permeability and retention.

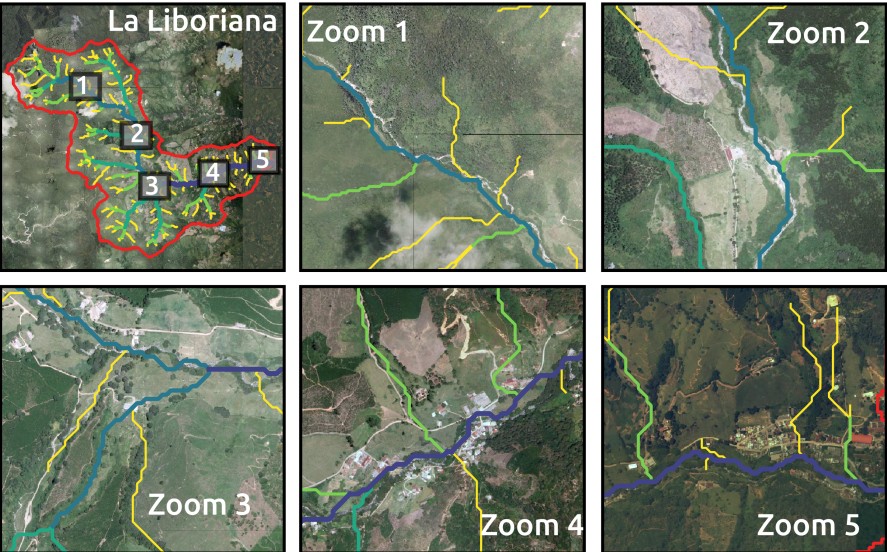

**Figure 3.** Aerial overview of La Liboriana basin (source: Department of Antioquia). The top-right panel presents the entire basin, showing the location of key regions detailed in the following panels, in zooms 1 to 5. The stream network is also presented, colored by order, from yellow to deep blue corresponding to orders 1 to 5.

## 2.2 Flash flood post-event observations

We conducted a field campaign a few days after the May 18th flash flood to assess the cross-section
geometry along the main channel in different sites, including at the outlet of the basin. During the
campaign, we measured sectional distances and the surface water speed, at different points of the
streamflow. The surface water speed was measured using a hand-held Stalker Pro II velocity radar.
We also identified traditional post-event terrain, land cover, vegetation and infrastructure markers to
record the approximate level associated with the peak flow during the flash flood. Figure 4 presents
the selected cross-section used for the estimation of the maximum discharge during the flash flood
given its geometrical and hydraulic regularity. The section has a rectangular shape, 4.6 m wide and a
height of 5 m for a total area of 23 $m^2$. A visual inspection of the flooded house around the section,
located 4-5 m away from the channel, reveals the presence of mud marks on the walls with heights
varying between 0.5 and 1.2 m (see Figure 4). The area of the section plus the flooded area during
the event was estimated to be 37 $m^2$. During the campaign, the surface speeds in the channel varied
between 2 and 3 ms$^{-1}$, for a 3 $m^3$ s$^{-1}$ discharge. Instrumented basins in the region, with similar
characteristics in terms of area and slopes, show peak flow surface water speeds ranging between 5
and 7 ms$^{-1}$ (See Figure A.1). By assuming an area of 37 $m^2$ and velocities between 5 and 6, we
estimate that the flash flood peak flow was between 185 and 222 $m^3s^{-1}$. Local authorities reported
that the peak streamflow reached the urban perimeter after 2:10 a.m. on May 18th (personal com-

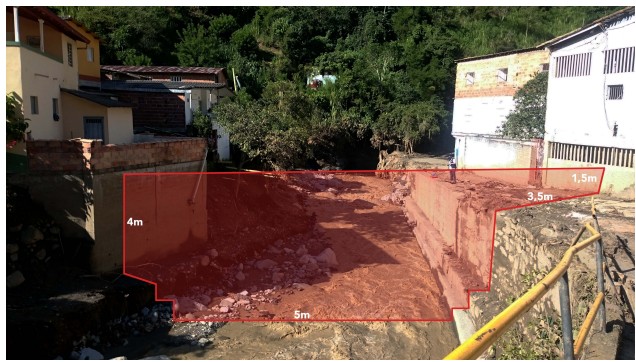

**Figure 4.** Channel cross-section showing an example of flooded infrastructure during the flash flood event. The section shows mud marks on the walls of adjacent houses, with heights varying between 0.5 and 1.2 m. The houses in the picture are located 4-5 m away from the channel. The photograph also shows the width of the channel and the total estimated depth during the flash flood. The cross-section is downstream from the bridge shown in the picture.

munication during the field visit). Reports state that the peak flow in the most affected community occurred near 2:40 a.m [1].

Aerial information before and after the occurrence of the event is relevant to analyze the location of the landslides and flooded areas. During 2012, the Department of Antioquia conducted a detailed aerial survey of the municipality of Salgar, and a few days after the event, DigitalGlobe and CNES/Airbus made available highly detailed satellite images of the same region. We performed a detailed contrast between both products by using a geographic information system (QGIS), which provided us with information about flooded areas and landslide locations (see Figures 1 and 16). Field campaign peak flow estimates and aerial imagery are used to validate the results obtained with the TESTIS model.

### 2.3 Rainfall information

The assessment of the 2015 Salgar flash flood event following a hydrological modeling strategy uses a radar-based QPE technique described in Sepúlveda (2016) and Sepúlveda and Hoyos (2017), using radar reflectivity fields, rainfall gauges and disdrometers. The QPE technique uses retrievals from a C-band polarimetric Doppler weather radar operated by the Sistema de Alerta Temprana de Medellín y el Valle de Aburra (SIATA, a local early warning system from a neighboring region, www.siata.gov.co). The radar is 65 km away from the basin. It has an optimal range in a radius of 120 km for rainfall estimation and a maximum operational range of 240 km for weather detection.

---

[1]As reported by the media and the national government: http://www.elcolombiano.com/antioquia/tragedia-en-antioquia-salgar-un-ano-despues-XX4145514, https://caracol.com.co/emisora/2015/12/25/medellin/1451076926_792470.html, http://portal.gestiondelriesgo.gov.co/Paginas/Noticias/2015/Antecion-Emergencia-Salgar-Antioquia.aspx

The radar operating strategy allows obtaining precipitation information at 5 minutes time step, with a spatial resolution of about 128 m. Despite the distance between the radar and the basin, and the mountains between them, there are no blind spots for the radar. A comparison between the radar QPE estimates and records from two rain gauges installed three days after the flash flood event show a correlation for an hourly time scale of 0.65. A detailed description of the rainfall estimation, as well as the overall meteorological conditions that led to the La Liboriana extreme event, are described in a companion paper (Hoyos et al., 2019). Radar retrievals are also used to classify precipitation into convective and stratiform areas following a methodology proposed by Yuter and Houze (1997) and Steiner et al. (1995), based on the intensity and sharpness of the reflectivity peaks. The methodology has been widely used in tropical regions as reported in the review by Houze et al. (2015).

Between May 15 and May 18, 2015, several storms took place over La Liboriana basin. During the night of May 17, between 02:00 and 09:00 a.m. (local time), a precipitation event covered almost all of the basin (hereafter referred to as precipitation Event 1). Twenty hours later, between 23:00 p.m. on May 17 and 02:00 a.m. on May 18, two successive extreme convective systems occurred over the basin with the maximum intensity in the upper hills (precipitation Event 2). Event 1 corresponds mainly to a stratiform event with an average precipitation accumulation of 47 mm over the basin. Event 2 corresponds to a moderate average of 38 mm, however the accumulation exceeded 180 mm over the upper watershed. Hoyos et al. (2019) show that the individual events during May 2015 were not exceptional, the climatological precipitation anomalies were negative-to-normal, and the synoptic patterns associated with the extreme events were similar to the expected ones for the region. However, but the combination of high rainfall accumulation in a 96-hour period as a result of successive precipitation events over the basin, followed by a moderate extreme event during May 18, is unique in the available observational radar record, in particular for the upper part of the basin. Figure 5a presents the temporal evolution of the estimated convective-stratiform rainfall partitioning during both Events 1 and 2. The main difference between both events is the timing of the convective versus stratiform participation within each case. Event 1 started as a stratiform precipitation event moving northeastward, from the Department of Chocó to the Department of Antioquia across the westernmost Andes mountain range. After 3 hours of stratiform rainfall, training convective cores move over La Liboriana basin generating intense precipitation peaks in a 2.5 hour period. It is important to note that these cores did not strengthen within La Liboriana basin; these systems formed and intensified over the western hills of Farallones de Citará, draining to the Department of Chocó towards the Atrato river. This is not a minor fact because, as a result of the latter process, the maximum intensity cores did not fall over the steepest hills of La Liboriana basin but rather near the basin outlet where the slopes are considerably flatter. Figure 5b shows the spatial distribution of cumulative rainfall during Event 1, with the maximum precipitation located toward the bottom third of the basin. Event 2, on the other hand, started as a thunderstorm training event with two convective cores moving from the southeast, followed by the remaining stratiform precipitation. Even though the average

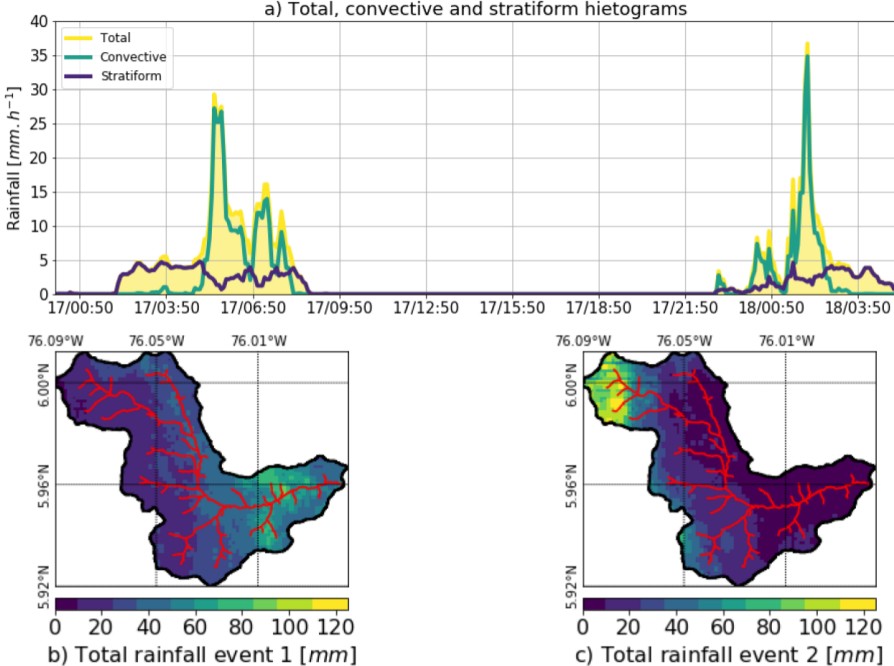

**Figure 5.** a) Temporal evolution of the convective-stratiform rainfall partitioning during both Events 1 and 2 (precipitation intensity in mm per hour, for 5-minute periods). The figure shows the total rainfall (yellow), the convective (blue) and stratiform (green) portions integrated over La Liboriana basin. b) and c) Spatial distribution of the cumulative rainfall during Events 1 and 2 over La Liboriana basin, respectively.

cumulative rainfall over the basin was 9 mm less than during Event 1, this event is characterized by orographic intensification within the basin, leading to a more heterogeneous spatial distribution with the highest cumulative precipitation in the steepest portion of the basin (see Figure 5b).

The data requirements and rainfall preprocessing needed for the overall methodology followed in the reconstruction of the 2015 Salgar flash flood, are summarized in Table 2 and are presented in a schematic diagram in Figure 6.

## 3 Methodology

### 3.1 TETIS hydrological model

We used a physically-based, distributed hydrological model developed and fully described in Vélez (2001) and Francés et al. (2007). The spatial distribution and the hydrological flow path schema is based on the 12.75 m-resolution DEM. In each cell, five tanks represent the hydrological processes including capillary (tank 1), gravitational (tank 2), runoff (tank 3), baseflow (tank 4) and channel storage tanks (tank 5). The state of each tank varies as a function of vertical and lateral flows as shown in Figure 6, where the storage is represented by $S_i$ [mm] and the vertical input to each tank

| Item | Description/Source | Period | Usage |
|------|--------------------|--------|-------|
| Radar data | QPE rainfall estimations | 2015-05-17 to 2015-05-18 | TETIS runs, rainfall characterization and event analysis. |
| Field campaign | Maximum streamflow estimation through visual inspection | 2015-05-20 | TETIS model comparison for indirect validation. |
| Satellite imagery | Visible channel compositions from the DigitalGlobe CNES imagery | 2015-05 (post-event) | Flash flood model validation, shallow landslides model validation, and comparison with pre-event conditions. |
| Aerial photos | Aerial photos taken by the government of Antioquia during 2012. | 2012 | Pre-event conditions comparison. |
| Soils description | Physical description of the soils of the region by Osorio (2008) | 2008 | SimulationS using TETIS (model setup). |

**Table 2.** Summary of the data used for the setup of TETIS.

by $D_i$ [mm], which in turns depends on the vertical flow through tanks $R_i$ [mm]. $E_i$ [mm] represents the downstream connection between cells, except for tank 1, where $E_1$ represents the evaporation rate.

The original model is modified to improve the representation of the flow processes that occur during flash floods (see section 3.1.1). In addition, two analysis tools of the TETIS results are introduced: virtual tracers tracking convective and stratiform precipitation as well as water paths over or through the soils; and a catchment-state analysis by cell grouping (see Figure 13). The goal is to analyze the spatially distributed response of the watershed to precipitation events of distinct nature.

### 3.1.1 Lateral flow modeling modifications

The TETIS model relies on the concept of mass balance where the storage of tank $i$ at the end of the simulation interval $S_i(t)^*$ [mm] is function of the storage at the start of the simulation interval $S_i(t)$ [mm] and the storage outflow $E_i(t)$ [mm] during the interval $t$, as follows:

$$S_i(t)^* = S_i(t) - E_i(t) \tag{1}$$

The storage outflow $E_i$ is estimated by transforming the storage $S_i(t)$ into an equivalent cross sectional area $A_i$ [m$^2$], as follows:

$$A_i(t) = S_i(t)F_c/L, \tag{2}$$

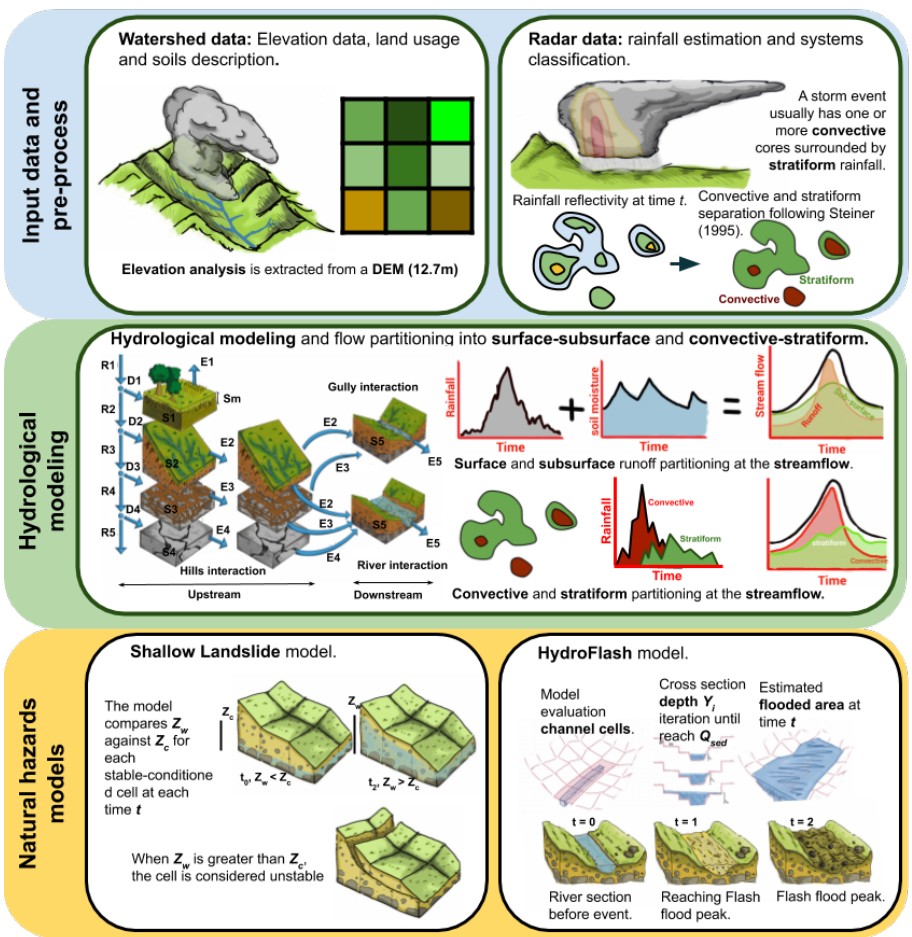

**Figure 6.** Illustrative diagram of the methodology followed in the present study. The top row represents the key input data, specifically a digital elevation model (DEM) and radar-based QPE as the basis of the modeling framework. The second row represents the conceptual basis of the TETIS model. In each cell, five tanks represent the hydrological processes including capillary (tank 1), gravitational (tank 2), runoff (tank 3), baseflow (tank 4) and channel storage tank (tank 5). The state of each tank varies as a function of vertical and lateral flows as shown in the diagram, where the storage is represented by $S_i$ and the vertical input by $D_i$, which in turns depends on the vertical flow through tanks $R_i$. $E_i$ represents the downstream connection between cells and evaporation. The implementation of convective and stratiform rainfall separation and virtual tracers is also portrayed. The implementation of the landslide and HydroFlash submodels are schematized in the bottom row.

where $L$ depends on the model cell width $\Delta x$ [m], $L = \Delta x$ for orthogonal flow and $L = 2\Delta x$ for diagonal flow, and $F_c$ [m$^3$ mm$^{-1}$] is a units conversion factor that is equal to the area of each cell element $A_e$ [m$^2$] multiplied by 1 m/1000 mm. According to Vélez (2001), $E_i$ changes as a function of $A_i$, the flow speed $v_i$ [ms$^{-1}$], and the model time step $\Delta t$ [s], as follows:

$\qquad E_i(t) = A_i(t)^* v_i(t)\Delta t / F_c.$ $\hfill$ (3)

The expression for the cross sectional area at the end of the simulation period $A_i(t)^*$ is found by replacing $S_i(t)$ in equation (2) for $S_i(t)^*$, and then resulting expression and equation (3) into equation (1),

$$A_i(t)^* = \frac{S_i(t)F_c}{L + v_i(t)\Delta t}.$$  (4)

Equation (4) is solved coupled with the equation for the speed $v_i$:

$$v_i(t) = \beta A_i(t)^\alpha$$  (5)

Equation 5 is the generic formulation for the speed used in this work to represent nonlinearities in the relationship between $v_i$ and $A_i$. In the formulation, both, $\beta$ and $\alpha$ change depending on the type of flow: overland, subsurface, base, and channel flow. The solution for $v_i$ is obtained by using the successive substitution method described by Chapra (2012). In the model, we use a 5-minute time step which ensures the stability of the computations. When a solution is reached, $E_i$ is computed using equation (3) and $S_i$ is updated using equation (1).

Nonlinear equations in lateral flows result in a better representation of processes at high resolutions (Beven, 1981; Kirkby and Chorley, 1967). A nonlinear approximation of runoff is presented in equation 6. This approximation is a modification of Manning's formula for flow in gullies. According to Foster et al. (1984), $\varepsilon$ and $e_1$ are a coefficient and an exponent used to translate the Manning channel concept into multiple small channels or gullies. The values of $\varepsilon$ and $e_1$ are 0.5 and 0.64, respectively (Foster et al., 1984). $A_{i,2}$ [m$^2$] is the corresponding sectional area obtained from $S_{i,2}$ by using equation (4). In addition, $M_{i,0}$ is the slope of the cell, and $n_i$ is the Manning coefficient.

$$v_{i,2} = C_7 \frac{\varepsilon}{n} M_{i,0}^{1/2} A_{i,2}(t)^{(2/3)e_1}$$  (6)

The nonlinear equation 7 corresponds to an adaptation of the Kubota and Sivapalan (1995) formula for subsurface runoff $v_{i,4}$, where $k_{i,s}$ is the saturated hydraulic conductivity of cell $i$, and the exponent $b$ is dependent on the soil type, and it is assumed to be equal to 2. $A_{i,g}$ is the equivalent cross-section area of the maximum gravitational storage ($H_{i,g}$ [mm]). $A_{i,3}$ is the corresponding sectional area for the gravitational storage ($S_{i,3}$) obtained by using equation (4). There is also return flow from tank 3 to tank 2, when $S_{i,3} = H_{i,g}$, which represents runoff generation by saturation. In the case of the base-flow, we assume that the speed $v_{i,4}$ is constant for each cell and depends on the aquifer hydraulic conductivity $k_{i,p}$ (see equation 8).

$$v_{i,3} = C_8 \frac{k_{i,s} M_{i,0}^2}{(b+1)A_{i,g}^b} A_{i,3}(t)^b$$  (7)

$$v_{i,4} = C_9 k_{i,p} \tag{8}$$

Finally, the stream flow velocity is calculated by using the geomorphological kinematic wave approximation (Vélez, 2001; Francés et al., 2007), in which $\Lambda$ [km$^2$] represents the upstream area, and $\Omega$ and $\omega_i$, a regional coefficient and regional exponents, respectively

$$v_{i,5} = C_{10} \Omega M_{i,0}^{\omega_1} \Lambda_i^{\omega_2} A_{i,5}^{\omega_3} \tag{9}$$

An extended discussion of the regional parameters can be found in Vélez (2001). The streamflow speed expression is a version of equation (5). This considering that the terms $\Omega$, $M_{i,0}^{\omega_1}$, $\Lambda^{\omega_2}$, and the exponent $\omega_3$ are constant with time.

### 3.1.2 Tools for spatial analysis of the results: virtual tracers and catchment cell grouping

Virtual tracers are implemented in the model to discriminate the streamflow sources into surface runoff and subsurface flow, and to assess the portion of streamflow from convective rainfall and stratiform precipitation, recording the source at each time step and for each cell. The model archives the results of the virtual tracing algorithm at the outlet of the basin and for each reach, enabling to study the different flow paths and water origins at different spatial scales.

The flow tracing module operates in tanks 2 (runoff storage) and 3 (subsurface storage). The module marks water once it reaches either of these tanks, and the runoff-subsurface flow percentage is taken into account once the water enters tank 5 (the channel). At this point, the scheme assumes that the water in the channel is well mixed, implying that the flow percentage is constant until new water enters the channel.

With a similar concept, the model also follows convective and stratiform rainfall. For this, at each time step, the model takes into account the rainfall classified as convective or stratiform and assumes that at each particular cell, the precipitation is either entirely convective or entirely stratiform. This assumption could lead to estimation errors at basins represented by coarse cells (low DEM resolution) where convective and stratiform precipitation are likely to coexist. In the present study, the spatial resolution of the DEM is 12.7 m, higher than the resolution of the radar retrievals, so the potential convective and stratiform rainfall concurrence is very low, and it could not be identified using the Steiner et al. (1995) approach.

Additionally, we propose a graphical method to analyze, at the same time, the evolution of multiple hydrological variables in the entire basin. The first step is to classify all the cells within the watershed in a predetermined number of groups according to their localization and the distance to the outlet. The aim is to establish a coherent and robust spatial discretization, thus allowing the concurrent spatiotemporal variability of the different processes to be summarized in 2D diagrams.

| Parameter Name | Symbol | Scalar Factor | Spatial distribution |
|---|---|---|---|
| Capillarity storage | $Hu = Hu'C_1$ [mm] | $C_1 = 1$ | As a function of the slope |
| Gravitational storage | $Hg = Hg'C_2$ [mm] | $C_2 = 1$ | As a function of the slope |
| Evaporation rate | $Etr = Etr'C_3$ [mm s$^{-1}$] | $C_3 = 0.1$ | As a function of the DEM |
| Infiltration rate | $k_s = k_s'C_4$ [mm s$^{-1}$] | $C_4 = 2.7$ | Lumped |
| Percolation rate | $k_p = k_p'C_5$ [mm s$^{-1}$] | $C_5 = 0.8$ | Lumped |
| System losses | $k_f = k_f'C_6$ [mm s$^{-1}$] | $C_6 = 0.0$ | Lumped |
| Surface speed | $v_2 = v_2'C_7$ [ms$^{-1}$] | $C_7 = 0.5$ | Coefficient of eq (6) |
| Subsurface speed | $v_3 = v_3'C_8$ [ms$^{-1}$] | $C_8 = 1$ | Coefficient of eq (7) |
| Subterranean speed | $v_4' = v_4'C_9$ [ms$^{-1}$] | $C_9 = 0.5$ | Lumped |
| Channel speed | $v_5 = v_5'C_{10}$ [ms$^{-1}$] | $C_{10} = 1$ | Coefficient of eq (9) |

**Table 3.** TETIS model parameters. Primed variables correspond to values prior callibration. Values for the parameters with a scalar factor of 1 are left uncalibrated. Parameters $C_1$ to $C_6$ are not presented in the explanation of the model. $C_1$ modulates the maximum capillary storage, and $C_2$ the maximum gravitational storage. $C_3$ to $C_5$ modulate evaporation, infiltration, and percolation rates, respectively. $C_6$ is assumed as zero, as this variable determines the subterranean system losses. More detail about the calibration parameters is presented at Francés et al. (2007).

### 3.1.3 TETIS model calibration

The TETIS model requires a total of 10 parameters. Table 3 includes all the parameters used in the model. The values of the parameters were derived from the soil properties described in section 2. Due to the lack of detailed information in the region, parameters such as the infiltration and percolation rates are assumed to be constant in the entire basin. Other parameters, such as the capillary and gravitational storages, vary as a function of the geomorphological characteristics of the basin such as the elevation and slope. The calibration consists of finding the optimal scaling for each physical parameter, using a constant value for the entire basin (Francés et al., 2007). The model simulation is set to reach a base flow of 3 m$^3$ s$^{-1}$, a value that corresponds to the discharge measurements during field campaigns days and weeks after the flash flood event, during dry spells. To set the soil wetness initial conditions realistically, the model simulations start two days prior to Event 1. Before this period, there were only a couple of weak rainfall events; for this reason, the overall wetness was set to represent dry conditions at the start of the simulation. Table 3 shows the mean value for all of the parameters used in the model, and the scalar factor adjusted during the model calibration phase. For the 2015 Salgar flash flood reconstruction, we calibrate the evaporation rate, the infiltration, the percolation, the overland flow speed , and the subterranean flow speed (see Table 3). The values for not calibrated parameters are inherited from a local watershed with similar characteristics.

## 3.2 Landslide submodel

The landslide submodel coupled to the TETIS model is proposed by Aristizábal et al. (2016). The stability of each cell is calculated through the assessment of the different stresses applied to the soil matrix. The coupling between TETIS and the landslide submodel is required because the stability of the soil decreases with the pore water pressure (Graham, 1984). The saturated soil depth $Z_{i,w}$ depends on the gravitational storage $S_{i,3}(t)$, the soil wilting point $W_{i,pwp}$, and the soil field capacity $W_{i,fc}$, as follows:

$$Z_{i,w}(t) = \frac{S_{i,3}(t)}{W_{i,cfc} - W_{i,pmp}} \tag{10}$$

When $Z_{i,w}$ is greater than the critical depth $Z_{i,c}$ (equation (11)), failure occurs. The critical saturated depth depends on the shallow soil depth $Z_i$, the soil bulk density $\gamma_i$, the water density $\gamma_w$, the gradient of the slope $M_{i,0}$, the soil stability angle $\phi_i$, and the soil cohesion $C_i'$.

$$Z_{i,c} = \frac{\gamma_i}{\gamma_w} Z_i \left(1 - \frac{\tan M_{i,0}}{\tan \phi_i}\right) + \frac{C_i'}{\gamma_w \cos^2 M_{i,0} \tan \phi_i} \tag{11}$$

Figure 7 describes the variables of the model and the balance of forces considered, and Table 4 presents the required parameters for this model. According to the soil stability definition, the topography and the soil properties, all cells are classified into three classes: unconditionally stable, conditionally stable and unconditionally unstable. In particular, three parameters determine the stability of each cell: (i) residual soil water table $Z_{i,min}$ (equation (12)), (ii) the maximum soil depth at which a particular soil remains stable $Z_{i,max}$ (equation (13)), and (iii) the maximum slope at which the soil remains stable $M_{i,c}$ (equation (14)).

$$Z_{i,min} = \frac{C_i'}{\gamma_w \cos^2 M_{i,0} \tan \phi_i + \gamma_i \cos^2 M_{i,c}(i,0 - \tan \phi_i)} \tag{12}$$

$$Z_{i,max} = \frac{C_i'}{\gamma_i \cos^2 Mi,0(i,c - tan\phi_i)} \tag{13}$$

$$M_{i,c} = \tan^{-1}\left[\tan \phi_i \left(1 - \frac{\gamma_w}{\gamma_i}\right)\right] \tag{14}$$

A cell is unconditionally stable when $Z_i$ is smaller than $Z_{i,min}$ or when the cell slope is smaller than $M_{i,0}$. On the other hand, a cell is unconditionally unstable when $Z_i$ is greater than $Z_{i,max}$, and finally, a cell is conditionally stable when $Z_i$ is between $Z_{i,min}$ and $Z_{i,max}$. Shallow landslides are calculated at each time step of the hydrological simulation, based on the latter cell class, where the soil stability depends on the storm event, becoming unstable when $Z_{i,w}(t)$ is greater than $Z_{i,c}$.

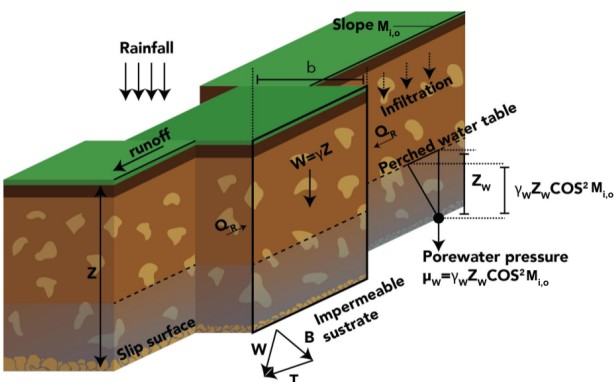

**Figure 7.** Schematic diagram of the landslide submodel. The figure and description are adapted from Aristizábal et al. (2016). $Q_L$ and $Q_R$ are the resultant forces on the sides of the slice of soil.

| Parameter Name | Symbol | Scalar Parameter | Mean Value | Spatial distribution |
|---|---|---|---|---|
| Soil depth | $Z_i$ [mm] | 3.5 | 300 | As a function of the slope |
| Topography slope | $M_{i,0}$ [-] | 1 | 0.01 - 5.3 | From the DEM |
| Soil bulk density | $\gamma_s ed$ [KNm$^{-3}$] | 1 | 18 | Assumed constant |
| Water density | $\gamma_w$ [KNm$^{-3}$] | 1 | 9.8 | Constant |
| Soil stability angle | $\phi_i$ [$^0$] | 1 | $30^0$ | Assumed constant |
| Soil cohesion | $C_i'$ [KN] | 1 | 4 | Assumed constant |

**Table 4.** Landslide model parameters.

### 3.3 Floodplain submodel (HydroFlash)

The HydroFlash submodel is designed to interpret the TETIS simulations as floodplain inundations (Figure 8). For each stream cell and at each time step, the submodel (i) calculates the stream discharge including sediment load (equations 15 - 20, see Takahashi (1991)), and (ii) determines the inundated cells according to the stream cross-profile, the sectional area, and the stream velocities when including the sediment load (equations 19 - 21, Takahashi (1991)). To determine the discharge

including sediment load ($Q_{i,load}$), a realistic channel width is calculated according to Leopold (1953) approach as

$$W_i = 3.26 Q_i^{-0.469} \tag{15}$$

where $Q_i$ corresponds to the streamflow estimated based on a long-term water balance.

    Assuming an infinite sediment and ruble supply, equations 16, 17, 18 are used to deduce, from the

channel width $W_i$, the water level $Y_i$ (equation (16)), the friction velocity $v_{i,fr}$ (equation (16, de-

scribed in Takahashi (1991)), the sediment concentration $c_i$ (equation (18)), and finally the sediment-loaded stream discharge (equation (20)), as follows:

$$Y_i(t) = \frac{Q_{i,sim}(t)}{v_{i,sim}(t)W_i} \tag{16}$$

$$v_{i,fr}(t) = \frac{v_{i,sim}(t)}{5.75log\left(\frac{Y_i(t)}{D_{i,50}}\right) + 6.25} \tag{17}$$

$$c_i(t) = C_{max}(0.06Y_i(t))^{\frac{0.2}{v_{i,fr}(t)}} \tag{18}$$

$$r_i(t) = \frac{1}{D_{i,50}}\left[\frac{g}{0.0128}\left(c_i + (1-c_i)\frac{\gamma_w}{\gamma_{sed}}\right)\right]^{1/2} \cdot \left[\left(\frac{C_{max}}{c_i}\right)^{1/3} - 1\right] \tag{19}$$

$$Q_{i,load}(t) = \frac{Q_{i,sim}(t)}{1 - c_i(t)} \tag{20}$$

where $v_{i,sim}$ and $Q_{i,sim}$ are the simulated velocity and streamflow, respectively. Also, $r_i$ is the constitutive coefficient of the flow, that summarizes the flow dynamics associated with sediments and colliding particles. The above mentioned relationships depend of 2 parameters: the maximum sediment concentration ($C_{max}$ [-]) and the characteristic diameter of the sediments $D_{i,50}$ [m]. Both terms are assumed to be constant and equal to 0.75 (Obrien, 1988) and 0.138 (Golden and Springer, 2006), respectively.

To determine the inundated cells, the flood depth ($F_{i,d}$) and the sectional area of the stream including sediments ($A_{i,load}$) are iteratively calculated by reducing the difference between $Q_{i,load}$ and $\hat{Q}_{i,load}$. The channel cross-section for cell $i$, $E_{i,bed}$, is defined by the DEM. In each iteration $N$, the model updates $F_{i,d}$ with a $\Delta y = 0.1$ m increase. The cross sectional area $A_{i,load}$ is calculated by taking difference between $F_{i,d}$ and the elevation of each cell $j$ in the cross-section $E_{i,bed}$.

$$\hat{Q}_{i,load}(t) = 0.2r_i(t)(N\Delta y)^{\frac{3}{2}}S_{i,0}A_{i,load}(t) \tag{21}$$

$$F_{d,i}^N = F_{d,i}^{N-1} + \Delta y \tag{22}$$

$$A_{i,load}^N = \Delta x \sum_{j=1}^{N} F_{i,j,d}^N - E_{i,j,bed} \text{ with } E_{i,j,bed} < F_{i,j,d}^N \tag{23}$$

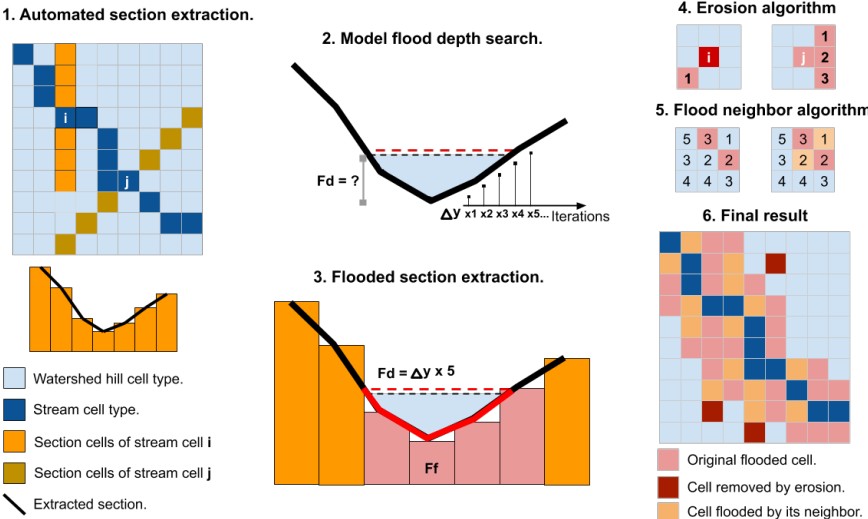

**Figure 8.** Illustrative diagram of the HydroFlash submodel scheme. Step 1. The submodel extracts the cross-profile from the network considering the DEM and flow direction. Step 2. Based on equation (21), the submodel obtains the first approximation of the flash flood streamflow; then, the flood depth and the cross-section area are obtained from equations (22) to (21). Step 3. The submodel obtains the flooded portion of the cross section. Step 4. Erosion post-process. Step 5. Filling post-process. 6. The final result for a time step $t$.

The resulting flood maps might include the presence of small isolated flood spots and discontinuities where the flow direction changes from orthogonal to diagonal across or vice versa. We included two post-processing steps to correct these issues by (i) using an image processing erosion algorithm (Serra, 1983) to remove the small and isolated flood spots (step 4 in Figure 8), and, to solve the flow direction discontinuities, (ii) for each flooded cell the model seeks to inundate the eight neighboring cells: A neighbor cell is also flooded if the altitude of the original flooded cell, plus the flood depth, is higher than its elevation (step 5 in Figure 8). The image erosion is performed once with a 3 by 3 kernel. An example of the final result for a time step $t$ is shown in the step 6 in Figure 8.

## 4   Results

The main results of the present study include the reconstruction of the 2015 Salgar flash flood, the assessment of the importance of soil moisture in the hydrological response of the basin, and the evaluation of the relative role of stratiform and convective precipitation cores in the generation of the observed extreme event. This section is based on the analysis of the hydrological simulation, as well as the occurrence of shallow landslides and flash floods, and their simulation. A comparison of the results from both submodels and the observed landslide scars and flooded spots allows to evaluate the overall skill of the proposed methodology.

### 4.1 TETIS validation and sensitivity analysis

Figure 9a presents the results of the hydrological simulation at the outlet of the basin. The simulation shows that Event 1 generates a hydrograph with a peak flow of $Q_{max} = 160$ m$^3$ s$^{-1}$. It is important

to note that during Event 1, there were no damage or flooding reports by local authorities. Even though this precipitation event did not generate flooding, it set wet conditions in the entire basin before the occurrence of Event 2 (see the purple line in Figure 9b representing the capillary storage). Additionally, it is clear from the simulation that during the flash flood event, the two successive convective cores over the same region (training convection) generated a peak flow of $Q_{max} = 220$ m$^3$

s$^{-1}$, a value that is in the upper range of the estimated streamflow based on post-event field evidence (185-222 m$^3$ s$^{-1}$). Figure 9a also presents the simulated runoff and subsurface flow separation as well as the convective-stratiform-generated discharge discrimination. The modeling evidence during Event 2 suggests the convective rainfall fraction dominates the hydrograph formation. In both events, convective (stratiform) precipitation appears to be closely related to the simulated runoff (subsurface

flow). The simulated subsurface flow is more important in magnitude than that runoff in describing Event 1, while runoff is more relevant for Event 2. Figure 9b presents not only the capillary storage (purple), but also the runoff (continuous blue) and the gravitational storage (dashed blue) temporal variability, as represented by the proposed model. As expected, runoff storage is only nonzero during the storm duration, while gravitational storage increases considerably during rain events, followed

by a slow recession. There is an increase in basin-wide capillary storage during Event 1, remaining considerably high during the time leading to the occurrence of Event 2. According to the model simulations, the peak flow occurred at 2:20 a.m. LT on May 18th, which is accurate compared to the reports from local authorities (between 2:10 and 2:40 a.m. LT), considering all the data limitations.

Figure 10 shows the results of a sensitivity analysis of the hydrological simulation during the

second rainfall event, varying the surface speed, infiltration rate, and the subsurface speed factors. The aim of the sensitivity analysis is to evaluate the robustness of the overall results, considering the fact that the quality and quantity of some of the watershed information is limited. In the sensitivity analysis, we vary the surface speed factor between 0.01 and 20, the infiltration factor between 0.02 and 20, and the subsurface speed factor between 0.1 and 10. The overall sensitivity results show that

the main findings described in the previous paragraphs are, in fact, robust to almost all changes in the mentioned parameters, with the surface runoff associated with convective rainfall controlling the magnitude of the peak discharge during the Event 2. The model's highest sensitivity, and hence the largest uncertainty source, appears to be related to the surface speed parameter (Figure 10a), particularly during the peak flow and the early recession. On the other hand, changes in the infiltration rate

factor (Figure 10b) and subsurface velocity factor (Figure 10c) are associated with with a simulation sensitivity smaller than 7 and 20% of the peak flow, respectively.

After the flash flood event, a stream gauge level station was installed near the outlet of the basin (see Figure 2). We use these records to validate the model results without further calibration. Since

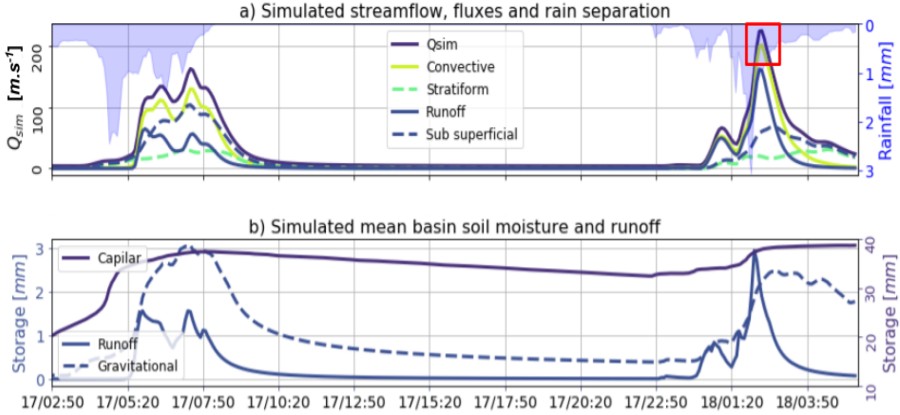

**Figure 9.** Summary of the results from the TETIS hydrological simulation. a) Simulated streamflow, convective-stratiform-generated discharge discrimination, and runoff and subsurface flow separation. The red square represents the flash flood peakflow interval that is estimated based on field campaign evidence. b) Basin average capillary, runoff and gravitational storages during the simulation period.

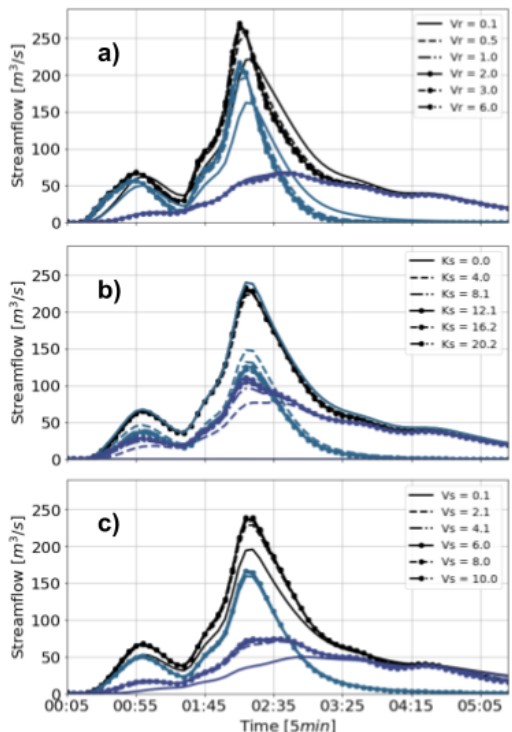

**Figure 10.** Hydrological simulation sensitivity analysis. Similarly as in Figure 9, all panels show the simulated streamflow (purple), and the runoff (green) and subsurface flow (dashed purple) separation. From top to bottom, the panels show the simulation sensitivity to changes in the a) surface speed, b) infiltration rate, and c) subsurface speed factors.

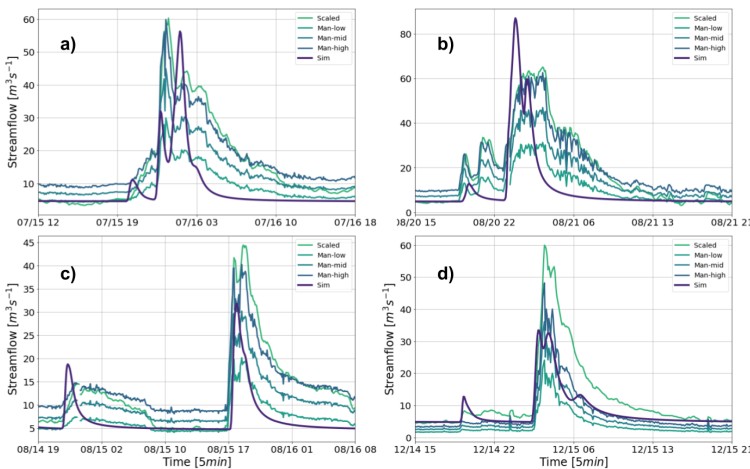

**Figure 11.** Comparison between TETIS simulations and streamflow estimations from a stage level station installed on a bridge at the outlet of the basin (see Figure 2)

.

the observed series correspond to stage level records, the streamflow estimation is performed follow-
ing two different approaches. The first approach, the empirical one, consists of subtracting the 10th
percentile of the observed stage time series from the observational record, and the 10th percentile
of the simulated streamflow, from the same series. On the other hand, the second method uses the
Manning formula. For this, we consider the geometry of the section in Figure 4, and the slope from
the DEM. Additionally, due to the potential uncertainties, we consider three different Manning val-
ues (0.015, 0.02, 0.03). Figure 11 shows the estimated streamflow using the two methods for four
different hydrographs during July, August (2 events) and December 2015. The simulated magni-
tudes appear relatively close to the observations, and the peak discharge time is captured skillfully in
three of the four cases presented. The discharge values using the "high" Manning number estimation
(0.015) are similar to the empirical method. The performance of the model is acceptable (Figure
11), considering the lack of calibration, the size of the basin, and the magnitude of the recorded
events. The results shown include cases where the peak flow was over-estimated (panels c and d),
and under-estimated (panel b).

Figure 12 shows the temporal evolution of discharge during Event 2 in different locations along
the watershed's main channel. The upper location corresponds to 15% of the area of the basin, and
the other downstream locations correspond to 52%, 76%, and 100% of the watershed. The difference
in the time of the peak discharge between the upper location and the outlet of the basin is around 35
minutes, which is plausible with travel speeds between 5 and 7 ms$^{-1}$ and an effective distance of 14
km. In terms of volume, about 737,000 m$^3$ of the total 1,438,000 m$^3$ simulated at the outlet of the
basin are generated on the 15% upstream part of the watershed, corresponding to about half of the
total mass. In terms of peak flow, due to the slope and velocity changes, the simulated discharge at

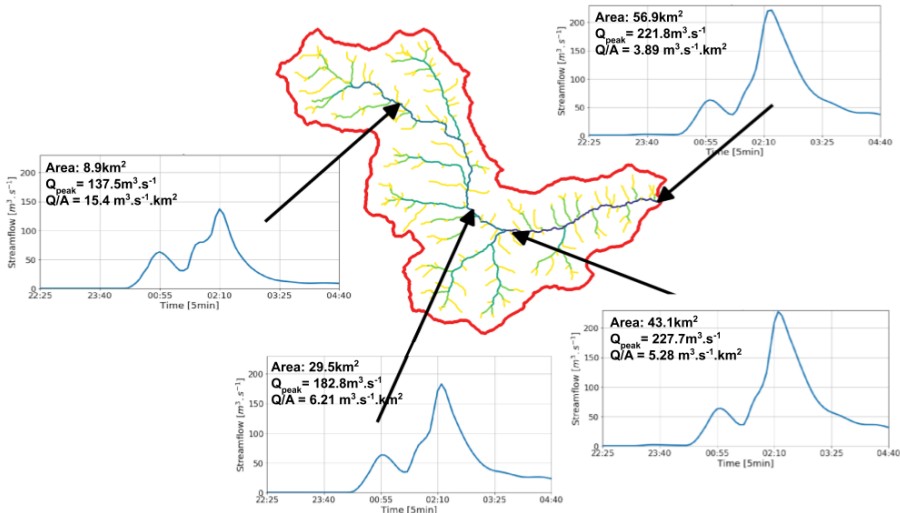

**Figure 12.** Temporal evolution of discharge during Event 2 in different locations along the watershed's main channel. The upper location corresponds to 15% of the area of the basin, and the other downstream locations correspond to 52%, 76%, and 100% of the watershed, respectively.

the 15% upstream part of the watershed corresponds to 50% of the peak discharge at the outlet of the basin.

### 4.2 Flash flood processes

Figure 13 presents the proposed 2D diagrams obtained for the simulation of the La Liboriana basin flash flood using a spatial discretization with 50 groups. Figure 13a includes the evolution of the average rainfall over the basin (black line), and the spatiotemporal evolution of capillary storage (filled isolines) and return flow (colored isolines from white to red) by groups. For the analysis, it is relevant to highlight that higher numbered groups are located away from the outlet of the basin and correspond in this case to considerably steeper slopes. Figure 13b presents the evolution of stream-flow at the outlet of the basin (black line), as well as the gravitational storage (filled isolines) and runoff (colored isolines) spatiotemporal evolution. Figure 13 shows variations in the capillary and gravitational storages associated with Event 1 in the higher numbered groups. The capillary storage remains high in almost all the basin until the start of Event 2. According to the conceptualization of the model, the gravitational storage and surface runoff start to interact when the capillary storage is full. In this case, this situation is set up by Event 1. The model runs for Event 2 using dry initial states, show no flooding in the results.

The temporal variability of rainfall intensity plays an important role in the hydrograph structure. During Event 1, rainfall accumulated over the basin at a relatively stable rate (Figure 14a). On the

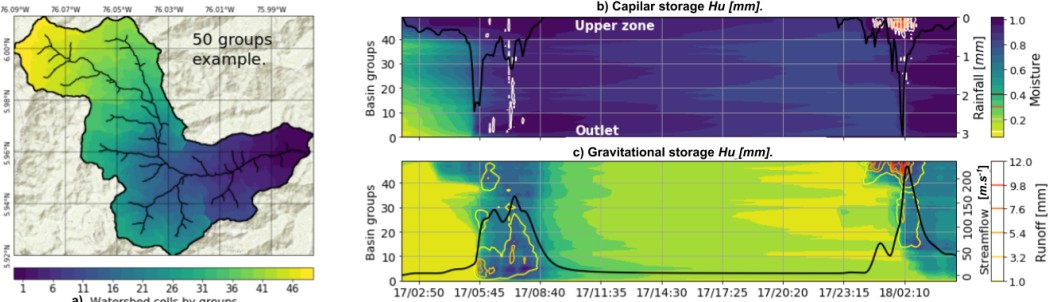

**Figure 13.** a) Example of watershed grouping as a function of of their localization and distance to the outlet for La Liboriana basin using a 50-group categorization. b) Simulated capillary moisture (filled green-to-blue contours) and returned flow occurrence (white to red isolines). The black line represents the average rainfall over the basin. c) Simulated gravitational moisture (filled green-to-blue contours) and runoff (yellow-to-red isolines). The black line represents streamflow at the outlet of the basin. The green-to-blue color bar serves as a reference for capillary moisture and gravitational water content.

other hand, Event 2 presents a significant increase in rainfall rate in the second half of the life cycle (Figure 14b). This change in precipitation intensity is associated with a considerable enhancement of the training convective cores due to orographic effects. Events 1 and 2 also exhibit differences in the elapsed time between rainfall occurrence and streamflow increment given the relative timing of stratiform versus convective rainfall (see the gray band in Figure 14a and b). We compute the elapsed

time between the rainfall and the simulated streamflow by measuring the time differences between the lines for the cumulative rainfall and streamflow in Figure 14. For Event 1, the median elapsed time between rainfall and streamflow ($Et_{p50}$) is 1.12 hours, while for Event 2, $Et_{p50}$ is 0.79 hours. The median elapsed time between the convective portion and the streamflow ($Etc_{p50}$) in Event 1 is 0.75 and 0.46 in Event 2. The minimum value of the convective elapsed time $Etc_{min}$ also descends

from 0.42 to 0.25 hours. On the other hand, there is an increase in median elapsed time between stratiform rainfall and streamflow ($Ets_{p50}$) from 1.21 to 1.83 hours. The observed differences are largely due to the timing of the convective precipitation during each of the events. During Event 1, the convective precipitation occurred near the end of the event, explaining the delayed peak discharge time (see Figure 5).

According to Figure 14b for Event 2, the accumulations of streamflow runoff and convective rainfall become similar with the increase in time. This fact highlights the strong control that, in this case, the convective portion has on the runoff, with almost no effect of the stream network filtering out the convective signal, most likely due to the size and the rapid response of the basin. This description, however, only applies for the runoff portion, since the evolution is different when

we consider the total simulated streamflow.

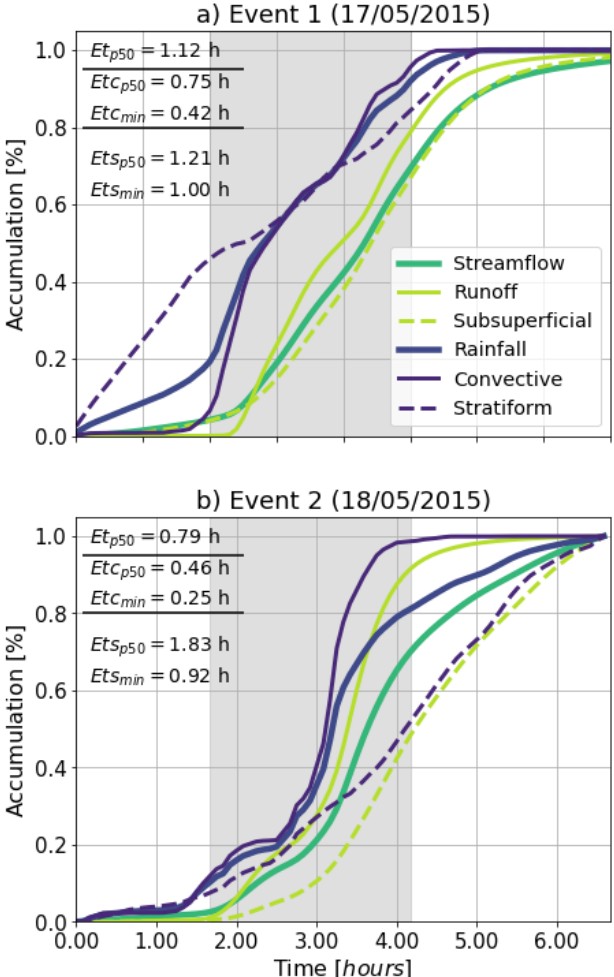

**Figure 14.** Accumulated rainfall and streamflow for a) Event 1 and b) Event 2. The accumulation is expressed in percentage with respect to the total value in each case. The median elapsed time and minimum elapsed time are estimated between total ($Et_{p50}$, $Et_{min}$), convective ($Etc_{p50}$, $Etc_{min}$), and stratiform ($Ets_{p50}$, $Ets_{min}$) rainfall and the runoff portion of the streamflow. Gray bands correspond to the periods for elapsed time estimation.

### 4.3 Landslide and flood simulations

Figure 15a presents the observed landslides triggered by Event 2 based on aerial photos and satellite images taken before and after the flash flood. Figure 15b shows, by hills, the map of total unstable cells during the simulation period, and Figure 15c shows the time series of the number of simulated unstable cells during Event 2 (continuous purple line) and the mean rainfall over the basin (inverse axes, blue line). Calibration of the landslide submodel was performed by finding the maximum overlap between simulated and observed unstable and stable cells, and at the same time reducing the overall number of false positives and false negatives. It is important to note that the calibration strat-

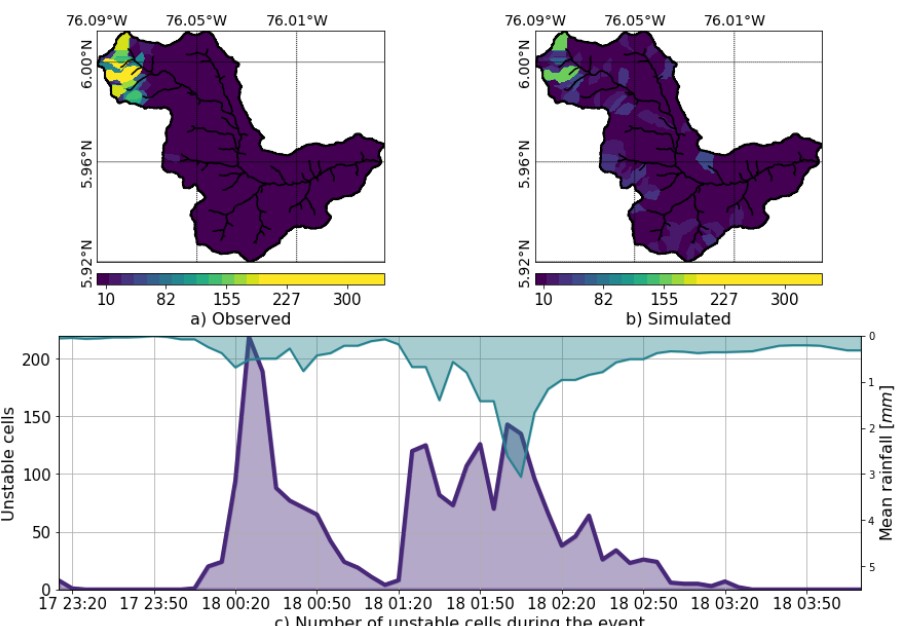

**Figure 15.** a) Observed landslides triggered by Events 1 and 2. The figure is based on aerial photos and satellite images taken before and after the flash flood event. b) Map of total unstable cells during the simulation period. c) Time series of the number of simulated unstable cells during Event 2 (continuous purple line) and mean rainfall over the basin (inverse axes, blue line).

egy is not a cell-by-cell modification of the parameters involved but rather a basin-wide modification of soil properties. A sensitivity analysis of soil parameters is carried out by making small variations of the variables within specified intervals: $\phi$ between 25 and 32, $\gamma$ between 17 and 19, $C'$ between 3.5 and 4.2, and $Z$ between 0.1 and 3 m. The sensitivity analysis suggests that slight variations in the parameter in $Z$ produce the largest changes in the number of unstable cells. Following Table 1, the average soil depth in the basin is only 0.3 m, a value that likely corresponds to underestimation according to the inspections during field visits. For this reason, the results presented in Figure 15 use a $Z$ map scaled by a calibration factor of 3.5, preserving the spatial dependence on the slope but achieving a more realistic soil depth and better spatial distribution of a landslide occurrence. The model represents the spatial distribution of the areas that are prone to trigger shallow landslides during Event 2 reasonably well, especially in the upper part of the basin, showing a significant density of unstable cells in the hills where slides took place.

Figure 16 shows the identification of the flood spots at the peak of Event 2 (May 18, 2015, 2:00 a.m.) as simulated using HydroFlash. Figures 16b to f present a detailed view of the results from the outlet of the basin to the upper region. Cases presented in Figures 16e and f exhibit a satisfactory agreement with observed flood spots (blue shadow). Cases in Figures 16c and d also show a good approximation, but with minor spatial shifts in some sections. The largest spatial differences are

observed in Figures 16b. At the entrance of the urban zone, the model overestimates the flood spots. The model results indicate that 11% of flood spots occur at elements of order 1 and 2, and 18, 38 and 32% occur at orders 3, 4 and 5, respectively. Table 5 summarizes the described percentages and the total length of each order. These results also highlight a coherent geomorphological representation of the flooded channels and hills relative to the order.

**Table 5.** Channels and flooded cells percentages summary. $Sh_0$ and $Ss_0$ correspond to the mean hill and stream slope, respectively. $L$ corresponds to the total channel length. F Spots and S spots correspond to the flooded and slides percentages, respectively.

| Order | $Sh_0$ [%] | $Ss_0$ [%] | $L$ [km] | F Spots | Ss Spots |
|---|---|---|---|---|---|
| 1 | 60 | 37 | 59 | 5 | 64.5 |
| 2 | 57 | 27 | 26 | 6 | 26.3 |
| 3 | 49 | 13 | 16 | 18.5 | 5.5 |
| 4 | 43 | 9 | 10 | 38.5 | 3.6 |
| 5 | 42 | 6 | 6 | 32 | 0.05 |
| **Mean/total** | 50 | 18 | 117 | 100 | 100 |

## 5   Discussion

On the morning of May 18, 2015, a flash flood occurred in the steep La Liboriana basin, in the municipality of Salgar, Department of Antioquia, Colombia, leaving more than 100 human casualties, 535 houses destroyed, and significant infrastructure loses. Due to the lack of local information of soil type, land use and real-time hydrometeorological data, the La Liboriana case implies a challenge for flash flood prediction, modeling and, consequently, risk management. The present paper introduces a hydrological model-based approach and an integral graphical analysis tool (an integrated spatiotemporal analysis of rainfall evolution, together with soil storages in the basin) for the following purposes: 1) to simulate and understand the soil-rainfall-discharge processes that led to the 2015 Salgar flash flood, and 2) to propose it as a radar QPE-based and modeling-based landslide and flash flood guidance low-cost tool for basins with scarce data and regions with limited resources.

The methodology implies changes and additions to the TETIS distributed hydrological model including tracking independently convective and stratiform precipitation within the model, as well as keeping track of the runoff and subsurface portions of the streamflow. TETIS was coupled with a shallow landslide submodel and HydroFlash, a one-dimensional floodplain scheme. The model proposed here indeed allows studying the different hydrological processes relevant to flash flood and landslide occurrence by using different simulation resources, serving as the basis for a better understanding of the overall basin response. Despite the lack of data, the evidence suggest that the results represents, to a large degree, the magnitude of the disaster; considering also that the simulated

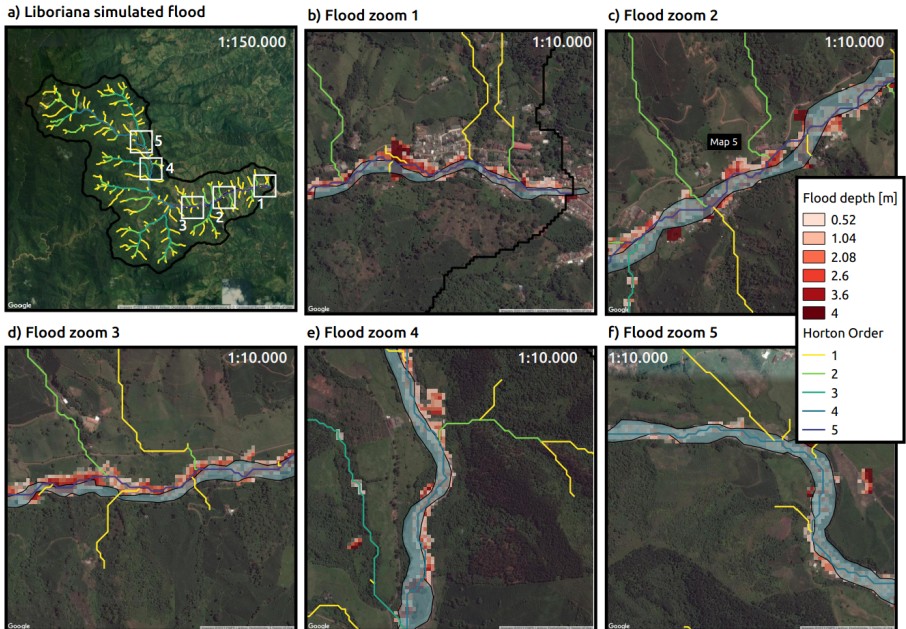

**Figure 16.** Simulated flood spot at the peak of Event 2 in different locations (Image courtesy of CNES/Airbus).a) Basin drainage network. White squares correspond to regions of interest highlighted in panes b) to f). The colors of the streams correspond to the Strahler order of the network. b) Zoom at the outlet of the basin, where an important portion of the human and infrastructure losses took place. c) Zoom at La Margarita settlement also affected by the flash flood. d) to f) Zoom at key locations along the principal stream. Observed flood spots are shown in blue polygons and model flood spots in red to white grids.

peak flow is consistent with the peak flow envelope proposed by Gaume et al. (2009) for flash floods. This approach helps to examine the first-order flood-generating mechanisms or causative factors both in time and in space, focusing on the most important physical processes (Klemes, 1993; Merz and Blöschl, 2003), potentially allowing the anticipation of flash flooding events, the issue of warnings, and response by risk management entities.

The evolution of the simulation of Events 1 and 2 show evidence of remarkable behavioral differences. During Event 1, both gravitational and capillary tanks are filled along and across the basin as a result of the quasi-homogeneous rainfall spatial distribution. Zoccatelli et al. (2011) found similar results for watersheds in Europe with areas ranging between 982 and 52 $km^2$. The return flow is low, and most of the runoff occurs within the first 20 groups (40% of the watershed closest to the outlet).

In the period between both events, there is a recession in the capillary and gravitational storages in the entire basin. Capillary storage decays considerably slower than gravitational storage. During Event 2, the flash flood triggering event, the first convective core saturates both capillary and gravitational storages in the upper part of the basin and generates both return flow and significant runoff. Due to soil saturation, the second convective core results mainly in surface runoff. During this event,

extreme runoff rates are evident in the upper part of the basin, collocated with the steeper slopes. On the other hand, subsurface flow is more important in magnitude than runoff describing Event 1, while runoff is more relevant for Event 2. The precedent storage and the presence of thunderstorm training profoundly condition the streamflow during Event 2. The overall evidence suggests that precedent capillary moisture in the basin plays an essential role in modulating river discharge. This behavior

could be linked to the temporal occurrence and relative importance and timing of stratiform and convective formations previously described. During the extreme event, when the soils were already wet, the convective rainfall fraction dominated the hydrograph formation. While stratiform rainfall plays an important role moistening the entire basin, convective rainfall generates considerable runoff, leading to flash flooding. Several authors have argued about the role of convective rainfall trigger-

ing flash floods (Doswell et al., 1996; Kahana et al., 2002; Schumacher and Johnson, 2005; Delrieu et al., 2005; Šálek et al., 2006; Milelli et al., 2006; Rozalis et al., 2010; Fragoso et al., 2012; Berne and Krajewski, 2013; Gochis et al., 2015; Bruni et al., 2015; Piper et al., 2016; Douinot et al., 2016; Llasat et al., 2016; Baltaci, 2017), however, to our knowledge no other study has tracked convective and stratiform water in a modeling setting to explore their relative role leading to flash flooding.

While convective and stratiform partitioning influence the runoff and subsurface flow separation, the spatial distribution of rainfall relative to watershed network morphometry structure also imposes a condition on the hydrological response of the basin (Douinot et al., 2016). In other words, hydrograph formation is determined not only by the rainfall accumulation or maximum intensity but also by its spatial structure (Zoccatelli et al., 2011; Douinot et al., 2016). As mentioned before, av-

erage rainfall accumulation over the basin for Events 1 and 2 is 47 mm and 38 mm, respectively. During Events 1 and 2, convective (stratiform) average accumulations are 28 (23) and 17 (14) mm, respectively. The maximum rainfall intensities are relatively similar with 150 mm/h and 180 mm/h for Events 1 and 2, respectively, but the location was significantly different. Convective rainfall occurrence at the upper subbasins has significant implications due to geomorphological conditions

associated to zero-order subbasins (Sidle et al., 2018). Besides, at Event2 with a moist soil, the convective portion of the rainfall significantly influences the hydrograph formation. Additionally, when we compare Events 1 and 2, there is an interplay between the rainfall spatial structure and the soil storage capacity. During Event 1, there is almost no saturation, hence runoff production is low, while Event 2 is influenced by the pre-event water and the occurrence of multiple convective systems over

the same region. The structure of the rainfall associated with the La Liboriana event and its interaction with the soils highlights the need to consider in more detail the role of orographic rainfall intensification in practical applications such as early warning systems. Evidence suggests the spatial structure of the rainfall is at least as important as the geomorphological features of the basin in regulating the generation of flash flood events.

An integrated spatiotemporal analysis of rainfall evolution, together with soil storages in the basin, is necessary to study the relevance of antecedent conditions and precipitation type, intensity, and

location in the generation of flash flood events. Event 1 increased the overall soil moisture with an associated decrease in infiltration rates, similar to the results reported by Marchi et al. (2010); Penna et al. (2011) and Zehe et al. (2010); additionally, low infiltration increased the runoff rates, ultimately affecting the susceptibility of the basin to flash flood occurrence (Wagner et al., 1999; Penna et al., 2011; Tramblay et al., 2012). Due to geomorphological characteristics (see Table 5), water tends to reach faster the channels in hills of order 1 and 2, and, at the same time, the sediment production and transport in these hills tend to be larger. Order 3 subbasins most likely act as transport elements, with no important energy losses (Table 5), and floods tend to occur at order 4 and 5 subbasins due to the widening of the channel and slope attenuation.

Different authors have focused on trying to understand the general causative factors behind the occurrence of flash floods (Marchi et al., 2010), also suggesting a significant combined role of geomorphology, orography, soil characteristics and local convection. For example, Lehmann and Or (2012), using a shallow landslide model, found an important role of the topography and the rainfall conditions. Turkington et al. (2014) showed how intense locally driven convection is the main meteorological trigger for flash occurrence in the French Alps.Camarasa-Belmonte (2016) showed the important role of rainfall intensity and duration on the shape of the hydrograph, with intense rainfall shortening the response time of the basin, and large durations increasing the flood peak. In the Mediterranean region, Boudou et al. (2016) stated that in addition to the rainfall, geomorphological characteristics and antecedent soil conditions are key in the generation of flash flooding.

The landslides submodel presents an overall acceptable performance with limitations in certain regions. In particular, there are some false positives in the middle of the basin. These limitations could be associated with the assumptions and approximations inherent to the submodel, including that it only determines unstable cells by slowly filling the soil matrix with water, which, in this case, given the lack of information, depends on the soil depth derived from the topography, and that the model does not consider instability due to intense rainfall events. The lack of detailed soil depth information could explain the false positives landslides. On the other hand, the relation between landslides and high-intensity rainfall must be explored and included in this kind of models. There is also an apparent contradiction regarding the depth of the soils in the basin: While the values derived from topography appear to work well for the hydrological model, the depth had to be calibrated to obtained a better representation of landslides. There are two possible explanations for the contradiction, (i) that the soils are in fact thicker in the entire basin, but the calibration of the infiltration and percolation rates corrected the hydrological simulations, and (ii) that the landslides submodel is too simplistic, or that no other parameters were calibrated, possibly resulting in over calibration of the soil depth. This is an aspect that needs to be explored further.

The landslide submodel has been used in a nearby watershed with similar characteristics, but with high-quality distributed information (Aristizábal et al., 2016). In that case, the model shows a better performance, which highlights the relevance of the quality of the input data. It is also important to

consider that, a pinpoint localization of the unstable cells is still considered a hard task, in part due to the small temporal and spatial scale at which landslide processes take place (Aristizábal et al., 2016; Dhakal and Sidle, 2004; Wu and Sidle, 1995).

Similarly, results of the HydroFlash submodel are satisfactory despite the hydraulic over simplifications, and are potentially useful for issuing warnings to the community. From that point of view, it is important to stress that the low computing cost of HydroFlash, different to that of detailed 2D/3D hydraulic geomorphological models, makes it possible to be executed in real time coupled with rainfall observations, providing valuable information that, while not 100% accurate spatially, helps discriminating to a high degree, for example, which communities need to be evacuated given an extreme event. In addition, the floodplain submodel provides an indirect estimation of the sediment load during extreme events. In the 2015 Salgar simulations, the peak discharge obtained with the hydrological model was 220 m$^3$s$^{-1}$; the total streamflow considering the sediment load reached values around 285 m$^3$s$^{-1}$, for a $Q_{sed}/Q_{sim}$ ratio of 1.3. The extra 30% discharge corresponding to the sediment load is certainly a relevant contribution to the total discharge, with impacts in the floodplain determination. Considering the stream network slope, the simulated ratio is comparable with reports in the literature Rickenmann and Koschni (e.g. 2010); Chang et al. (e.g. 2011); **?**. The sediment load is mainly constrained by the maximum sediment concentration $C_{max}$ and the depth of the flow, suggesting that better information about $C_{max}$ could improve the simulation of flood spots. It is important to note that the model was calibrated considering only the water discharge since the uncertainty in the erosion processes and their representation in the model is still significant, and the fact that only 1D processes in the channel are taken into account: The hydrological model relies on water balance. In spite of that, and in order to be in the conservative side for risk management applications, the potential increase in the total discharge associated with the sediment load is considered.

However useful, the evidence in this work only takes into account two successive events; an analysis of more cases and different spatial scales (different basins) would provide robust conclusions in this direction. It is clear that focusing on a single extreme event, rather than on a spectrum of floods, is not conclusive enough Merz and Blöschl (2003). The model simulation results suggest it is imperative to study in depth the long-term link between the relative basin and drainage network orientation and the preferred path of precipitation events and its role in defining the frequency of flash flood occurrence. A better understanding of the network-hills-preferential rainfall advection structure could provide information about basins prone to flash floods when information is scarce.

## 6 Conclusions

Extreme rainfall events such as the one that triggered the La Liboriana tragedy frequently take place in Colombia and the entire global tropical belt over ungauged basins, often triggering flash floods

and debris flows, which endanger vulnerable communities due to poor long-term planning and lack of functional early warning systems. There is a global need for better knowledge and understanding of the hydrological and meteorological conditions that, combined, lead to the manifestation of disasters linked to natural hazards. Such an understanding must result in useful practical applications that improve risk management practices and thus save lives. In the current work, we approach the problem from a hydrological modeling point of view, trying, despite the data limitations and the uncertainty of the results, to shed some light on the first-order processes that modulate the occurrence of flash floods in the region of study.

In the case of the La Liboriana flash flood, radar reflectivity fields were available from a C-Band radar operated by the Early Warning System of Medellín and its metropolitan area, as part of a local risk management strategy. While the municipality of Salgar is located far from Medellín's metropolitan area, the radar is 90 km away from Salgar, and the reflectivity retrievals enable the classification of precipitation fields into convective and stratiform areas, using widely accepted methodologies by the meteorological community. Radar reflectivity also serves as a proxy for precipitation, allowing a quantitative estimation of rainfall fields. This estimation was used together with the TETIS model to assess the different basin-wide processes taking place during the flash flood triggering rainfall event. The limitations of the methodology presented in this work do not allow representing all the detailed small-scale preferential pathways of the water in the watershed, but rather focus on the first-order processes to study the partitioning between runoff vs. subsurface flow. Additionally, the model results are used to obtain a conceptual idea about the general processes, but it must be taken into account that the simulations are subject to a calibration process that could lead to erroneous conclusions about the mentioned processes. This consideration could be true even considering that different steps were taken trying to avoid this situation.

The overall model simulation methodology reproduces the estimated magnitude and reported timing of the La Liboriana flash flood discharge peak quite well, showing robustness to changes in the most important model parameters. Simulation results suggest that the soil storage capacity available before flooding event, impacted not only the flood magnitude itself, but also the response time of the catchment, highlighting the role of soil wetness distribution within the basin. The model also reproduces the areas of regional landslide occurrence and flood spot locations satisfactorily. The model simulation results indicate that the flash flood and the regional landslide features were strongly influenced by the observed antecedent rainfall associated with a northwesterly stratiform event that recharged the gravitational and capillary storages in the entire basin. The TETIS model simulation shows that the antecedent event set wet conditions in the entire basin before the occurrence of the flash flood event, governing the streamflow during the latter. The results of the model simulation also suggest that the first of the two successive convective cores (training convective elements) over the same region during the second precipitation event (the flash flood event) saturated both capillary and gravitational storages in the upper part of the basin and generated both return flow and signifi-

cant runoff. The second convective core resulted mainly in surface runoff spatially collocated with the steeper slopes, generating the kinetic energy needed to produce the La Liboriana flash flood. The overall results also show a good agreement between the simulated flood spots and the observed ones, despite the limitations imposed by the resolution of the DEM used for extracting cross-sections and the model oversimplifications.

Results of the landslide submodel and HydroFlash, while satisfactory, are far from perfect, showing significant differences compared to observations. The evidence suggests, by and large, that most of the observed differences are mainly due to the lack of higher spatial resolution DEM, in the case of HydroFlash, and due to the lack of a detailed soil dataset, in the case of the landslide submodel. However, there is also is considerable room for improvement in both submodels, including a better representation of non-Newtonian hydraulic processes in HydroFlash, and a direct link between landslides and flood spots following, for example, a similar strategy to the one presented in the STEP-TRAMM model (Fan et al., 2017). Notwithstanding the difficulties, the results suggest that the submodel simulations could have been used and should be used in the future for early detection and warning to improve both short- and long-term risk reduction strategies.

Considering all the shortcomings and generalizations, the described model-based approach is potentially useful to assess flood-generating mechanisms and as a tool for policy-makers, not only for short-term decisions in the context of an early warning system but also as a planning resource for long-term risk management. The results suggest it is possible to use low-cost methodologies such as the one introduced here as a risk management tool in countries and regions with scarce resources.

## 7  Acknowledgments

This work was supported by SIATA (Sistema de Alerta Temprana de Medellín y el Valle de Aburrá) funds provided by Area Metropolitana del Valle de Aburrá (AMVA), Municipio de Medellín, Grupo EPM, and ISAGEN under the Research and Technology Contract CD511, 2017. Universidad Nacional de Colombia partly funded Nicolás Velásquez under the Facultad de Minas graduate scholarship program. Both authors would like to thank anonymous reviewer 1 for the detailed and insightful comments that helped to clarify and highlight the message of this work. Both authors also thank Dr. Eric Gaume, reviewer 2, for his thoughtful comments.

For the technically inclined reader, the TETIS hydrological model and submodels are written in Fortran 90, and the interface to the model, pre-process, and post-process tools are in Python 2.7. The Fortran code is warped to Python using **f2py** (Peterson, 2009), and it is publicly available under the Watershed Modeling Framework **WMF** in a web repository (**GitHub**).

## Appendix A:  Figures

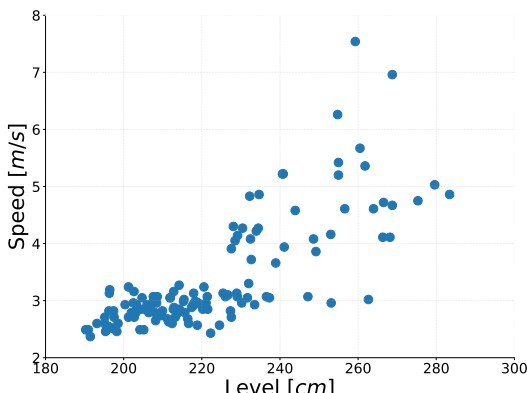

**Figure A.1.** Scatter plot of water level (depth) (cm) and surface speed (ms$^{-1}$) for Doña María basin, located in the Aburrá Valley (Basin outlet coordinates: 75.651ºW, 6.190ºN). The basin slope is 34.09%, the area :72.84 km$^2$, and the maximum (minimum) height is 2,835 m.a.s.l. (1,562 m.a.s.l.)

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
