# Peer review of "Reconstructing the 2015 Salgar Flash Flood Using Radar Retrievals and a Conceptual Modeling Framework in an Ungauged Basin"

_Hydrology and Earth System Sciences, 2018_

## Referee Comment (RC1) · Anonymous Referee #1 · 22 Oct 2018

The paper of Nicolás Velásquez et al., entitled 'Reconstructing the Salgar 2015 Flash Flood Using Radar Retrievals and a Conceptual Modeling Framework: A Basis for a Better Flood Generating Mechanisms Discrimination' addressed: i) the flash flood forecast issue, specially assessing flood plain and landslide occurrences, and ii) the understanding of flood processing mechanisms during two contrasted flash flood events, using virtual tracers of water origin and paths.

The authors proposed a novel and ingenious method as it is adapted to the data scarcity of the region. The overall method is specially of interest as it does not pro-

vide only flood flows, but also the localization of flood plains and landslide, which are critical information for stakeholders. Finally, the work provided a rare and interesting double flash flood event study located in the Northern Andes mountains, which actually supports numerous extreme hydrological events but are scarcely documented.

However first at all, the paper having two distinct objectives, it suffers from clarity. The descriptive potential of the model, tracing water origins and paths, as well as the 'low cost' method for assessing floodplain and landslides are both interesting works, but both are totally independent. I would suggest to either follow the process understanding objective or the forecast one.

In addition, the applied method suffers from a lack of validation and the limits of the results are poorly explained. While all the results are tributary of the hydrological model, the maximal peak discharge, used as solely validation criteria, should be more carefully calculated and discussed. The equifinality issue in the hydrological model is not presented and this is clearly missing, as it does have consequences when assessing flood plains and landslide. Explanation of the models, calibration set up should be provided. It is not clear how the parameter are set up according to whether the catchment properties, literature values or adapted through calibration for hydrological likelihood purposes. Finally the limits of the results, assessing the floodplain and the landslide areas, should be discussed in the appropriate section and compare to literature.

The manuscript does not properly respect the announced structure and some paragraph should be reorganized (see below). Also note that numerous annotations are missing, and makes difficult to understand the scientific assumptions beyond several equation (specially I personally didn't understand the flood plain method).

**- Specific comments**

Please note that the proposed rewriting are suggestions that may clarify my concerns and comments. Please feel free to consider and modify those drafts.

1. Abstract: quite long : the first 8 lines might be removed.

-

2. Introduction: the introduction is too long. The bibliography done is significant but irregular and sometimes out of the scope. I would suggest to follow the plan : i) Flash flood: definition, hazard risk; ii) Catchment and meteorological features controlling the flash floods AND landslides processing; iii) Flash flood in Colombia, the specific Salgar flash flood events; IV) ssue : Flood forecast and modelling with scarce data; V)Objectives of the paper, method, plan.

-

3. Introduction: I would suggest the following modifications:

3.1. from page 2, line 52 to page 4, line 106 : the literature should be more concise and be reduced up to 2 or 3 paragraphs. The plan of the statements declared page 2, line 48 – 52 should give guidelines for the organization of the paragraphs.

3.2. Page 2, line 53 – 55 and page 4, line 108 – 116 : to my point of view, those lines are out of the scope and could be removed, making clearer the introduction.

Lines 53-55 and lines 108-116 have been removed from the introduction.

3.3. Page 125 – 129 : are the figure related to Colombia alone or the Caribbeans and the South of America? As the paragraph starts with a description of Colombia, I suggest to give statistics related to the country itself, in order to avoid confusion.

3.4. Page 5, line 131 – 148 : the full description of the events should be located in the 'Data and area of the Study'. Here a succinct description with argument for showing the interests of that particular study case are expected. As example: 'The paper focuses of two consecutive rainfall storms that took place in May 15th and May 18th 2015 in La Liboriana upstream, a 56 km2 catchment located in the Western range of the Colombian Andes. The resulting flash floods dramatically affected the local population, leading to more than 100 casualties, and to an estimated total cost of 36 000 millions Colombian pesos ($\sim$ 12.5 millions $, considering the 2018 rate) for infrastructure reconstruction

and community supports. In spite of the data scarcity, including discharge measurements, that two successive events provide an interesting study case for assessing the flash flood mechanism processes with contrasted rainfall forcing distribution and soil moisture conditions.'

3.5. Page 5, line 161: At the end of the paragraph, it should be specified which method is chosen here (the third one, I guess).

3.6. Page 6, line 175 – 191: here again the description of the method should be more concise and detail has to given in the third section. I suggest the following modifications: 'The methodology followed in this study makes use of a conceptual modeling framework that includes a hydrological model (Vélez (2001) and Francés et al. (2007)), a shallow land-slide sub-model (Aristizábal et al., 2016), and a hydraulic sub-model (HydroFlash). The hydrological model includes virtual tracers to explore separately the role of runoff and subsurface flow, as well as the relative importance of convective and stratiform precipitation in flash flood generation. A comparison between the results from both sub-models and the observed landslides scars and flooded spots helps to evaluate the overall skill of the proposed methodology.

-

4. Section 2: Study site and data

4.1. To clarify the structure, two subsections might be established: 2.1 = Catchment description / properties – 2.2 = Flash flood event observations

4.2. Figure 2: Please indicate the size resolution of cells used when calculating HAND.

4.3. Pages 8-9, lines 218 – 245 : Maps of the slopes, vegetation cover (even roughly designed), and soil properties would provide a better illustration of the description.

4.4. Figure 3: The information content of this figure is quite repetitive with the figure 2. I suggest to remove it.

4.5. Page 9, lines 235 – 245 : soil information. The lines 236 and 240 are not of interest and could be removed. In contrast, some information about how Osorio (2008) obtained the data could be relevant.

4.6. From page 9, line 246 to page 10, line 255. The description of the data used for validation of the models should be more detailed in this section. Specially we should find the following information : How was determine the maximum flood discharge, the landslides and flood plain areas ? Which data were used ? Which method was used to extracts area contours as showed in the results ? Which uncertainty ?

4.7. Page 9, line 253: 'Assuming flow speeds between 5 and 6 m.s-1' Please give more detail about this assumption: is it according to literature value? Or the ange of speeds were calculated from the section roughness and stream bed slope consideration (as Neppel et al, 2010) ? Is there also an estimation of the flood peak time?

-

5. Section 3: Methodology

5.1. This section in the current form is difficult to follow, as the description jumps from one model to another and finally comes back at the first mentioned one. I would suggest to reorganize this section in order to follow the method firstly announced in the introduction : 3.1 hydrological model description (that have to include hydrological scheme modification and the tracers implementation within the hydrological model); 3.2 landslide model description; 3.3 floodplain model description. Rainfall data processing has to be presented in Section 2.

5.2. The methodology section should strictly provide the method description, and the underlying assumptions made. Any argument to justify the objective of the study should be remove from this section (as example : lines 293 – 302; 316 – 319; 326 – 332; 345 – 347; 375- 382).

5.3. The Figure 5 should be the key figure offering a clear visual description of the

overall method applied in this manuscript. To my understanding of the achieved work, this diagram should rather have three levels (top to down of the diagram): i) two panels showing the data inputs of the hydrological models, i.e. the DEM, the radar-based QPE, and the radar-based QPE processing according to Steiner, 1995); ii) one panel showing the hydrological model (Francés, 2007); iii) three panels showing the 3 results of the overall methods : the discharge simulation with water origins and paths information, the landslides submodel, and finally the flood plain assessment.

5.4. Descriptions of the models : it should useful to clarify through tables, which parameter/information is required, how the parameters were set up (calibration ? literature value, observed data?), and whether they are spatially distributed or uniform.

5.5. Flash flood model (I would say flood plain model) : I personally have some trouble to understand the method applied. There is no reference or explanation of the assumptions behind the equations (10, 11, 12). In addition, several symbols are not defined: $C_{j,s}$ $C_{max}$, and others are detailed ($S_{i,0}$) while not used in the equations, suggesting that an equation is missing in the manuscript.

5.6. Page 16, line 423: About the hydrological runoff modification : why did you proceed to this modification? Is an adaptation of the model to the studied catchment?

-

6. Section 4 : Results

6.1. In the current manuscript's form, the section 4 contains only one subsection, which doesn't make sense. One logical plan considering the objective of the paper should be: 4.1 Validation of the hydrological model; 4.2 Description of the flash flood mechanism processing; 4.3 Assessment of the landslide simulation; 4.4 Assessment of the flood plain simulation (or 4.3 and 4.4 could be also merged).

6.2. The description of the flash flood mechanism processing are really detailed while the other results are summary explained : to be more balance

6.3. fig 12: if I understand well, the figure 12 shows spatial distribution characteristics of the precipitation over the catchment, but it is not a result of the hydrological simulation. This figure 12 should be better located in the section 2.

7. Section 5: the discussion deals only with the description of the hydrological simulation. The discussions on the landslide model and on the flood area assessment are clearly missing.

-

**- Technical comments**

T1. page 6, line 199: section 5

T2. page 7: reword the section 2 'Study site and data'

T3. Figure 2: remove 'Las margaritas village' as it doesn't appear anywhere else in the text.

T4. Page 9, line 246: please write 'carried out for assessing' instead of 'instrumental in obtaining'

T5. Page 9, line 246: specify 'after the second flash event'

T6. Page 10, line 268 : write 'minute' instead of 'min'

T7. Page 10, lines 269 – 274. The sentences were twice written. Please, remove the duplicate.

T8. From page 9 line 246 to page 11 line 274. The two paragraphs give i) the available observations used in the study for assessing the hydrological model and sub-models and ii) the rainfall input data used to force the hydrological model. I would change the order to respect the chronological use of the data. In addition, I would suggest to add a table summarizing the observation information available and used to validate the models.

T9. Figure 5: we can't read any legend of the hydrological model, please uniform the size of the annotation in this figure.

T10. Page 11, line 285: what is Ri?

T11. Page 11, line 304 - 307: this information has to be provided at the end of the manuscript.

T12. Page 15-16: please revise all the symbol definition of the Hydroflash model.

T13. Page 16, line 426: please define A in the vicinity of it first occurrence

T14. Figure 7: the peak discharge interval used for validation should be indicated in this figure.

T15. Figure 9 and 10: These figure 9 and 10 might be merged, presenting on the left side the 50 groups categorization (right panel of the figure 9), and the current figure 10 on the right side. The same key color defining the 50 groups categorization should be used in the map (Fig. 9) and to define somehow the related scale of the fig. 10.

T16. Along all the manuscript, the unity has not to be in italic font. In addition, change the annotation m/s to m.s-1.

**- References:**

Neppel, L., Renard, B., Lang, M., Ayral, P.-A., Coeur, D., Gaume, E., Jacob, N., Payrastre, O., Pobanz, K. & Vinet, F. (2010) Flood frequency analysis using historical data: accounting for random and systematic errors. Hydrol. Sci. J. 55(2), 192–208.

---

## Referee Comment (RC2) · Gaume (Referee) · 10 Dec 2018

The proposed manuscript presents and analysis an interesting and relatively well documented extreme flash flood occurred in May 2015 in Columbia that killed about 100 people an induced severe damages. A large part of the manuscript is devoted to the implementation of several numerical models (a distributed rainfall-runoff model, a sediment transport model and a slope stability model) used to reproduce some observed patterns of the event and to propose some interpretation on the dominant flood generating mechanisms during this specific flood event. The manuscript is suited to

HESS, potentially interesting but deserves some major improvements before it can be published. The major concern lies in the proposed approach consisting in pretending providing explanation on dominant hydrological processes based on a rainfall-runoff model that is essentially calibrated or validated based on a single downstream peak discharge estimate. This is really unreasonable. Distributed rainfall-runoff models may be implemented to reproduce such events and their performances may be evaluated, but it must be clearly explained how the values of their numerous parameters and state variables (especially soil moisture and possible groundwater levels) are fixed. This is partly done in the manuscript, but not sufficiently. The manuscript must clearly define which parameter values are determined a priori, based on which observed data and which parameter values are possibly adjusted to fit the model. Moreover, even if distributed, hydrological models especially when implemented at large scale, can not account for the complexity of rainfall-runoff processes related to small scale variability an preferential flows. The models may reproduce the dynamics of the rainfall-runoff response, but are far from perfect. The interpretations of the authors based on this model implementation exercise are not consistent with the real accuracy of the models. The modelling exercise is interesting but first the authors should try to find additional information to support their analyses (a single downstream discharge value is not sufficient, see detailed comments for suggestions) and remain prudent in their conclusions.

Detailed comments: P1L15: the virtual tracer experiment separates the simulated "runoff" and "subsurface flow " contributions in the model (i.e. fast and delayed contributions), but real-world tracer experiments could provide very different partitioning as illustrated by numerous past geochemical hydrograph separation studies. Simulated processes can not be simply considered as representing effective processes on the considered watershed. This is a much to simplistic point of view on hydrological processes. P4L103: difficulting does not exist. P8L216: I do not know if it is possible to really say that some watershed are geomorphomogically prone to flash floods. At least, several studies (Marchi et al., 2010 ; Smith et al, 2018) do not show clear relations between geomorphological settings and magnitude of extreme peak

discharges. P10L254: the selected velocities are relatively high, especially for average cross-sectional velocities (see Lumbroso et al., 2012). The provided estimates may be a little high. Are their some films that could help reduce the estimation uncertainties and provide some ideas of possible peak velocities. According to figure 15, the flood extent has been mapped, and probably flood marks identified, along a large part of the main stream. Since the second event has been produced mainly in the upstream part of the watershed, it would have been interesting to base the analysis on some other peak discharge estimates along the main stream and its tributaries. The ability of the proposed model to reproduce the spatial distribution of the flood peaks on the watershed could have then been tested. P10L270: the same paragraph is repeated twice. Figure 5 : impossible to read. The legends must be increased. The five compartments described in the text must be clearly identified. P13-17: this part could be placed for most of it in an appendix. Moreover, all the variables used in each equation must be clearly defined, which is not always the case and makes it difficult to follow the explanations. The variable names are changing from one equation to the other as for instance A, A2, A3 between eq. 13-17. If it is the same variable, use the same symbol. P18: The radar rainfall rates must be quality checked. The area is relatively far from the radar and mountainous, two settings that could introduce uncertainties and errors. Are their some available raingauge measurements on the affected watershed or on nearby areas ? How do the radar-based rainfall estimates compare with corresponding raingauge measurements ? The two considered rainfall events are spatially heterogeneous unlike what is stated further in te manuscript. The upper part of the watershed is almost not affected by the first event. It would be essential to distinguish this upper part in the rest of the analysis since the average simulated soil moisture and and runoff components may hide a significant spatial variability. Conclusions drawn on the importance of the first event for the saturation of the soils could be largely nuanced by a more detailed spatial analysis. P18L486: What about the first flood event. Is their any possibility - based on local information - to have an idea of the possible value of the first peak discharge. It would be interesting to know if the simulated discharge - 160

m3/s corresponding almost to full-bank flow according to the estimates on page 10, has been observed or no. This would give one more reference point for the evaluation of the implemented rainfall-runoff model. P19L494: It is essential for the second event, according to the spatial rainfall heterogeneities, to provide some distributed simulation results : what part of the flood volume has been produced on the 15% upstream part of the watershed ? Is the contribution form the intermediate watershed significant ? P21L530: It would be essential to provide some information on the real timing of the floods that could for sure been provided, at least approximately, by eyewitnesses. Discussion on simulated timings, that may be wrong is of little interest. P21L539: the sentence "Event 1 does not trigger a flash flood event" is not supported by the facts and probably excessive. It did certainly not produce significant overflows and damages, but may have produced a significant flood events (an estimated discharge for this first event is clearly missing in the manuscript). If so, according to the duration of the event, the flood can also be considered as a flash flood. Part 4: the simulation part, and the interpretation of the results is not uninteresting. But the spatial variability must be shown and commented as suggested before and a clear difference must be made between these simulation results and the real-world. What is presented is the outcome of a numerical model, with the selected parameter and initial state values: some other choices could have provided equally good results if compared to the only available estimated peak discharge but very different flow separation between the various simulated components. Again, the choice of the various parameter values must be clearly justified. Some sensitivity analyses of the results and partitions to these values would also be welcome to strengthened the analyses: are the conclusions always the same if the values are varied over a reasonable range ? I ave doubts. P22L562: Why should the soils be wet upstream since this area has not strongly been affected by the first rainfall event ? P24L590: The spatial agreement is not a real surprise since the landslide model has been calibrated and according to the spatial distribution of rainfall.. P24L600: Can the concentration of landslides in the first part of the rainfall event be confirmed in any way (eyewitnesses). By the way, this is a surprising result. In general,

the landslide density has a general tendency to be related to the rainfall amounts and the progressive increase of soil saturation in mountainous areas (sign that infiltration dominates the hydrological processes throughout intense storm events). The obtained results would imply that even a very short-lived intense rainfall would have produced landslides in the area. I have many doubts that this is realistic. P25L619: The notions of order are not presented (probably not Strahler order according to the value). Please explain. Figure 15: this figure shows that the post-event survey database is much richer that what is used and presented in the manuscript. Intermediate values of discharges could have been estimated for instance. The comparison between simulated flow depths and observed flood extents is far from perfect. Can this be attributed to the Digital terrain model ? A critical analysis of the digital terrain model could be provided in the manuscript (comparison between observed and extracted cross sections for example). P27L647: there is no information about flood 1, the authors can not speak about evidence of remarkable behavioral difference. We do not even know if a flood of significant magnitude occurred... The rainfall can not be described as spatially quasi-homogeneous. What do the author mean with "return flow" and "20 groups" L654: the authors can not state that the second convective core results mainly in surface runoff. First, the most affected area has hardly been saturated by the first event (spatially detailed result will probably show it) and in anyway, that is a simulation result and not necessarily reality. P28: The conclusions have to be deeply rewritten to introduce nuance and prudence according to all previous comments.

Lumbroso D., Gaume, E,. 2012. Reducing the uncertainty in indirect estimates of extreme flash flood discharges, Journal of Hydrology, doi:10.1016/j.jhydrol.2011.08.048 Marchi, L., Borga, M., Preciso, E., and E. Gaume, 2010: Characterization of selected extreme flash floods in Europe and implications for flood risk management. Journal of Hydrology, 394 (1-2), 118-133. doi:10.1016/j.jhydrol.2010.07.017. Smith J. et al., 2018. Strange Floods: The Upper Tail of Flood Peaks in the United States. WRR, doi:10.1029/2018WR022539

---

## Author Comment (AC1) · 12 Jan 2019

article [utf8]inputenc [top=1in, bottom=1.25in, left=1.25in, right=1.25in]geometry

natbib graphicx xcolor tabularx graphicx adjustbox

[Figure]

**Response to Reviewer 1**

January 12, 2019

**Manuscript title:** Reconstructing the Salgar 2015 Flash Flood Using Radar Retrievals and a Conceptual Modeling Framework: A Basis for a Better Flood Generating Mechanisms Discrimination

**Authors:** Nicolás Velásquez, Carlos D. Hoyos, Jaime I. Vélez, and Esneider Zapata

We sincerely thank the anonymous reviewer #1 and Dr. Eric Gaume for their careful and thoughtful reviews. We have taken their considerations into account and have responded to their concerns both in the paragraphs below and within the manuscript. We feel the current manuscript is indeed better thanks to the reviewer comments. Below, the reviewer comments are in black and our comments are in "blue".

**Response to Anonymous Reviewer #1 General Comments**

We really appreciate the detailed work by anonymous reviewer #1. We now recognize the point raised by reviewer #1 about the potential lack of clarity of the paper having two main objectives. We decided to focus on the process understanding goal and we rewrote several paragraphs of the paper trying to address the clarity issue. More weight is thus given to the process understanding goal, and the forecast issue is only mentioned directly in the discussion section; we believe mentioning the forecast issue makes the paper stronger from an applied point of view.

Also, in the current version of the manuscript, there is a detailed explanation regarding the model set-up, parameter selection, and model calibration, a point raised by reviewer #1. As described in the current version of the manuscript, some parameters are indeed from literature, and others from model calibration from other basins in the region with similar properties, and from sensitivity analysis.

We fully agree with reviewer #1 in that the conclusions derived from modeling results have limitations and those were not fully acknowledged nor discussed properly in the original manuscript. This issue was also raised by reviewer #2 (Dr. Eric Gaume). The current manuscript addresses the potential uncertainty of the modeling results in the discussion section, including a comparison with existing literature. Due to the data scarcity in the region there is an inherent impossibility to validate the model. To address this issue we included a sensitivity analysis showing the main conclusions regarding process understanding are the same for different model calibrations using parameter values within physically plausible ranges. Reviewer 1 also points to the fact that the equifinality issue is not discussed in the original manuscript; we now address this issue in the results and discussion section based on the sensitivity analysis, also highlighting the fact that the original model calibration not only considered peak discharge but also the spatial distribution of the flood plains and landslides.

Several changes were made to the manuscript to improve the description of the meth-
ods and to correct problems with the structure of the document.

**Response to Specific Comments**

1 **Abstract:** quite long : the first 8 lines might be removed

The abstract has been modified accordingly. It was also shortened and partly rewritten to focus on the process understanding goal.

2 **Introduction:** the introduction is too long. The bibliography done is significant but irregular and sometimes out of the scope. I would suggest to follow the plan : i) Flash flood: definition, hazard risk; ii) Catchment and meteorological features controlling the flash floods AND landslides processing; iii) Flash flood in Colombia, the specific Salgar flash flood events; IV) ssue : Flood forecast and modelling with scarce data; V)Objectives of the paper, method, plan.

The introduction has been shortened considerably following reviewer 1 general plan and the detailed comments in the 3. We really appreciate reviewer 1 suggestions regarding the Introduction.

3 **Introduction:** I would suggest the following modifications:

3.1 from page 2, line 52 to page 4, line 106 : the literature should be more concise and be reduced up to 2 or 3 paragraphs. The plan of the statements declared page 2, line 48 – 52 should give guidelines for the organization of the paragraphs.
The first paragraph has also been shortened, and the lines 52 to 106 were reorganized and shortened.

3.2 Page 2, line 53 – 55 and page 4, line 108 – 116 : to my point of view, those lines are out of the scope and could be removed, making clearer the introduction.

Lines 53-55 and lines 108-116 have been removed from the introduction.

3.3 Page 125 – 129 : are the figure related to Colombia alone or the Caribbeans and the South of America? As the paragraph starts with a description of Colombia, I suggest to give statistics related to the country itself, in order to avoid confusion.

The statistics are only for Colombia. This issue has been clarified in the text.

3.4 Page 5, line 131 – 148 : the full description of the events should be located in the 'Data and area of the Study'. Here a succinct description with argument for showing the interests of that particular study case are expected. As example: 'The paper focuses of two consecutive rainfall storms that took place in May 15th and May 18th 2015 in La Liboriana upstream, a 56 km2 catchment located in the Western range of the Colombian Andes. The resulting flash floods dramatically affected the local population, leading to more than 100 casualties, and to an estimated total cost of 36 000 millions Colombian pesos (âĹij 12.5 millions $, considering the 2018 rate) for infrastructure reconstruction and community supports. In spite of the data scarcity, including discharge measurements, that two successive events provide an interesting study case for assessing the flash flood mechanism processes with contrasted rainfall forcing distribution and soil moisture conditions.'

We modified the paragraph following closely referee #1 comments.

3.5 Page 5, line 161: At the end of the paragraph, it should be specified which method is chosen here (the third one, I guess).

Yes, we followed the third approach. This is now explicitly mentioned in the manuscript. We also deleted most of lines 163-174 for clarity.

3.6 Page 6, line 175 – 191: here again the description of the method should be more concise and detail has to given in the third section. I suggest the following modifications: The methodology followed in this study makes use of a conceptual modeling framework that includes a hydrological model (Vélez

(2001) and Francés et al. (2007)), a shallow land-slide sub-model (Aristizábal et al., 2016), and a hydraulic sub-model (HydroFlash). The hydrological model includes virtual tracers to explore separately the role of runoff and subsurface flow, as well as the relative importance of convective and stratiform precipitation in flash flood generation. A comparison between the results from both sub-models and the observed landslides scars and flooded spots helps to evaluate the overall skill of the proposed methodology.

In this case we also modified the paragraph following closely referee #1 comments.

4 **Section 2:** Study site and data

4.1 To clarify the structure, two subsections might be established: 2.1 = Catchment description / properties – 2.2 = Flash flood event observations.

We modified the document to include the subsections suggested by referee # 1.

4.2 Figure 2: Please indicate the size resolution of cells used when calculating HAND.

HAND values where estimated using the same resolution of the DEM (12.7 m). We included this information in the caption of the Figure 2.

4.3 Pages 8-9, lines 218 – 245 : Maps of the slopes, vegetation cover (even roughly designed), and soil properties would provide a better illustration of the description.

We have added the slope map as part of Figure 2. Soil properties and vegetation cover maps are part of a national cartography with a scale of

1:400.000; at this scale there is almost no variability in the watershed. Nevertheless, we added Figure 1 showing land use in different regions of the watershed from a 2012 satellite image. According to zoom 1, the vegetation is denser in the upper region of the basin, with the presence of grass. Downstream (zoom 2), it is evident the presence of crops among forest and grass. Near the middle of the basin (zoom 3), the presence of crops is more notorious and human settlements and roads start to appear. In zoom 4, the first affected urban area from upstream to downstream, it is also possible to see a marked presence of crops and some forest patches. Finally, zoom 5 shows the main urban area of Salgar surrounded by crops, grass and an important loss of forest coverage. We modified Lines 229-234 in the original manuscript, accordingly.

4.4 Figure 3: The information content of this figure is quite repetitive with the figure 2. I suggest to remove it.

We agree with referee # 1 and we have removed the figure. Slope information was added to new Figure 2 in the new manuscript.

4.5 Page 9, lines 235 – 245 : soil information. The lines 236 and 240 are not of interest and could be removed. In contrast, some information about how Osorio (2008) obtained the data could be relevant.

Lines 236 and 240 were removed following referee # 1 suggestions. Also, we included some information about how (?) obtained the information.

4.6 From page 9, line 246 to page 10, line 255. The description of the data used for validation of the models should be more detailed in this section. Specially we should find the following information : How was determine the maximum flood discharge, the landslides and flood plain areas ? Which data were used ? Which method was used to extracts area contours as

[Figure]

**Fig. 1.** Land use in different regions of La Liboriana watershed from a 2012 satellite image.

showed in the results ? Which uncertainty ?

We conducted a field campaign a few days after the flash event to assess
cross-section geometry along the main channel in different sites, including
at the outlet of the basin. During the campaign, we measured sectional

distances and surface water speeds at different points of the streamflow. We also identified traditional post-event terrain, land-cover, vegetation and infrastructure markers to assess the high-water mark associated with the peak of the flash flood. Figure 4 presents the selected cross-section used for the estimation of the maximum discharge during the flash flood given its geometrical and hydraulic regularity. The section has a rectangular shape, 4.6 $m$ wide and a height of 5 $m$ for a total area of about 23 $m^2$. A visual inspection of the flooded house around this section, located 4-5 $m$ away from the channel, reveals the presence of mud marks on the walls with heights varying between 0.5 and 1.2 $m$ (see Figure 4). The area of the section plus the flooded area during the event was estimated to be approximately 37 $m^2$. During the campaign we measured surface speeds in the channel oscillating between 2 and 3 $ms^{-1}$. In instrumented basins in the region, with similar characteristics in terms of area and slopes, we have recorded peak flow surface water speeds oscillating between 5 and 7 $ms^{-1}$. By assuming an area of 37 $m^2$ and the mentioned surface speeds, we estimate that the observed flash flood peak flow may have been between 185 and 222 $m^3s^{-1}$. The timing of the peak flow is also important information. Local authorities reported that the flash flood impacted the urban area around 02:00 A.M.

There is also relevant aerial information before and after the occurrence of the event. During 2012, the Department of Antioquia conducted a detailed aerial survey of the Salgar municipality, and few days after the event Google displayed publically high detailed satellite images of the same region. The contrast between both products provides information about flooded areas and landslide locations. Field campaign estimates and aerial imagery are central to validate the results obtained from the proposed models.

5 **Section 3:** Methodology

5.1 This section in the current form is difficult to follow, as the description jumps from one model to another and finally comes back at the first mentioned one. I would suggest to reorganize this section in order to follow the method firstly announced in the introduction : 3.1 hydrological model description (that have to include hydrological scheme modification and the tracers implementation within the hydrological model); 3.2 landslide model description; 3.3 flood-plain model description. Rainfall data processing has to be presented in Section 2.

We rearranged the section according to reviewer 1 suggestions.

5.2 The methodology section should strictly provide the method description, and the underlying assumptions made. Any argument to justify the objective of the study should be remove from this section (as example : lines 293 – 302; 316 – 319; 326 – 332; 345 – 347; 375- 382).

We followed the suggestion of reviewer and, after rearranging and rewriting some parts of the methodology section, the lines mentioned were removed.

5.3 The Figure 5 should be the key figure offering a clear visual description of the overall method applied in this manuscript. To my understanding of the achieved work, this diagram should rather have three levels (top to down of the diagram): i) two panels showing the data inputs of the hydrological models, i.e. the DEM, the radar-based QPE, and the radar-based QPE processing according to Steiner, 1995); ii) one panel showing the hydrological model (Francés, 2007); iii) three panels showing the 3 results of the overall methods : the discharge simulation with water origins and paths information, the landslides submodel, and finally the flood plain assessment.

We changed Figure 5 according to the reviewer suggestions and we think it offers now a clearer picture of the work. See Figure 2 in this document.

```
figure-3.png
```

**Fig. 2.** New methodology figure.

5.4 Descriptions of the models : it should useful to clarify through tables, which param- eter/information is required, how the parameters were set up (calibration ? literature value, observed data?), and whether they are spatially distributed or uniform.

The hydrologic model requires a total of ten parameters. Table 1 includes

[Figure]

| Parameter Name | Symbol | Scalar Parameter | Mean Value | Spatial distribution |
|---|---|---|---|---|
| Capilary storge | Hu [mm] | 1 | 39 | In function of the slope |
| Gravitational storage | Hg [mm] | 1 | 34 | As a function of the slope |
| Evaporation rate | Etr [mm/s] | 0.1 | 0.01 | As a function of the DEM |
| Infiltration rate | ks [mm/s] | 2.7 | 0.0012 | Lumped |
| Percolation rate | kp [mm/s] | 0.8 | 0.00012 | Lumped |
| System losess | Kf [mm/s] | 0 | 0.0 | Lumped |
| Surface speed | vr [m/s] | 0.5 | 6.4 | As a function of the slope and storage |
| Subsurface speed | vs [m/s] | 1 | 7.1 | As a function of the slope, Hg and storage |
| Subterranean speed | vb [m/s] | 0.5 | 0.000095 | Lumped |
| Channel speed | vc [m/s] | 1 | 0.95 | As a function of the slope, acumulated area, and storage |

**Table 1.** Hydrologic model parameters.

all the parameters used in the model. Due to the lack of detailed information in the region, parameters such as the infiltration and percolation rates are assumed as constant in all the basin. There are also parameters that vary as a function of geomorphological characteristics of the basin such as the elevation and slope. According to (?), during the calibration process, each physical parameter is scaled by a constant value. Table 1 includes the mean value for all the parameters used in the model and the scalar value parameter adjusted during the calibration.

5.5 Flash flood model (I would say flood plain model) : I personally have some trouble to understand the method applied. There is no reference or explanation of the assumptions behind the equations (10, 11, 12). In addition, several symbols are not defined: Cj,s Cmax, and others are detailed (Si,0) while not used in the equations, suggesting that an equation is missing in the manuscript.

We re-wrote the methodology for the flash flood model and made sure all the terms in the equations are properly described..

5.6 Page 16, line 423: About the hydrological runoff modification : why did you proceed to this modification? Is an adaptation of the model to the studied catchment?

Before our implementation, there were two major versions of the model. There is the SHIA (Simulación Hidrológia Abierta) version presented by (**?**) in his PhD thesis dissertation, and the TETIS version presented by **?**. Compared to TETIS, SHIA allows the modeler to use linear and no-lineal approximations for the estimation of the horizontal speeds. From the literature (**??**) there is evidence that non-linear approximations increase modeling performance at detailed spatiotemporal scales. For this reason, we implemented the SHIA approximation proposed by (**?**).

**6 **Section 4 :** Results**

6.1 In the current manuscript's form, the section 4 contains only one subsection, which doesn't make sense. One logical plan considering the objective of the paper should be: 4.1 Validation of the hydrological model; 4.2 Description of the flash flood mechanism processing; 4.3 Assessment of the landslide simulation; 4.4 Assessment of the flood plain simulation (or 4.3 and 4.4 could be also merged).

We subdivided the results section into 3 sub-sections: 4.1. Hydrologic model validation and sensitivity analysis, 4.2. Flash flood processes, and 4.3. Landslide and flood simulations

6.2 The description of the flash flood mechanism processing are really detailed while the other results are summary explained : to be more balance

We have balanced the different subsections in the results section. We have also added results from a sensitivity analysis of the hydrologic simulation.

6.3 fig 12: if I understand well, the figure 12 shows spatial distribution characteristics of the precipitation over the catchment, but it is not a result of

the hydrological simulation. This figure 12 should be better located in the section 2.

Former Figure 7, only dealing with rainfall features, was moved to section 2. Figure 12 do show the spatial distribution of the characteristics of the precipitation over the catchment. The Figure is not a result of the simulation, nevertheless, along with Figure 13 explains relations between observed and simulated processes and we decided to keep it in the results section.

7 **Section 5:** the discussion deals only with the description of the hydrological simulation. The discussions on the landslide model and on the flood area assessment are clearly missing.

In the interest of the overall length, and as announced in the last paragraph of the introduction, we decided to maintain the discussion section focused on the hydrological simulation results, as well as the processes generating the Salgar flash flood, and the importance of the spatio-temporal variability of rainfall.

**Response to Technical Comments**

T1 page 6, line 199: section 5

Section corrected.

T2 page 7: reword the section 2 'Study site and data'

The title was modified following the comment.

T3 Figure 2: remove 'Las margaritas village' as it doesn't appear anywhere else in the text.

The reference to 'Las Margaritas' was removed.

T4  Page 9, line 246: please write 'carried out for assessing' instead of 'instrumental in obtaining'

The section was rewritten and the sentence was modified.

T5  Page 9, line 246: specify 'after the second flash event'

The line was edited.

T6  Page 10, line 268 : write 'minute' instead of 'min'

The line was changed.

T7  Page 10, lines 269 – 274. The sentences were twice written. Please, remove the duplicate.

The paragraph was corrected.

T8  From page 9 line 246 to page 11 line 274. The two paragraphs give i) the available observations used in the study for assessing the hydrological model and sub-models and ii) the rainfall input data used to force the hydrological model. I would change the order to respect the chronological use of the data. In addition, I would suggest to add a table summarizing the observation information available and used to validate the models.

The section was modified and restructured. Also, a summary table was included to provide a quick reference for the reader.

T9  Figure 5: we can't read any legend of the hydrological model, please uniform the size of the annotation in this figure.

The Figure was edited; please see Figure 2.

T10  Page 11, line 285: what is Ri?

| Item | Description | Period | Usage |
|------|-------------|--------|-------|
| Radar data | QPE rainfall estimations | 2015-05-17 to 2015-05-18 | Hydrological model runs. Rainfall characterization and event analysis. |
| Field campaign | Maximum streamflow estimation trough visual inspection | 2015-05-20 | Hydrological model comparison for indirect validation. |
| Satellite imagery | Real visual compositions of Digital Globe CNESimagery | 2015-05 after the event | Flash flood model validation, shallow land-slides model validation, and comparison with pre-event conditions. |
| Aerial photos | Aerial photos taken by the government of Antioquia during 2012. | 2012 | Pre-event conditions comparison. |
| Soils description | Physical description of the soils of the region done by (?) | 2008 | Hydrological model setup. |

**Table 2.** Summary of the data used for the model setup.

$R_i$ is the vertical flow through tanks. The description of this variable is now included in the document and in the caption of Figure 4.

T11  Page 11, line 304 - 307: this information has to be provided at the end of the manuscript.

The information was moved to the acknowledgements section.

T12  Page 15-16: please revise all the symbol definition of the Hydroflash model.

The model description was revised and edited.

T13  Page 16, line 426: please define A in the vicinity of it first occurrence

The paragraph was edited in order to give a proper representation of variable A.

Horizontal flow equations could be either linear or potential, as shown in equation (1). In the modified hydrological model, $\beta$ and $\alpha$ are estimated by the user and

then set into the model. From equations (1) to (3) $A_i(t)$ corresponds to the sectional area of each tank, $A_i(t)$ vary in function of the tank storage $S_i(t)$ (equation (4).

T14 Figure 7: the peak discharge interval used for validation should be indicated in this figure.

In the new version, the discharge interval use for validation is marked now with a red square.

T15 Figure 9 and 10: These figure 9 and 10 might be merged, presenting on the left side the 50 groups categorization (right panel of the figure 9), and the current figure 10 on the right side. The same key color defining the 50 groups categorization should be used in the map (Fig. 9) and to define somehow the related scale of the fig. 10.

In the Figure 3 we present how Figure 9 and 10 were merged in the revised document.

T16 Along all the manuscript, the unity has not to be in italic font. In addition, change the annotation m/s to m.s-1.

The units were changed.

```
figure-1.png
```

**Fig. 3.** Edited figure.

---

## Author Comment (AC2) · 12 Jan 2019

article [utf8]inputenc [top=1in, bottom=1.25in, left=1.25in, right=1.25in]geometry

natbib graphicx xcolor tabularx graphicx adjustbox

[Figure]

**Response to Reviewer 2**

January 12, 2019

**Manuscript title:** Reconstructing the Salgar 2015 Flash Flood Using Radar Retrievals and a Conceptual Modeling Framework: A Basis for a Better Flood Generating Mechanisms Discrimination

**Authors:** Nicolás Velásquez, Carlos D. Hoyos, Jaime I. Vélez, and Esneider Zapata

We sincerely thank the anonymous reviewer #1 and Dr. Eric Gaume for their careful and thoughtful reviews. We have taken their considerations into account and have responded to their concerns both in the paragraphs below and within the manuscript. We feel the current manuscript is indeed better thanks to the reviewer comments. Below, the reviewer comments are in black and our comments are in "blue".

**Anonymous reviewer 2**

After taking into account both reviewer comments, we agree with Dr. Eric Gaume the manuscript needed improvements before final publication. The manuscript indeed focuses on the model reconstruction of the Salgar 2015 flood, focusing on trying to understand the main processes leading to the flash flood generation. The data availability in the region does not allow to conduct an in data analysis study. We decided to include a sensitivity analysis to add robustness to the general results. In the current version of the manuscript, there is a detailed explanation regarding the model set-up, parameter selection, and model calibration, a point raised by both reviewers. As described in the current version of the manuscript, some parameters are indeed from literature, and others from model calibration from other basins in the region with similar properties, and from sensitivity analysis. We also agree hydrological models can not account for the complexity of rainfall-runoff at the small scale; we recognize this fact in the discussion, however, we note that the rainfall data and the simulation results suggest the leading processes triggering the flash flood do not lie in the smallest scales. The current version of the discussion and the conclusions directly state the potential weaknesses of the study.

We fully agree with both reviewers the conclusions derived from modeling results have limitations and those were not acknowledged nor discussed properly in the original manuscript. The current manuscript addresses the potential uncertainty of the modeling results in the discussion section, including a comparison with existing literature. Due to the data scarcity in the region of there is an inherent lack of validation of the model that is not possible to resolve directly. To address this issue we included a sensitivity analysis showing the main conclusions regarding process understanding are the same for different model calibrations using parameter values within physically plausible ranges.

Several changes were made to the manuscript to improve the description of the
methods and to correct problems with the structure of the document.

**Detailed comments**

P1L15 the virtual tracer experiment separates the simulated "runoff" and "subsurface flow" contributions in the model (i.e. fast and delayed contributions), but real-world tracer experiments could provide very different partitioning as illustrated by numerous past geochemical hydrograph separation studies. Simulated processes can not be simply considered as representing effective processes on the considered watershed. This is a much to simplistic point of view on hydrological processes.

We agree with Dr. Gaume regarding the fact that real-world tracer experiments and multiple separation techniques show evidence of a lack of consensus (?). We also know that there are limitations related to the model flow partitioning. Despite those limitations, there exist successful modeling flow partitioning cases (??). Furthermore, we do not intend to represent all the detailed small-scale preferential pathways of water, but rather the first-order approximation of runoff vs. subsurface flow. For this, we use the model results to obtain a conceptual idea about the general processes. We mentioned this explicitly in the conclusions.

P4L103 : difficulting does not exist.

The text was modified following the reviwer comments.

P8L216 I do not know if it is possible to really say that some watershed are geomorphomogically prone to flash floods. At least, several studies (Marchi et al., 2010; Smith et al, 2018) do not show clear relations between geomorphological settings and magnitude of extreme peak.

We agree that not all the geomorphological properties of a watershed are linked to the occurrence of flash floods, and geomorphology itself does not imply flash floods. But literature and experience suggest that some properties do play a role in the triggering of flash floods (**??**). In several flash-floods cases in mountainous regions, the slope and the area are mentioned as important (**???**). Different authors suggest that the slope of the channels (**?**) influence the travel speed and the transport capacity of channels. Also, the slope of the hills is related to the sediment and debris production during the event (**???**). We have added some of these references to the comment in the manuscript.

P10L254 : the selected velocities are relatively high, especially for average cross-sectional velocities (see Lumbroso et al., 2012). The provided estimates may be a little high. Are their some films that could help reduce the estimation uncertainties and provide some ideas of possible peak velocities. According to figure 15, the flood extent has been mapped, and probably flood marks identified, along a large part of the main stream. Since the second event has been produced mainly in the upstream part of the watershed, it would have been interesting to base the analysis on some other peak discharge estimates along the main stream and its tributaries. The ability of the proposed model to reproduce the spatial distribution of the flood peaks on the watershed could have then been tested.

**About speeds:** In local gauged basins with similar characteristics we have recorded peak flow speeds oscillating between 5 and 7 $m/s$ during cases still not considered as flash floods (Figure 1). We have included this comment in the manuscript. By assuming an area of 37 $m^2$ and the described peak speeds, we estimate that the observed flash flood peak flow may vary between 185 and 222 $m^3s^{-1}$.

**About peak discharge in the main stream and tributaries:** The hydraulic model works along with the hydrologic model. Because of this, we do have results for all the tributaries and for both events. In the tributaries, the model shows

[Figure]

**Fig. 1.** Speeds recorded at the Santa Rita Basin located in the Aburra Valley. This watershed has an area of $48km^2$ and a mean slope of $45\%$.

almost no flooding, as observed. And during event 1 the model shows almost no flooded cells.

**About the possible test for the model:** The main purpose of the test-case in the manuscript was to give a first idea of a low-cost hydraulic-hydrological model and its capability to work in real time with limited data. We do not have other cases in the region to test the model. Nevertheless, the maximum flooded area coincide with the peak flow, and both of them coincide with the reported hour of the disaster.

P10L270 the same paragraph is repeated twice. Figure 5 : impossible to read. The legends must be increased. The five compartments described in the text must be clearly identified.

Figure 5 has been modified and corrected following suggestions from both reviewers.

P13-17 this part could be placed for most of it in an appendix. Moreover, all the variables used in each equation must be clearly defined, which is not always the case and makes it difficult to follow the explanations. The variable names are changing from one equation to the other as for instance A, A2, A3 between eq. 13-17. If it is the same variable, use the same symbol.

We decided to leave the mentioned text as part of the main document, but we have restructured the section and subsections, and we have made the corrections following suggestions from both reviewers.

P18 The radar rainfall rates must be quality checked. The area is relatively far from the radar and mountainous, two settings that could introduce uncertainties and errors. Are their some available raingauge measurements on the affected watershed or on nearby areas ? How do the radar-based rainfall estimates compare with corresponding raingauge measurements ? The two considered rainfall events are spatially heterogeneous unlike what is stated further in te manuscript. The upper part of the watershed is almost not affected by the first event. It would be essential to distinguish this upper part in the rest of the analysis since the average simulated soil moisture and and runoff components may hide a significant spatial variability. Conclusions drawn on the importance of the first event for the saturation of the soils could be largely nuanced by a more detailed spatial analysis

**About rainfall estimates** The radar QPE methodology was developed by Sepulveda and Hoyos (2019) using rainfall gauges and disdrometers within the radar

domain. Their results indicate that the rainfall estimation works well within a ra-
dius of 120km (La Liboriana is located at 90km). Despite the distance between
the radar and the basin, and the mountains between them, there are no blind
spots in this region for the radar. Figure 2 shows a comparison between the radar
QPE and two rain gauges stations installed 3 days after the event The correlation
among the hourly precipitation records is 0.65.

[Figure]

**Fig. 2.** Hourly rain gauge-radar QPE comparison using two rain gauges installed 3 days after
the flash flood event.

**About the spatial differences in rainfall** We included former Figures 7 and 10
(Figures 5 and 9 in the revised version) in order to asses the spatial variations of

the variables involved. In these figures, we evaluate the spatiotemporal variations of rainfall and capillary and gravitational storages, and also the amount of saturated and total runoff production. As shown in the figures, there is rain in almost all the basin during Event 1 (about 21 mm in the upper part of the basin), which increases the overall soil moisture including the upper part of the basin. Due to the rainfall characteristics, there is almost no runoff, which coincides with the fact that there are no reports about the occurrence of flash floods during Event 1.

P18L486 What about the first flood event. Is their any possibility - based on local information - to have an idea of the possible value of the first peak discharge. It would be interesting to know if the simulated discharge - 160 $m^3/s$ corresponding almost to full-bank flow according to the estimates on page 10, has been observed or no. This would give one more reference point for the evaluation of the implemented rainfall-runoff model.

Unfortunately there is no stage information available from Event 1. We looked for videos from the community but there is only material available for Event 2. During Event 1 there are no damage reports, but authorities and locals report high fluctuations in the streamflow. This is likely due to the fact that there was no flash flooding during Event 1.

P19L494 It is essential for the second event, according to the spatial rainfall heterogeneities, to provide some distributed simulation results : what part of the flood volume has been produced on the 15% upstream part of the watershed ? Is the contribution form the intermediate watershed significant ?

Considering the reviewer comments, we decided to add an additional Figure for Event 2. Figure 3 shows the temporal evolution of discharge during Event 2 in different locations along the watershed's main channel. The upper location corresponds to 15% of the area of the basin, and the other downstream locations to 52%, 76%, and 100% of the watershed, respectively. In terms of volume, 73

Mm$^3$ of the total 144 Mm$^3$ simulated at the outlet of the basin are generated on the 15% upstream part of the watershed, corresponding to about half of the total mass. In terms of peak flow, due to the slope and velocity changes, the simulated discharge at the 15% upstream part of the watershed corresponds to 61% of the peak discharge at the outlet of the basin.

P21L530 It would be essential to provide some information on the real timing of the floods that could for sure been provided, at least approximately, by eyewitnesses. Discussion on simulated timings, that may be wrong is of little interest.

According to reports from authorities, the peak streamflow reaches the urban perimeter after 2:10 a.m. on May 18th. Some reports state that the peak flow in the most affected community occurred around 2:40 a.m. (https://es.wikipedia. org/wiki/Avalancha_de_Salgar_de_2015). According to our model simulations, the peak flow occurred around 2:20 a.m., which is very accurate considering all the data limitations. The weather radar shows that the rainfall event started at 11:00 p.m. on May 17th.

P21L539 the sentence "Event 1 does not trigger a flash flood event" is not supported by the facts and probably excessive. It did certainly not produce significant overflows and damages, but may have produced a significant flood events (an estimated discharge for this first event is clearly missing in the manuscript). If so, according to the duration of the event, the flood can also be considered as a flash flood.

Given our close contact and interviews with local authorities and the community during the field campaign we conducted to the region immediately after the May 18th flooding, there were no floods reported associated to the event of May 17th.

Part 4 the simulation part, and the interpretation of the results is not uninteresting. But the spatial variability must be shown and commented as suggested before and a clear difference must be made between these simulation results and the real-world. What is presented is the outcome of a numerical model, with the selected parameter and initial state values: some other choices could have provided equally good results if compared to the only available estimated peak discharge but very different flow separation between the various simulated components. Again, the choice of the various parameter values must be clearly justified. Some sensitivity analyses of the results and partitions to these values would also be welcome to strengthened the analyses: are the conclusions always the same if the values are varied over a reasonable range ? I have doubts.

We have restructured some parts of the manuscript, and we have added results from a sensitivity assessment and additional spatio-temporal analysis of Event 2. Taking into account comments made by both reviewers, we added a summary table with the model parameters and an explanation of their origin. Figure 4 shows the results of a sensitivity analysis of the hydrological simulation during the second rainfall event, varying the infiltration rate, and the surface and subsurface speed parameters. The aim of the sensitivity analysis is to evaluate the robustness of the overall results, considering the fact that the quality of some of the watershed information is limited. The overall simulation sensitivity results show the main results described in the previous paragraphs are, in fact, robust to almost all changes in the mentioned parameters, with surface runoff associated with convective rainfall controlling the magnitude of the peak discharge during the Event 2. Changes in the infiltration rate (left panel in Figure 4) result in peak flow changes with a magnitude less than 7%, and changes in the subsurface velocity parameter (right panel in Figure 4) lead to peak flow changes with a magnitude less than 20% the original simulation. The model highest sensitivity, and hence the largest uncertainty source, appears to be related to the surface speed parameter (middle panel in Figure 4), particularly in the low-end values. Although some of the surface speed values used in the analysis are unrealistically low, it is noteworthy to report that these values lead to the attenuation of the hydrograph and the reduction of the peak flow.

P22L562 Why should the soils be wet upstream since this area has not strongly been affected by the first rainfall event?

As mentioned before, and as shown in Figure 5 in the revised manuscript, there is rain in almost all the basin during Event 1, about 21 mm in the upper part of the basin. Also, from Figure 10 in the revised version, we can see variations in the capillary and gravitational storages associated with Event 1 in the higher numbered groups. The capillary storage remains high in almost all the basin until the start of the second event. According to the conceptualization of the model, the gravitational storage and surface runoff start to interact when the capillary storage is full. In this case, this situation is set up by Event 1. We also made some evaluations for Event 2 using dry initial states, with no flooding in the results (Figure 5).

P24L590 The spatial agreement is not a real surprise since the landslide model has been calibrated and according to the spatial distribution of rainfall.

The reviewer is right in that it might not be a surprise, but it is important to clarify that there was no spatial calibration in order to obtain the right location of the landslides. The calibration only includes the change of the soil depth using a single scalar, constant for the entire basin, in order to maximize the number matching observed and simulated slides. In other words, there is just one single basin-wide parameter modified, and not an independent modification of the parameter for every pixel in order to obtain the right distribution. This is important because in that sense, it serves to check the capability of the model to estimate risk areas only considering topography and rainfall data.

P24L600 Can the concentration of landslides in the first part of the rainfall event be confirmed in any way (eyewitnesses). By the way, this is a surprising result. In general, the landslide density has a general tendency to be related to the rainfall amounts and the progressive increase of soil saturation in mountainous areas

(sign that infiltration dominates the hydrological processes throughout intense storm events). The obtained results would imply that even a very short-lived intense rainfall would have produced landslides in the area. I have many doubts that this is realistic.

Unfortunately, there is no eyewitnesses information available to confirm the landslides in the first part of Event 2. The landslide model determines that a cell is unstable when the perched water table is larger than the critical water table. In the conceptualization of the model, this level is represented by the gravitational storage, which is filled by infiltration. Due to high infiltration rates, our results suggest that the gravitational storage was filled during the start of the rainfall event which eventually triggered the landslides. This type of process is common in the tropics (**??**).

P25L619 The notions of order are not presented (probably not Strahler order according to the value). Please explain.

The colors of the streams in Figure 16 correspond to the Strahler order of the network. The simulated stream network was delineated by setting a threshold area (0.1 km$^2$) in order to get a match with the observed network in the region.

Figure 15 This figure shows that the post-event survey database is much richer that what is used and presented in the manuscript. Intermediate values of discharges could have been estimated for instance. The comparison between simulated flow depths and observed flood extents is far from perfect. Can this be attributed to the Digital terrain model ? A critical analysis of the digital terrain model could be provided in the manuscript (comparison between observed and extracted cross sections for example)

**About the post-event survey**: The observed flood spots shown in blue polygons in former Figure 15 correspond to areas delineated from satellite imagery made available by Google days after the flash flood and not to the post-event field trip.

Most of these affected areas were not accessible after the event so we could not obtained reliable stage information in other locations along the main channel other than at the outlet of the basin as reported in the manuscript.

**About simulated and observed floods**: We agree that our results are far from a perfect match with observations. We also agree that a key issue is likely related to the cell size of the available digital elevation model. Nevertheless, our results show a good agreement in different regions of the stream and we consider it a step forward in the development of a low-cost tool for risk management.

P27L647 there is no information about flood 1, the authors can not speak about evidence of remarkable behavioral difference. We do not even know if a flood of significant magnitude occurred... The rainfall can not be described as spatially quasi-homogeneous. What do the author mean with "return flow" and "20 groups".

**About flood 1**: This could be a misunderstanding. As mentioned before (and as it has been made clear in the manuscript), there was only one flash flood event on May 18th. In the manuscript, we talk about two important rainfall events (Events 1 and 2), both leading to the flash flood occurring during event 2.

**About the '20 groups' and return flow**: As explained in the manuscript, for the spatio-temporal analysis, we divided the watershed cells into 50 groups according to their localization and distance to the outlet. The results of this analysis are shown in Figure 10. In this case, we are talking about the cells near the outlet (the first 20 groups). On the other hand, the return flow is a conceptualization of the model to represent saturation runoff generation. When the gravitational storage is full, the leftover is returned to the runoff.

L654 The authors can not state that the second convective core results mainly in surface runoff. First, the most affected area has hardly been saturated by the first event (spatially detailed result will probably show it) and in anyway, that is a simulation result and not necessarily reality.

See Figures 11 and 12. The spatio-temporal analysis of the simulation reveals a high runoff production on the upper region of the basin. Figure 11 shows that the convective streamflow accumulation has a behavior similar to the one obtained for the runoff portion of the streamflow. We realize this analysis is obtained from hydrologic model simulations, but it gives us an idea of the processes that might have taken place prior to the flash flood event.

Evolucion_evento.png

**Fig. 3.** Temporal evolution of discharge during Event 2 in different locations along the water-shed's main channel. The upper location corresponds to 15% of the area of the basin, and the other downstream locations to 52%, 76%, and 100% of the watershed, respectively.

Parameter_variation_analysis.png

**Fig. 4.** Sensitivity analysis of a) infiltration, b) runoff speed and c) subsurface speed parameters

VariandoHuEvento2.png

**Fig. 5.** Event 2 simulated varying the percentage of capillary storage in the initial conditions.

---

## Author Comment (AC3) · 12 Jan 2019

The comment was uploaded in the form of a supplement: https://www.hydrol-earth-syst-sci-discuss.net/hess-2018-452/hess-2018-452-AC3-supplement.zip

---

## Author Comment (AC4) · 12 Jan 2019

The comment was uploaded in the form of a supplement:
https://www.hydrol-earth-syst-sci-discuss.net/hess-2018-452/hess-2018-452-AC4-supplement.zip

---

## Author Comment (AC5) · 12 Jan 2019

The comment was uploaded in the form of a supplement:
https://www.hydrol-earth-syst-sci-discuss.net/hess-2018-452/hess-2018-452-AC5-supplement.zip

---

## Referee Report (RR1)

Review of the 2[nd] version of the paper : Reconstructing the Salgar 2015 Flash Flood Using Radar Retrievals and a Conceptual Modeling Framework: A Basis for a Better Flood Generating Mechanisms Discrimination. Nicolás Velásquez et al.

**General comments about the revision:**

I appreciate the revision of the manuscript, that make the paper to be more concise and better structured. Specially the rewriting of the introduction, the study site and data description and part of the methodology appears in a clearer way. I would however still suggest several rewording of the section 3.2 and 3.3. The theoretical background of the latter one must be clarified to be maintained in the manuscript, either by referring to the related literature, or by explaining the hypotheses behind the floodplain sub-model.

Finally concerning the methodology, I have one main doubt about the actual connection between the hydrological model set up and the landslide model one. The strength of the overall paper methodology, is to use the soil storage dynamic simulated by the hydrological model to deduce landslide occurrence. However, it does not appear clear any more if the soil storage set up (Z) is the same for both simulations. This assumption has to be clearly clarified.

The results and the discussion appear in a clearer way as well in that second version, specially when describing the results of the hydrological model, explaining the different flow processes during the two distinct events. I think that the results could even be better inserted within the current literature, showing standardize figures or adjusting the discussion. My comments below support this point.

The discussion about the limitation of the landslide model might be more more detailed. I think, the authors should be able to settle on the reasons of the result limitations: does it come from the lack of spatial information about soil and land cover properties, or from too strong assumptions of the landslide model? Ruling on the reasons of the limitation would bring a direct outlook of the study.

Finally, I found really interesting that the soil storage capacity available before flood event impacted not only the flood magnitude but also the response time of the catchment. From my point of view, it could be appear as one of the main incomes of the paper.

**I. Comments on the methodology's description:**

**I.1. Description of the shallow landslides sub-model :**

From my point of view, the description of the sub-model could be clearer expressed. It doesn't emphasize the crucial definition of the model. The stability state of the cells, which is presented on the first part of the section, depends on the stability criteria which is presented later. I suggest here a draft, but please feel free to adapt it :

"The shallow landslides sub-model coupled to the hydrological model is proposed by Aristizabal et al. 2016. The stability of each cell is calculated through the assessment of the different stresses applied to the soil. The stability of the soil decreases with the pore water pressure (Graham, 1984). The slope failure occurs when the saturated soil thickness above the slip surface ($Z_w$, here related to

the gravitationnal storage $S_3$, eq. 9) is greater than a critical saturated depth ($Z_c$), which depends on the soil cohesion (C), the hillslope (beta), […] (eq. 10).

Eq 9 : $Z_w = S_3 / (W_c − W_{fc})$; with $W_c$ and $W_{fc}$ the soil saturation depth and the field capacity respectively, $S_3$ the gravitationnal storage.

Eq 10 : […]

According to that soil stability definition, the topography and the soil properties, cells of the catchments are classified into 3 groups : i) the unconditionally stable cells for which the maximal value of $Z_w$ (i.e. Z) is smaller than $Z_c$; ii) the conditionally stable cells, for which the stability will depends on the saturated soil thickness ($Z_w$), and iii) the unconditionally unstable cells, for which their properties lead to unstable conditions for any value of $Z_w$. Shallow landslides are calculated at each time step of the hydrological simulation, on the second cell class, where the soil stability depends of the storm event."

In addition, I suggest to summarize the specific parameters to the landslide sub-model in a table as done for the hydrological model. I suggest as well to indicate the references used to set up the parameter values.

Finally, the lines 563 – 566, page 19, correspond to the landslide sub-model description; they should be inserted in that section.

     *I.2. Description of the flood plain sub-model :*

I still have some difficulties to understand the theory behind the calculation of the flood plain. I'm ok with the assessment of the water depth (eq. 11), the friction velocity (eq. 12) and the sediment concentration (eq. 13). Then, I have some trouble to follow the method. Could you please define what is a constitutive coefficient (r) and add a reference for the eq. 14 ?

The main trouble I had, is that you're going into sediment fluxes assessment to calculate flood plain area. What the gain of that method compared to a direct assessment of the flood plain area through the simulated discharge, the simulated velocity and the DEM ?

**II. Comments on the results and the discussion:**

- page 15, line 451 – 453: "The simulation shows that Event 1 generates a hydrograph with a peak flow of Qmax = 160 m3s−1 . It is important to note that during precipitation Event 1 there were no damage nor flooding reports by local authorities." Can you link the simulated peak flow value to a flooding / no flooding status ? I mean, could we consider the simulated peak flow as consistent with the fact that they were no flooding reports, or is that assessment to high ?

- Page 16, line 483. : "Although some of the surface speed values used in the analysis are unrealistically low". It would be better to directly limit the sensitivity analysis over a range of realistic speed values. The assessment of such a range might be done choosing a range of realistic roughness coefficient (Manning, Strickler, ...), and using the relation between runoff speed (v), and water level (h) under cinematic wave hypotheses ($v = S^{1/2}n^{-1}h^{2/3}$), with S the slope, and n the Manning coefficient. As you mentioned somewhere in the manuscript, this range of observed speed values finally has to be rescaled to be adapted to the model

resolution: the transit time along the water paths must be maintain although the DEM data processing can have modified the simulated water path lengths ($L_{num}$). In other terms, the ratio $L_{obs} / v_{obs} = L_{num} / v_{num}$ must be kept when rescaling the surface speed range.

- Page 16, comments on Figure 10: To describe the sub-catchment I would suggest to indicate as well the surface area [$km^2$] in addition to the percentage of the basin. I also think it could be really interesting to describe the flow peaks in terms of $m^3.s^{-1}.km^2$ which can be compared to flash flood features found in the literature (for example, see figure 6, Page 8 in Gaume et al, 2016).

- Page 17, comments on Figure 12: a) The transit times of the events 1 and 2 are really interesting, because the catchment response is slightly shorter in event 2, while the rainfall storm was located really in the upstream and remote part of the catchment. In contrast the first event was located closer to the outlet, and therefore we might have expected faster response. It's mean that the storage capacity before the event has not only an impact on the magnitude of the flood but also on it timing. The result is not straight forward, and could be mentioned here or in the discussion. b) It's also really interesting to see that the runoff and the subsurface flow start at the same in the event 2, while in the first one, we can see a delay of around 2 hours between the start of those two processes.

- Page 18, line 528-530: "In event 2, the convective rainfall and the runoff show a similar evolution, denoting a strong influence of the convective portion (figure 12b)". From my point of view, it rather means that the stream network (as there are mainly runoff) does not temporize, convert and attenuate the rainfall input signal.

- Page 18, Figure 13: I'm still not convinced by the interest of this figure, but it remains your paper.

- Page 19, line 578, about the landslide model set up. Do you mean that you have two different calibrations of Z (i.e. $S_3$) for the landslide model and for the hydrological model? If so, that makes the connection between both model simulations inconsistency...

- Page 19, about the landslide results: First, I would suggest no to insist so much to the difference between the observed and the simulated number of unstable cells. The way to observe landslide are quite qualitative, as – I assumed – it was certainly based on color differentiation between aerial views. The successive soil transport to the landslides and the soil spread through the runoff might have lead to detect unstable cells where there were only sediment charged overflow occurring over it. Second, I would suggest to rather focus on the spatial distribution. I slightly disagree with 'considerably well spatial distribution representation' . I agree that the upstream landslide were really well detected. However the false positive detection in the south part of the catchment should be discussed. Where are the limitations of the landslide model? Is it the fact that the same cohesion or other soil parameter are choose uniform over the catchment ? In that case, different land cover, soil textures, at the good positive and the false positive cells could support this hypothesis. Or is that the landslide model it self that is to simplified ? → improvement taking rainfall intensities, …

- page 20, lines 609 – 612: It would be interesting to add the proportion of the river length for the different orders.

- Page 21, lines 654 – 661: I would suggest to refer to the importance of the rainfall spatial distribution in connection / interaction with the soil storage capacity ones (Zocatelli et al,

2010, and to Douinot et al, 2016). Those are exactly the two main differences between the two events.

- Page 22, lines 671 – 674. The description of the different order are interesting but I don't see the link with the results present in section 4. Could you specify how did you deduce those assessments?

**III. Technical comments:**

- Page 4, line 121 – 129, when introducing the 3 models. Similarly to the short description of the hydrological model done between the lines 123 – 126, I suggest to keep the short description of both sub-model initially given in the first version of the manuscript :

"The shallow landslide sub-model follows the formulation described in (Aristizábal et al., 2016). The hydraulic sub-model corresponds to a low-cost 1D model (hereafter referred to as HydroFlash) that [assumes infinite sediment supply and] estimates the cross-sectional filled area at all time step."

- Page 4, line 123; page 5, line 137, page 13, line 390: the sub-model simulated flood plain inundation is alternatively called : "hydraulic sub-model", "inundation sub-model", "flash flood submodel", and "HydroFlash". For a sake of consistency, and clarity, I suggest to choose only one of those terms and use it everywhere in the manuscript when calling that sub-model.

- Page 5, line 161: please put bracket around "HAND"

- Page 6, line 190 – 192: By soil properties map, I was thinking to spatial distribution of the soil properties, as used to define the hydrological model. It would give an overview of the spatial distribution of the soil classes. I'm still thinking it is of interest. If you don't want to add another map, I would suggest just to adapt the color scale of the slope map to the soil classification. In that way, the reader could guess the spatial distribution and the proportion of each soil class. Another option, might be to add a column in table 1, with those proportions.

- Page 7, line 196 – 198 : " Unfortunately […]." I would suggest to remove this sentence for a sake of conciseness.

- Page 7, line 211 – 212: the range of value of the stream discharge [185 – 222] $m^3.s^{-1}$ does not correspond to the range of value of the velocities [5 – 7] $m.s^{-1}$. A maximum discharge of 259 $m^3.s^{-1}$ is expected. Please change either the velocity of the discharge stream range.

- Page 7, line 213: the sentence "The timing of peak flow is also [an] important information" can be removed.

- Page 7, line 216 – 222: Please refer to the figure 16. Some details of how the contrast between both images are calculated would be welcomed.

- Page 8, line 230 – 236 : the optimal distance for radar rainfall observation is described twice. Please remove "optimum" line 233 in "optimal optimum" and the sentence line 236 : "The results of the radar QPE methodology indicate that the rainfall estimation works well within a radius of 120 km."

- Page 8, line 256 – 258 : the sentence "Chocho is the [...]" can be removed.

- Page 10, line 297 : add a space between "Figure" and 6.

- Page 12, table 3: are the capillarity and the gravitational storage really in mm? Or rather in cm?

- Page 12, table 3: write "capillarity" instead of "capilary".

- Page 12, line 358 : write "10 parameters" instead of "ten parameters".

- Page 12, line 363 – 365 : please specify here the objective of the calibration. Here a suggestion: "The model simulation is calibrated to reach a base flow of 3 $m^3.s^{-1}$. The calibration consists in scaling each physical parameter by a constant value in the entire basin (Francés et al. (2007b)). Table 3 includes the mean value for all the parameters used in the model and the scalar value adjusted during the model calibration."

- Page 13, line 367 : the reference date is missing for "Aristizabal et al. ..."

- Page 13, equation 6, 7, 8, 9, 10 : In the equation, the index "i" refering to the cell "i" of the catchment is specified in the left side of the equation but not in the right one. Please either specify the "i" index to any or no one of the parameter with cell dependent value.

- Page 15, equation 16: the index "j" should be added to the parameter $F_{d,i}$ which depends on it.

- Page 15, line 449 – 451 : please remove the sentence "The model simulation is set up [...]". It refers to the method and therefore should be inserted in the section 3.

- Page 16, line 470: the observed timing could be indicated in brackets, as a remember.

- Page 16, line 482 : "particularly in the low end values", do you mean the discharge recession?

- Page 17, line 496 – 500 : those lines correspond to the method description and should be inserted in section 3.2.

- Page 17, line 516: The first sentence "It is well known [...]" is not relevant and could be remove for a sake of conciseness.

- Figure 2: unit of the slopes is [-] and not [mm.$^{-1}$]

- Figure 5, plot a) y-axis: please confirm the rainfall unit : [mm / 5 min] ?

- Within all the manuscript : choose between "sub-model" or "submodel" and maintain the same spelling.

References:

Eric Gaume, Marco Borga, Maria Carmen Llassat, Said Maouche, Michel Lang, et al.. Mediterranean extreme floods and flash floods . *The Mediterranean Region under Climate Change. A Scientific Update*, IRD Editions, pp.133-144, 2016, Coll. Synthèses, 978-2-7099-2219-7.

Audrey Douinot, Hélène Roux, Pierre-André Garambois, Kévin Larnier, David Labat, Denis Dartus, Accounting for rainfall systematic spatial variability in flash flood forecasting, Journal of Hydrology, Volume 541, Part A, 2016, Pages 359-370, ISSN 0022-1694, 10.1016/j.jhydrol.2015.08.024.

D. Zoccatelli, M. Borga, A. Viglione, G. B. Chirico, and G. Bl̈oschl. Spatial moments of catchment rainfall: rainfall spatial organisation, basin morphology, and flood response, 2011, HESS, doi:10.5194/hess-15-3767-2011

---

## Referee Report (RR2)

[referee-annotated manuscript omitted]

---

## Referee Report (RR3)

Review of the 3rd version of the paper : Reconstructing the Salgar 2015 Flash Flood Using Radar Retrievals and a Conceptual Modeling Framework: A Basis for a Better Flood Generating Mechanisms Discrimination. Nicolás Velásquez et al.

**General comments about the revision:**

I already did the revision of the two first versions of manuscript. I really appreciate the supplementary information added in the manuscript, namely the figure 9, explaining the floodplain iterative assessment; the new discharge data, that comforts the hydrological model results, and the information concerning the stream velocity in Annexe.

However, I'm a bit upset that there are still a lot of annotation issues, and writing mistakes that might exist on the first submission but really not of the third one. Several co-authors comments are also spread within the text. Quick checks of the homogeneity and the consistency of the equations and annotations should be done before submitting and NOT by the reviewer. As example in the hydrological model, you mentioned calibrated parameters that are not in the equations of the model. Also the indexation of the variables/ parameters are still sporadic.

I'm convinced there is a lot of work behind the paper, and that each part of the modeling framework should have been a main topic on its own, making easier the writing, the results analysis and the reading. Nevertheless I respect the choice of the paper to gather the modeling work in one paper. In addition I would have personally chosena slightly different way to highlight the results and to orientate the discussion  on several points. But again you're the only ones to choose the direction of your paper.

I divided my comment in three parts. The first one consists in several modifications of the section 3 to make clearer the methodology description, specially the hydrological model and the floodplain submodel description. Those comments and/or suggested modifications are very important and almost unavoidable (from my point of view) to clarify and make the method understandable. In the second part, I made some comments and suggestions on the results and on the discussion that should make the paper insights be highlighted. And finally in the last part, I pointed out the spelling or language mistakes I found.

**I. Comments on the methodology's description:**

**I.1. Description of the hydrological model :**

Page 13, line 303-305: I don't understand this sentence "Vertical flows are only time dependent, while lateral flows could also depend on the actual state of the tank (kinematic approximation)." The vertical flows also depend on the actual state of the tank, doesn't it? I would suggest to remove this sentence.

Page 13, lines 306-309: You mentioned 4 modifications of the hydrological model but I don't agree with the fact to classify two of them as modification:
- the first one, "the direct use of radar QPE [...]": it is not a modification, but it is a specific choice of rainfall inputs
- the 4th modification: "the development of two modules [...]": this refers to the "landslide submodel" and the "HydroFlash submodel". For my understanding, it is not a modification of the hydrological model, but additional modelling elements that use the results of the hydrological model. I'm sorry to be picky, but I think for an easy understanding of the paper,

you should strictly follow the subsections's structure and only mention here what is related to the hydrological model.

Page 13, 311-317: I prefer when this part was inserted in the results description. The cell classification is a tool to analyze the spatially heterogeneous response of the catchment. I would even suggest to merge the figure 7 and the figure 14, keeping only the 50 classes categorizations of the figure 7 used in the Figure 14. Moreover I don't understand the first sentence and the expression "soil-rainfall-discharge coupling holistically". Another option would be to introduce this in the same section as the virtual tracers.

Page 14, figure 6: in the Hydrological modelling panel, the storage tanks are called "Ti", but the "Si" parameters are not mentioned as the legend does.

Page 14, title section 3.1.1: the modelling modifications are not only related to runoff but to all the lateral flow; I would suggest to call this section "Lateral flow modelling modifications"

!!! Page 15 – 16: the variable $A_i(t)$, called in the text "sectional area of the storage", has actually no unit [-], according to the equation (6). Reading Velez's thesis, it seems rather to be a coefficient. The actual sectional area is $S_i(t)*\Delta x$. This error makes the understanding of the equation really complex, and even makes me doubt about the meaning of the equation.

Page 15: definition of $v_i$: I'm not familiar with the dimensionless variable $A_i$ you defined. I used to use relationship between velocity and hydraulic radius or storage water level.

Page 15-16, equations 1- 8: I would suggest to present the general equations that control all the lateral flow first (eq. 7, 6, 1), before indicating the particularity of each tank lateral flow (eq. 2-5).

Page 15, line 334: please call the slope in a different way, it might be confused with the tank levels $S_i$.

!!! Page16, equation 5: how we are suppose to understand the equation? There is more exponents than parameters.. please also simplify, specially if at the end you will use a regional parameter deduce from any catchment.. Here what is important is the fact that $v_5$ is depending not only on $A_5$ but also on the stream bed slope. Only 3 parameters should appear : $v_5 = \beta * Si(t)^\alpha * slope^\gamma$

Page 16, page 366-367: $\Delta x$ versus L: Actually according to Velez, 2011 page 89; the $\Delta x$ variable used his thesis correspond to the cell width; i.e. the resolution if the flow direction is orthogonal or res/sqrt(2) if the flow direction is diagonal...

To summary the remarks I did from page 12, line 290 to page 16, line 372, I would suggest to reword more or less as follow:
* * *
*"3.1 The Tetis hydrological model*

*We used a physically-based and distributed model developed and described in Vélez (2001) and Frances et al. (2007). The spatial distribution and the hydrological flow path schema is based on the 12.75 m resolution DEM data. In each cell, five tanks represent the hydrological processes including capillary (tank 1), gravitational (tank 2), runoff (tank 3), baseflow (tank 4) and channel storage (tank 5). The state of each tank varies as a function of vertical and lateral flows as shown in the diagram, where the storage is represented by Si [m] and the vertical input to each tank by Di*

*[m], which in turns depends on the vertical flow through tanks Ri [m]. Ei represents the downstream connection between cells, except for tank 1, where E1 represents the evaporation rate.*

*The original model fully described in Vélez (2001) and Frances et al. (2007) are modified to improve the representation of the flow processes that occur during flash floods (see section 3.1.1). In addition, two analysis tools of the hydrological modelling results are introduced: virtual tracers tracking precipitation origins as well as water paths over or through the soils; and catchment cell grouping (see section 3.1.2). The tools objective is to allow an analysis of the spatially distributed response of the catchment.*

*3.1.1 The hydrological lateral flow*

*From water balance applied on a cell, Velez (2001) defined the lateral output of each tank (Ei) as follow:*

*(1)      Ei(t) = Ai(t) * vi(t) *Δt*

*Where Δt [s] is the calculation time step, vi(t) [m/s] the lateral flow velocity and Ai(t) [-] the dimensionless section area of the tank defined by:*

*(2)      Ai(t) = Si(t) / (vi(t) *Δt + Δw)*

*with Si(t) the tank storage [m] and Δw the cell width [m].*

*To solve the equation (1) and calculate Ei(t), the unknown vi(t) has to be defined. The lateral flow velocity is usually linked to the water level (Si(t)) of the storage through the general definition*

*(3)      vi(t) = β * Si(t)$^{α}$      (or Ai(t) in your case, but I don't know how to introduce it).*

*where α and  β are parameters depending on the flow environment (flow in porous media, free surfacic flow, geometry of a channel, …). While the original version of the model used linear relationship between vi(t) and Si(t) for all the tanks (not sure); we modified the equations to better represent the non linear increase of the velocity in overland (v2), subsurface (v3) and channel flow (4).*

*+ description of the choice of (vi,Si) relationship base on lines 328 – 341.*

*3.1.2 Tools for spatial analysis of the results: virtual tracers and catchment cell grouping"*

*[...]*
* * *
Page 17, line 396: hydrological and not hydrologic

pages 17, line 410 – 412: I would suggest to speak about the calibrated parameters rather than the non calibrated parameters

page 18, table 3: the parameters you mentioned in the table are not in the related equations …

page 18, table 3: Assuming the velocity parameters correspond of the velocity of each flow when the related water storage is equal to 1; I would expect the increasing magnitude order: Subterranean

speed, subsurface speed, surface speed, channel speed. How can you explain that the subsurface speed is higher than all the other ones?

page 18, line 417: please remove "above the slip surface $Z_{i,w}$".

**I.2. Description of the* Hydroflash *model :**

page 19, line 441: I suggest the following title "the floodplain submodel Hydroflash"

page 21, line 477: $A_{i,sed}$ meaning: Is really the flooded area (area along x,y) or the sectional area along the cross profile (area along z,y, x being the stream flow direction)? According to the attributed name, it seems to be the first definition; but according to the figure 9, I would say the second definition. It makes a big difference...

page 19-20 : Hereafter I'll suggest some rewording, introducing ALL the annotations. The equation references has to be added. It is roughly drafted. Please, feel free to integrate or not.
* * *
*"The HydroFlash submodel is designed/developed to interpret the hydrological model simulations as floodplain inundations (figure 9). For each stream cell and at each time step, the submodel: i) calculates the stream discharge including sediment load (eq. 14 – 18, Rouse, […], Takahashi, 1991); and ii) determine the resulting inundated cells according to the cross-profile of the stream, the sectional area, and the stream velocities when including the sediment load (eq. 17 -21, Takahashi, 1991).*

*I) To determinate the stream discharge 'including sediment load/transport' (Qsed)* *(→ has to be reformulate), a realistic channel width is firstly calculated according to Leopold (1953) approach:*

*$Wi = 3.26Qi^{-0.469}$*

*Then assuming a infinite sediment and ruble supply, the equations 14, 15, 18 are successively applied to deduce from the channel width $W_i$, the water level $Y_i$ (eq. 14), the friction velocity $v_{fr,i}$ (eq. 15, Keulegan and rouse equation, REF), the sediment concentration $c_i$ (eq. 16) and finally the sediment loaded stream discharge (eq. 18). The above mentioned relationships depend of 2 meaning parameters: the maximum sediment concentration ($C_{max}$ [-]) and the characteristic diameter of the sediments $D_{50}$ [m]. Both are assumed to be constant and respectively equal to 0.75 (O'Brien, 1988) and 0.138 (Golden & Springer, 2006).*

*ii) To calculate the inundated cells, the flood depth ($F_{di}$) and the sectional area of the stream ($A_{i,sed}$) are iteratively calculated along the cross profile to reach the relation between the stream discharge with sediment load ($Q_{i,sed}$) and the morphological properties of the cross profile. The latter is defined by the DEM. The relationship between $Q_{i,sed}$ and the morphological properties of the stream ($F_{di}$, $A_{i,sed}$) is defined in eq. 19:*

*(19)        $Q_{i,sed}(t) = (1/5) ri(t) * (F_{di}^{N})^{3/2} S_o * A_{i,sed}^{N}$*

*where ri is the constitutive coefficient of the flow, defined in eq. 7, summarizing the flow dynamics associated with sediments and colliding particles.*

*And with:*

*(20)        $F_{di}^{N} = Fdi^{N-1} + \delta y$*

*(21)*          $A_{i,sed}{}^{N} = \delta x \ sum(from \ j=1, to \ = N)[F_{di}{}^{N} - E_{bedi}{}^{j}] \ with \ E_{bedi}{}^{j} < F_{di}{}^{N}$

*with :*
-       $E_{bedi}{}^{j}$ *the DEM elevation of the Nth cross-profile cell closest to the stream cell*
-       $\delta x$ *the cell resolution*
-       $\delta y$ *the flood depth incrementation in the iterative process.*
* * *
Page 19-20, about the HydroFlash model: I'm curious to know about the ratio Qsed/Qsim: what is the range of value of c? Is there a significant change to include the sediments when calculating the floodplain?

**II. Comments on the results and discussion**

- page 21-22, section 4.1: On the one hand, the model simulated a flood peak in the upper range of the discharge peak assessment and the simulated flood peak occurred 20 minutes earlier than the observed one. On the other hand, when doing the sensitivity analysis on the surface speed parameter, decreasing the runoff velocity, the simulated flood peak is diminished and occurs later. Why didn't you calibrate better the surface speed, as the model is sensitive ?

- Page 22, line 506-511: I think those results are insights of the paper. They should be discussed in the discussion part to confront them to the literature (if there is) and to highlight them.

- Page 25, figure 13: the specific flood peaks are interesting. The simulated values are below the envelopp Qpeak = $97*A^{-0.4}$ that make the simulated flood peak consistent with the litterature on flash flood (Gaume et al. 2009). You should mention it on the discussion to strengthen the flood peaks simulation consistency.

- Page 27, line 595-600: I think my previous comment was misunderstood. I think you were right saying: "In event 2, the convective rainfall and the runoff show a similar evolution, denoting a strong influence of the convective portion (figure 12b)". But I think there were one unmentioned condition to observed similar evolution. The similar evolution comes from the fact that the convective portion is totally controlling the runoff processes AND that there is no effect of the stream network to modify or temporize the runoff advent at the outlet. In other words, it's possible to get strong influence of the convective rainfall runoff without having similar evolution, if the stream network buffers the runoff advent.

- Page 27, line 615-617 about the soil depth definition: You justified here the scaling factor by adjusting an underestimated soil depth observation. But then it means that the soil depth definition previously chosen for your hydrological model are also underestimated. I would rather assume that you need to calibrate the model to make the landslide occurring. The scaling factor might explained as to be an artifact of a too simplistic model, and a non calibration of the other parameters.

- Page 29, line 631-632: This comment should appear in the end of the discussion or in the conclusion, not in the result section.

- Page 31, line 669-670. I would refer to Zocatelli et al (2011) as following "Zocatelli et al (2011) found similar results in … (where, and which size of catchement)." As you wrote, it seems that Zocatelli et al (2011) found your own results.

- Discussion: As said before, it would be nice if the results of the landslide model and of the floodplain model are discussed. Here some ideas for the landslide model.
  The facts:
  ◦ your model relates landslides to soil depths, soil water content and topography.
  ◦ the soil depth spatial distribution is roughly done according to the topography.
  ◦ Landslide occurring is therefore only related to soil filling and the combined 'topographical-soil depth properties'.
  ◦ 1) Crossing topographical map and false simulated landslides location, those latter ones seem to appear where there are slope greater than 2. → Is 20 cm soil depth on a 2 slope realistic? Or could it explain the false simulated landslides?
  ◦ 2) The observed landslide is observed where the amount but all the intensities of the rainfall are the highest. → Could the rainfall intensity have an impact on landslide and explain why the model is failing (as not taking into account).

**III. Technical comments:**

- Page 5, line 120: As said before, keep the same name to call the different submodels of your modelling framework : I would suggest to use 'hydrological model' and even use it name 'Tetis' (Velez et al, 2002), for the first modelling part; 'landslide submodel' for the second modelling part and 'HydroFlash floodplain submodel' for the third modelling part.

- Page 5, line 122: choose to totally insert or remove "assumes infinite sediment suppply and'

- Page 5, line 123: 'hydrological' and not 'hydrologic'

- Page 5, line 122-129: please put the small description in the order it appears in the text: first the hydrological model, second the landslide submodel, third the HydroFlash floodplain submodel.

- Page 6, line 149-153: From my point of view, I would remove those sentences from this section. The aim of the section is to describe the catchment, not to come back on the objectives of the study. If you want to emphasize the challenge to work with scarce physiographical information, you should mention within the introduction for example when speaking to ungauge catchment (end of line 112 for example).

- Page 6, line 166: by brackets, I would say ().

- Page 8, figure 3: please ad the zoom number on the first top window.

- Page 8, line 195 and somewhere else in the manuscript: unity should not be in italic font.

- Page 10, line 241-242: remove the sentence "the results of the radar [...]" as the same information is done in the sentence line 238-239.

- Page 12, title of the section 3.1: I would suggest to choose "the hydrological model Tetis" as there is only the description of the model in this section (and not the 2 linked submodels). Or do you consider the that the framework consists in the the model plus the analysing tools (tracers and catchment cells grouping)?

- Page 22, line 510: please remove "On the other hand".

- Page 22, line 517: 'According to the model simulations, the peak flow occurred at **approximately** 2.20am LT': Why did you say "approximately" ? You have a solely simulation, that should give **exactly** one flood peak time.

- Page 22, line 521, when describing the figure: To make easier the manuscript reading, you should mention the studied parameters in the same order they appear in the figure: parameter of the top panel, parameter of the center panel, parameter of the bottom panel.

- Page 23, line 534: please define the acronym 'SIATA'

- Page 23, line 545: write 'skillfully' and 'skillfylly'

- Page 24, figure 11: write 'top', 'center' and 'bottom' panels instead of 'left', 'middle' and 'right' panels

- Page 27, line 617 and table 4, page 19: The scaling parameter for the soil depth is not the same within the text and in the table.

- Page 28, line 622-624: I would remove those two sentences.

- Page 31, line 656: "abilities" or "capacities" instead of "capabilities".

- Page 32, line 691: Event 1 and 2 as you already choose the brackets to distinguish between convective (stratiform) events.

---

## Referee Report (RR4)

Review of the 4$^{th}$ version of the paper : Reconstructing the Salgar 2015 Flash Flood Using Radar Retrievals and a Conceptual Modeling Framework: A Basis for a Better Flood Generating Mechanisms Discrimination. Nicolás Velásquez et al.

**General comments about the revision:**

I already did the revision of the three first versions of manuscript. I really appreciated the clarification of the method as well as the new points supplied into the discussion. As the limitations of the results (related to the applied method and the data availability) are now clarified, the present manuscript gained in reliability. It is now easier to see what has been achieved, what has been limited, and what would be the next step. Also as the method has been now well exposed – and even though I'm not totally convinced by several avenues pursued by the authors – any reader has the possibility to understand the scientific guidelines and to make it own review. For those reasons, I contend that this manuscript could be accepted when answering only one question about the floodplain methodology and looking at some writing revisions (here below).

**Comments on the floodplain methodology:**

Concerning the 1D floodplain methodology, I really don't understand why you are going to such complex method, looking for the loaded sediment while you do not have any data to apply it. Why don't you look directly at the floodplain according to the flood peak discharge without sediment? Moreover, it introduced a big inconsistency within the overall methodology: on the one hand you assessed the flood peak according to mud and flood prints on one river section; on the other hand you use this value to validate your hydrological model while this value must actually content both water + sediment discharge. I agree this is mainly done in hydrological modelling assuming a negligible amount of sediment. But here, as you introduced the sediment discharge in a third step, and above all as you found a non negligible 30 % of loaded sediments, it means that the hydrological model validation on (water) flood peak is not correct anymore. Could you justify that?

**Comments on the writing:**

I thanks the authors for the effort done to describe the methodology (section 3) of the paper. This time, I have not detected any undefined variable or parameter. Being picky, I just noticed that the slope is called $M_{i,0}$ when being described in the THETIS model, while $\beta_{i,0}$ is used in the landslide submodel. That could be attuned.

A general comment: the authors often use the word "approximately" even for quantities that shouldn't be approximative (e.g the distance between the radar and the basin is approximatively 90 km, line 222, page 10). I suggest to simply remove that word in those case that doesn't make sense.

In the following,  I suggested some modifications to improve the writing and make easier the reading. The main idea was to simplify some sentence where too many details were given and where – from my point of view – the main message was then lost.

   *Abstract.*

Page 1, line 17 : "Simulation results indicate that the flash flood and regional landslide features were strongly influenced by the antecedent rainfall, which was associated with a northeasterly stratiform event. The latter  recharged the gravitational and capillary storages within the model,

moistening the entire basin before the occurrence of the flash flood event and  impacting the subsurface-runoff partitioning during the flash flood event.”

**1. Introduction**

Page 2, line 31-47: “Several authors have assessed the role of the geological and geomorphological features of the catchment, soil type, soil moisture conditions, and the spatiotemporal structure of rainfall on flash flood occurrence,  identifying the leading causative mechanisms of this hazard (Merz and Blöschl, 2003). Adamovic et al. (2016) and Vannier et al. (2016)  related the flash floods governing processes to the geological properties of the basin with mixed results. Wu and Sidle (1995) emphasized the role of the topography, ground cover, and groundwater in the occurrence of shallow landslides and associated debris flows.  Many authors have assessed the influence of hills and stream slopes, suggesting the slopes of the hills are significantly more important for flash flood occurrence and magnitude than the slope of the stream (Šálek et al., 2006; Roux et al., 2011; Yatheendradas et al., 2008, Younis et al., 2008). Rodriguez-Blanco et al. (2012) analyzed flash flood episodes in Spain and determined that antecedent soil moisture conditions play a significant role in runoff production. Castillo et al. (2003) also suggested an  significant correlation between flash flood magnitude  and the antecedent moisture conditions. Aronica et al. (2012) used spatial and statistical analysis to reconstruct landslides and deposits, finding a connection between flash flood occurrence and soil moisture antecedent conditions.”

Page 2, line 48-54: “The fact that small basins are more prone to flash floods  (Wagener et al., 2007) makes difficult their measurement and, consequently, their understanding and  their prediction (Hardy et al., 2016; Ruiz-Villanueva et al., 2013; Yamanaka and Ma, 2017; Borga et al., 2011; Marra et al., 2017). The local rainfall storm events related to flash floods require that  high spatio-temporal resolution precipitation data (Norbiato et al., 2008).  Some authors follow a climatological approximation to assess the recurrence of flash floods in particular regions, focusing on the atmospheric causative mechanisms.”

Page 3, line 57-61: “Schumacher and Johnson (2005) studied extreme rain events associated with flash flooding in the United States over a 3-year period, using the national radar reflectivity composite data.  They found that 65% of the total number of flash floods are[...]”

Page 4, line 96: please change “with non-existing” by “without any”.

Page 4, line 98: please remove “certainly not available in real time”.

Page 5, line 111: please change “the second issue” by “the second one”

Page 5, line 114-115: The modeling framework used in this study is the TETIS hydrological model (Vélez, 2001; Francés et al., 2007), that has been modified to include a shallow landslide sub-model, and a floodplain submodel termed HydroFlash.

Page 5, line 116: please replace “termed” by “called”.

Page 5, line 121: please put in brackets the reference *Aristizabal et al, 2016.*

Page 5, line 121-123: "HydroFlash is a low-cost 1D model that  estimates the cross-sectional filled area at all time steps on the basis of  the liquid discharge and the sediment transport."

Page 5, line 131-133: "Section 3 presents a description of the overall methodology and the TETIS model , including flow separation, [...]"

**2. Study site and data**

Page 6, line 139-141: "By 2015,  Salgar counted  17 400 inhabitants , including 8 800 residing in the urban area. La Liboriana basin joins the El Barroso river basin, and both drain to the Cauca River."

Page 6, line 142-143: "The availability of the ALOS-PALSAR DEM (ASF, 2011), with an approximate resolution of  12.7 m, allows to estimat the main geomorphological features of the basin."

Page 6, line 162-163:  The social challenges lie in the high vulnerability of Salgar, given the location of the main urban settlement (see Figure 2).

Page 7, line 173-174: In Figure 3, the Zoom 4, corresponds to the first affected urban area from upstream to downstream during the flash flood. It is also possible to see a marked presence of crops and some patches of forest.

Page 9, line 213: please replace "are central to" by "are used to".

Page 10, line 216-220: "The assessment of the 2015 Salgar flash flood event following a hydrological modeling strategy uses a radar-based QPE technique described in Sepúlveda (2016) and Sepúlveda and Hoyos (2017), using radar reflectivity fields,  rainfall gauges and disdrometers ."

Page 10, lines 222-223:
[…] The radar is  90 km away from the basin. It has an optimal range in a radius of 120 km for rainfall estimation and a maximum operational range of 240 km for weather detection.

Page 10, lines 224: please replace "every 5 minutes" by "at 5 minute time step".

Page 10, line 240: I would suggest to replace "Event 2 corresponds to approximatively 38 mm; however, over the upper watershed, the accumulation exceeded 180 mm according to the estimated rainfall amounts based on the radar measurements" by "Event 2 corresponds to a moderate average of 38 mm,  however the accumulation exceeded 180 mm over the upper watershed".

Page 10, line 244: please write "[…] for the region. However, the combination" instead of "[…] for the region; however, but the combination […].

**3. Methodology**

page 15, line 338-342: The model archives the results of the virtual tracing algorithm at the outlet of the basin and for each reach, enabling  to study  the different flow paths and water

origins  at different spatial scales.

---

## Author Response (AR2)

**Response to Reviewer 1**

May 28, 2019

Manuscript title: Reconstructing the Salgar 2015 Flash Flood Using Radar Retrievals and a Conceptual Modeling Framework: A Basis for a Better Flood Generating Mechanisms Discrimination

Authors: Nicolás Velásquez, Carlos D. Hoyos, Jaime I. Vélez, and Esneider Zapata

**General Concept:**

I appreciate the revision of the manuscript that makes the paper to be more concise and better structured. Specially the rewriting of the introduction, the study site and data description and part of the methodology appears in a clearer way. I would however still suggest several rewording of the section 3.2 and 3.3. The theoretical background of the latter one must be clarified to be maintained in the manuscript, either by referring to the related literature, or by explaining the hypotheses behind the floodplain sub-model.

We sincerely thank the anonymous reviewer #1 for the careful, detailed and thoughtful review. The first and the second round of reviews were key to improve the manuscript. We have followed the detailed suggestions and have rewritten some of the paragraphs in sections 3.2 and 3.3. We also have expanded the description of the floodplain submodel, adding an schematic diagram explaining step-by-step the flash flood submodel methodology. The explanation is included in the section: HydroFlash. Below, the reviewer comments are in black and our comments are in "blue".

Finally concerning the methodology, I have one main doubt about the actual connection between the hydrological model set up and the landslide model one. The strength of the overall paper methodology is to use the soil storage dynamic simulated by the hydrological model to deduce landslide occurrence. However, it does not appear clear any more if the soil storage set up (Z) is the same for both simulations. This assumption has to be clearly clarified.

The connection is given by Equation 9 in the manuscript, where the saturated soil thickness is defined as a function of the gravitational storage simulated by the hydrological model. The saturated soil thickness is then compared to the critical value, which in turns depends on Z.

The results and the discussion appear in a clearer way as well in that second version, specially when describing the results of the hydrological model, explaining the different flow processes during the two distinct events. I think that the results could even be better inserted within the current literature, showing standardize figures or adjusting the discussion. My comments below support this point.

We have taken in consideration all the detail suggestions, and we have modified the manuscript accordingly.

The discussion about the limitation of the landslide model might be more more detailed. I think, the authors should be able to settle on the reasons of the result limitations: does it come from the lack of spatial information about soil and land cover properties, or from too strong assumptions of the landslide model? Ruling on the reasons of the limitation would bring a direct outlook of the study.

We believe it is related mostly to the data limitations, as discussed in the new version of the manuscript.

Finally, I found really interesting that the soil storage capacity available before flood event impacted not only the flood magnitude but also the response time of the catchment. From my point of view, it could be appear as one of the main incomes of the paper.

We agree with the reviewer. This point has been highlighted in the conclusions.

**1 Comments on the sections**

**1.1 Comments on the methodology description**

**Description of the shallow landslides sub-model**

• From my point of view, the description of the sub-model could be clearer expressed. It doesn't emphasize the crucial definition of the model. The stability state of the cells, which is presented on the first part of the section, depends on the stability criteria which is presented later. I suggest here a draft, but please feel free to adapt it :

The shallow landslides sub-model coupled to the hydrological model is proposed by Aristizabal et al. 2016. The stability of each cell is calculated through the assessment of the different stresses applied to the soil. The stability of the soil decreases with the pore water pressure (Graham, 1984). The slope failure occurs when the saturated soil thickness above the slip surface (Zw, here related to the gravitationnal storage S3, eq. 9) is greater than a critical saturated depth (Zc), which depends on the soil cohesion (C), the hillslope (beta), [...] (eq. 10).

Eq 9 : Zw = S3 / (Wc - Wfc); with Wc and Wfc the soil saturation depth and the field capacity respectively, S3 the gravitationnal storage.

**Eq 10 : [...]**

We sincerely thank the reviewer for the helpful suggestion. We adopted most of the suggestion by the reviewer, and also changed the order of the text and the equations, trying to present a better explanation of the submodel. The modified text appears in the manuscript as follows:

The shallow landslide submodel coupled to the hydrologic model is proposed by (author?) [1]. The stability of each cell is calculated through the assessment of the different stresses applied to the soil. The stability of the soil decreases with the pore water pressure [?]. The slope failure occurs when the saturated soil thickness above the slip surface  $Z_{i,w}$  (equation (1)), which depends on the gravitational storage  $S_{3,i}(t)$ , the soil wilting point  $W_{i,pwp}$ , and the soil field capacity  $W_{i,fc}$ , is greater than a critical saturated depth  $Z_{i,c}$  (equation (2)). The critical saturated depth depends on the shallow soil depth  $Z_i$ , the soil bulk density  $\gamma_i$ , the water density  $\gamma_w$ , the gradient of the slope  $\beta_{i,0}$ , the soil stability angle  $\phi_i$ , and the soil cohesion  $C'_i$ . Figure 7 [in the manuscript] describes the variables of the model and the balance of forces considered, and Table 1 presents the parameters for this model.

$$Z_{i,w}(t) = \frac{S_{3,i}(t)}{W_{i,cfc} - W_{i,pmp}}$$
(1)

| Parameter Name       | Symbol                    | $\operatorname{Scalar}$ | Mean Value | Spatial distribution |
|----------------------|---------------------------|-------------------------|------------|----------------------|
|                      |                           | Parame-                 |            |                      |
|                      |                           | ter                     |            |                      |
| Soil depth           | $Z_i  [\mathrm{mm}]$      | 0.3                     | 300        | As a function of the |
|                      |                           |                         |            | slope                |
| Topography slope     | $\beta_{i,0}$ [adim]      | 1                       | 0.01 - 5.3 | Taken from the DEM   |
| Soil bulk density    | $\gamma_i [KNm^{-3}]$     | 1                       | 18         | Assumed constant     |
| Water density        | $\gamma_w [KNm^{-3}]$     | 1                       | 9.8        | Constant             |
| Soil stability angle | $\phi_i$ [ 0 ] | 1                       | $30^{0}$   | Assumed constant     |
| Soil cohesion        | $C'_i$ [KN]               | 1                       | 4          | Assumed constant     |

Table 1: Shallow landslides model parameters.

$$Z_{i,c} = \frac{\gamma}{\gamma_w} Z_i \left( 1 - \frac{tan\beta_i}{tan\phi_i} \right) + \frac{C'}{\gamma_w cos^2 \beta_{i,0} tan\phi_i}$$
(2)

According to the soil stability definition, the topography and the soil properties, all cells are classified into three: unconditionally stable, conditionally stable and unconditionally unstable. In particular, three parameters determine the stability of each cell: (i) residual soil thickness water table  $Z_{i,min}$  (equation (3)), (ii) the maximum soil depth at which a particular soil remains stable  $Z_{i,max}$  (equation (4)), and (iii) the maximum slope at which the soil remains stable  $\beta_{i,0}$  (equation 5)).

$$Z_{i,min} = \frac{C'_i}{\gamma_w \cos^2\beta_{i,0} tan\phi_i + \gamma_i \cos^2\beta_{i,0} (tan\beta_{i,0} - tan\phi_i)}$$
(3)

$$Z_{i,max} = \frac{C'}{\gamma_i cos^2 \beta_i, 0(tan\beta_i, 0 - tan\phi_i)}$$
(4)

$$\beta_{i,0} = \tan^{-1} \left[ \tan \phi_i \left( 1 - \frac{\gamma_w}{\gamma_i} \right) \right] \tag{5}$$

A cell is unconditionally stable when  $Z_i$  is smaller than  $Z_{i,min}$  or when the cell slope is smaller than  $\beta_{i,0}$ . On the other hand, a cell is unconditionally unstable when  $Z_i$  is greater than  $Z_{i,max}$ , and finally, a cell is conditionally stable when  $Z_i$  is between  $Z_m in$  and  $Z_{i,max}$ . Shallow landslides are calculated at each time step of the hydrological simulation, based on the latter cell class, where the soil stability depends of the storm event, becoming unstable when  $Z_{i,w}(t)$  is greater than  $Z_{i,c}$ .

• In addition, I suggest to summarize the specific parameters to the landslide sub-model in a table as done for the hydrological model. I suggest as well to indicate the references used to set up the parameter values.

We adopted the good suggestion by the reviewer and included a table with the description of the landslide submodel parameters and their values. Table 1 corresponds to the new table in the manuscript:

• Finally, the lines 563 – 566, page 19, correspond to the landslide sub-model description; they should be inserted in that section.

We included the parameters of the land slide model in Table 4 in the new version of the manuscript. For this reason, we deleted the lines corresponding to 563 to 566 in the previous version of the document.

**1.1.1 Description of the flood plain model**

• I still have some difficulties to understand the theory behind the calculation of the flood plain. I'm ok with the assessment of the water depth (eq. 11), the friction velocity (eq. 12) and the sediment concentration (eq. 13). Then, I have some trouble to follow the method. Could you please define what is a constitutive coefficient (r) and add a reference for the eq. 14 ?.

We agree with the reviewer; the original explanation of the submodel was confusing. For this reason we edited the corresponding subsection, and added a new schematic figure explaining the steps followed by HydroFlash as follows:

[revised manuscript text omitted]

Regarding the constitutive coefficient, we expanded the HydroFlash description including a definition of constitutive coefficient, as shown in the previous paragraphs. Also, in the text we mention that equation (14) is derived form the Keulegan and Rouse equations.

• The main trouble I had, is that you're going into sediment fluxes assessment to calculate flood plain area. What the gain of that method compared to a direct assessment of the flood plain area through the simulated discharge, the simulated velocity and the DEM?.

We could apply the proposed method using only the simulated streamflow. However, there is evidence that suggests that a significant portion of the total discharge corresponds to a concentration of sediments during the event (Qtotal= Qliquid + Qsediments). In the Colombian Andean region, flash flood events tend to have high sediment loads, and we wanted to develop a low-cost model that represents the different processes involved in the flood.

**Comments on the results and the discussion**

L451 The simulation shows that Event 1 generates a hydrograph with a peak flow of Qmax = 160 m3s1. It is important to note that during precipitation Event 1 there were no damage nor flooding reports by local authorities." Can you link the simulated peak flow value to a flooding / no flooding status ? I mean, could we consider the simulated peak flow as consistent with the fact that they were no flooding reports, or is that assessment to high ?.

Days after the event we talked with the community and the authorities of the region, and they reported high levels in the river the day before the flash flood event, with no inundation or affected people/infrastructure. According to our estimations, a discharge close  $200m^3s^{-1}$  is required for an inundation to occur in the area.

L483 "Although some of the surface speed values used in the analysis are unrealistically low". It would be better to directly limit the sensitivity analysis over a range of realistic speed values. The assessment of such a range might be done choosing a range of realistic roughness coefficient (Manning, Strickler, ...), and using the relation between runoff speed (v), and water level (h) under cinematic wave hypotheses ( $v = S^{1/2}n^{-1}h^{2/3}$ ), with S the slope, and n the Manning coefficient. As you mentioned somewhere in the manuscript, this range of observed speed values finally has to be rescaled to be adapted to the model resolution: the transit time along the water paths must be maintain although the DEM data processing can have modified the simulated water path lengths (*Lnum*). In other terms, the ratio *Lobs/vobs = Lnum/vnum* must be kept when rescaling the surface speed range.

We followed the suggestion by the reviewer and only used realistic speeds.

P16 Comments on Figure 10: To describe the sub-catchment I would suggest to indicate as well the surface area [km2] in addition to the percentage of the basin. I also think it could be really interesting to describe the flow peaks in terms of m3.s-1.km2 which can be compared to flash flood features found in the literature (for example, see figure 6, Page 8 in Gaume et al, 2016).

We had included the size of each sub-cachment in the figure, however, the text was very small. We have modified the figure to increase the text. In the same image we include the relation between the peak flow and the area.

P17 Page 17, comments on Figure 12: a) The transit times of the events 1 and 2 are really interesting, because the catchment response is slightly shorter in event 2, while the rainfall storm was located really in the upstream and remote part of the catchment. In contrast the first event was located closer to the outlet, and therefore we might have expected faster response. It's mean that the storage capacity before the event has not only an impact on the magnitude of the flood but also on it timing. The result is not straight forward, and could be mentioned here or in the discussion. b) It's also really interesting to see that the runoff and the subsurface flow start at the same in the event 2, while in the first one, we can see a delay of around 2 hours between the start of those two processes.

Figure 5 shows the temporal evolution of the rainfall over the watershed. According to this figure, the convective (and more intense) portion of the rainfall during Event 1 occurred near the end of the event, explaining the delayed peak discharge time. Additionally, Figure 13 presents the spatio-temporal evolution of the gravitational storage. According to the model, this storage responds quickly to the rainfall, and also, the more relevant increments ocurred at the end of event 1. The described behavior is coherent with Figure 15, in which the elapsed time between the streamflow and the rainfall is shorter near the end of the event. We added a comment in the manuscript.

L528 In event 2, the convective rainfall and the runoff show a similar evolution, denoting a strong influence of the convective portion (figure 12b). From my point of view, it rather means that the stream network (as there are mainly runoff) does not temporize, convert and attenuate the rainfall input signal.

According to former Figure 12b, streamflow runoff accumulation is similar but not equal to the convective accumulation. There is a time lag between both lines, denoting that there are conversions and temporizations due to the stream network. However, we find it very interesting how this relation changes among both events, and we think that this could be related to other factors such as the soil moisture and the spatiotemporal structure of the rainfall. Also, we believe that more work is required in this direction, exploring how rainfall-runoff relations evolve in different events. We included the following text in the document: According to [new] Figure 15b, the accumulations of streamflow runoff and convective rainfall become similar with the increase in time. Additionally, the runoff has a lag and shows signs of attenuation of the convective signal. However, this description only applies for the runoff portion, since the evolution is different when we consider the total simulated streamflow.

P18 Figure 13: I'm still not convinced by the interest of this figure, but it remains your paper.

We erased the Figure from the manuscript.

L578 About the landslide model set up. Do you mean that you have two different calibrations of Z (i.e. S 3 ) for the landslide model and for the hydrological model? If so, that makes the connection between both model simulations inconsistency.

The parameter  $S_3$  of the model refers to the gravitational water storage. On the other hand, Z is the soil depth. Although both variables are related, they correspond to two different characteristics of the soil.

P19 About the landslide results: First, I would suggest no to insist so much to the difference between the observed and the simulated number of unstable cells. The way to observe landslide are quite qualitative, as – I assumed – it was certainly based on color differentiation between aerial views. The successive soil transport to the landslides and the soil spread through the runoff might have lead to detect unstable cells where there were only sediment charged overflow occurring over it. Second, I would suggest to rather focus on the spatial distribution. I slightly disagree with 'considerably well spatial distribution representation'. I agree that the upstream landslide were really well detected. However the false positive detection in the south part of the catchment should be discussed. Where are the limitations of the landslide model? Is it the fact that the same cohesion or other soil parameter are choose uniform over the catchment ? In that case, different land cover, soil textures, at the good positive and the false positive cells could support this hypothesis. Or is that the landslide model it self that is to simplified ? improvement taking rainfall intensities.

We fully agree with this reviewer's take on the results. Considering both reviewer's suggestions, which do not necessarily coincide in this case, we change the manuscript as follows: The model represents the spatial distribution of the areas that are prone to trigger shallow landslides during Event 2 reasonably well, especially in the upper part of the basin, showing a significant density of unstable cells in the hills where slides took place. This result is important because in that sense, it serves to verify the capability of the model to estimate risk areas only considering topography and rainfall data. On the other hand, there are some false positives in the middle of the basin, which could be related to the poor description of the soils. The landslide model has been used in a nearby watershed with similar characteristics, but with high-quality distributed information [1]. In that case, the model shows a better performance, which highlights the relevance of the input data. A pinpoint localization of the unstable cells is still considered a hard task, in part due to the small temporal and spatial scale at which landslide processes take place [1, 2, 8]. Notwithstanding the difficulties, the results suggest that the model simulations could have been used and should be used in the future for early detection and warning to improve both short- and long-term risk reduction strategies.

L609 It would be interesting to add the proportion of the river length for the different orders.

We agree with the reviewer. We now include a table with a summary of the hills and streams slopes, stream lengths, flooded percentage by order, and relative flooded percentage respect to the total order length.

Table 2: Channels and flooded cells percentages summary.  $Sh_0$  and  $Ss_0$  correspond to the mean hill and stream slope respectively. L corresponds to the total channel length. F Spots and S spots correspond to the flooded and slides percentages respectively.

| Order      | $Sh_0$ [%] | $Ss_0$ [%] | $L \ [km]$ | F Spots | Ss Spots |
|------------|------------|------------|------------|---------|----------|
| 1          | 60         | 37         | 59         | 5       | 64.5     |
| 2          | 57         | 27         | 26         | 6       | 26.3     |
| 3          | 49         | 13         | 16         | 18.5    | 5.5      |
| 4          | 43         | 9          | 10         | 38.5    | 3.6      |
| 5          | 42         | 6          | 6          | 32      | 0.05     |
| Mean/total | 50         | 18         | 117        | 100     | 100      |

L654 I would suggest to refer to the importance of the rainfall spatial distribution in connection / interaction with the soil storage capacity ones (Zocatelli et al,2010, and to Douinot et al, 2016). Those are exactly the two main differences between the two events.

We agree with the reviewer and have included the reference to both papers. Additionally, we expanded the discussion including the following paragraph in the document: Additionally, when we compare Events 1 and 2, there is an interplay between the rain spatial structure and the soil storage capacity. During Event 1, there is almost no saturation, hence runoff production is low, while Event 2 is influenced by the pre-event water and the occurrence of multiple convective systems over the same region. The structure of the rainfall associated with the La Liboriana event and its interaction with the soils highlights the need to consider in more detail the role of

orographic rainfall intensification in practical applications such as early warning systems. Evidence suggests the spatial structure of the rainfall is at least as important as the geomorphological features of the basin in regulating the generation of flash flood events.

L674 The description of the different order are interesting but I don't see the link with the results present in section 4. Could you specify how did you deduce those assessments?

The reviewer is right, we didn't include support for this conclusion. We added a table containing a geomorphological summary by orders, and also we expanded the discussion based on this Table. The new version of the document contains a text stating the following: Due to geomorphological characteristics (see Table 2), water tends to reach faster the channels in hills of order 1 and 2, and, at the same time, the sediment production and transport in these hills tend to be larger. Order 3 subbasins most likely act as transport elements, with no important energy losses (Table 2), and floods tend to occur at order 4 and 5 subbasins due to the widening of the channel and slope attenuation.

**Technical comments**

L121 when introducing the 3 models. Similarly to the short description of the hydrological model done between the lines 123 – 126, I suggest to keep the short description of both sub-model initially given in the first version of the manuscript.

We agree with the reviewer. We changed the text as follows: The methodology followed in this study includes a hydrological model [7, 3], a shallow landslides model and a flash flood model. The landslide sub-model follows the formulation described in [1]. The hydraulic sub-model corresponds to a low-cost 1D model (hereafter referred to as HydroFlash) that [assumes infinite sediment supply and] estimates the cross-sectional filled area at all time step.

L123, 137, 390 the sub-model simulated flood plain inundation is alternatively called : "hydraulic sub-model", "inundation sub-model", "flash flood submodel", and "HydroFlash". For a sake of consistency, and clarity, I suggest to choose only one of those terms and use it everywhere in the manuscript when calling that sub-model.

We agree with the reviewer. We have changed the document to just use "HydroFlash", except in the introduction, before calling the model HydroFlash.

- L161 please put bracket around "HAND" We included the brackets following the reviewer's suggestion.
- L190 By soil properties map, I was thinking to spatial distribution of the soil properties, as used to define the hydrological model. It would give an overview of the spatial distribution of the soil classes. I'm still thinking it is of interest. If you don't want to add another map, I would suggest just to adapt the color scale of the slope map to the soil classification. In that way, the reader could guess the spatial distribution and the proportion of each soil class. Another option, might be to add a column in table 1, with those proportions.

We added another column to Table 1 in which we include the total percentage of each soil type in the basin. We also included the spatial distribution map by adapting the Figure 2,

however, it is important to note that this is just an extrapolation of the soil properties based on the slope and it is not a detailed soil map. See the following table and Figure.

...

| Table 3: Description of the soils in the region [5]. |         |                    |           |              |            |  |  |  |  |
|------------------------------------------------------|---------|--------------------|-----------|--------------|------------|--|--|--|--|
| Type                                                 | Slope   | Depth $[m]$ | Retention | Permeability | Percentage |  |  |  |  |
| Class III                                            | < 12    | 0.6                | Low       | High         | 3.2        |  |  |  |  |
| Class IV                                             | 12 - 25 | 0.6                | Mean      | Mean         | 8.3        |  |  |  |  |
| Class VI                                             | 25 - 30 | 1.0                | Mean      | Mean         | 2.1        |  |  |  |  |
| Class VII                                            | 30 - 50 | 0.3                | Too Low   | Low          | 25.5       |  |  |  |  |
| Class VIII                                           | > 50    | 0.2                | Too Low   | Low          | 60.0       |  |  |  |  |

Por formation of the second se

L196 "Unfortunately [...]." I would suggest to remove this sentence for a sake of conciseness.

We removed the sentence following the reviewer's suggestion..

76.00°W

593

76.05°W

L211 the range of value of the stream discharge [185 - 222] m 3 .s -1 does not correspond to the range of value of the velocities [5 - 7] m.s -1 . A maximum discharge of 259 m 3 .s -1 is expected. Please change either the velocity of the discharge stream range.

The range 5-7 corresponds to observations for another watershed in the region with similar characteristics. However, our estimations were based on a range between 5 and 6, which gives the values of 185 and 222 respectively. We now indicate this explicitly in the document.

Soils map

L213 the sentence "The timing of peak flow is also [an] important information" can be removed.

We removed the phrase following the reviewer's suggestion.

L216 Please refer to the figure 16. Some details of how the contrast between both images are calculated would be welcomed.

We included the following explanation in the document: We empirically performed a detailed contrast between both products by using a geographic information system (QGIS), which provided us with information about flooded areas and landslide locations.

L230 the optimal distance for radar rainfall observation is described twice. Please remove "optimum" line 233 in "optimal optimum" and the sentence line 236 : "The results of the radar QPE methodology indicate that the rainfall estimation works well within a radius of 120 km."

We edited the document following the reviewer's suggestion.

L256 the sentence "Chocó is the [...]" can be removed.

We remove the word following the reviewer's suggestiond.

L296 add a space between "Figure" and 6.

We added a space following the reviewer's suggestion.

Table 3 are the capillarity and the gravitational storage really in mm? Or rather in cm?.

They are in mm. The values appear to be low, but they are consistent with the soils description. However,  $H_u$  varies between 27 and 137 mm.

Table 3 table 3: write "capillarity" instead of "capilary".

We changed the word following the reviewer's suggestion.

L358 write "10 parameters" instead of "ten parameters".

We changed the word following the reviewer's suggestion.

L363 please specify here the objective of the calibration. Here a suggestion: "The model simulation is calibrated to reach a base flow of 3 m 3 .s 1 . The calibration consists in scaling each physical parameter by a constant value in the entire basin (Francés et al. (2007b)). Table 3 includes the mean value for all the parameters used in the model and the scalar value adjusted during the model calibration."

We included the reviewer's suggestion. The manuscript now states: The model simulation is set to reach a base flow of 3 m3s-1, a value that corresponds to the discharge measurements during field campaigns days and weeks after the flash flood event and during dry spells. To set the soil wetness initial conditions realistically, the model runs start two days prior to Event 1. Before this period, there were only a couple of small rainfall events; for this reason, the overall wetness was set to represent to dry conditions at the start of the simulation. Table 3 includes the mean value for all of the parameters used in the model, and the scalar value is adjusted during the model calibration. In this implementation of the model, we left uncalibrated the channel speed, the subsurface speed, the aquifer losses, and the capillary and gravitational storages (see Table 3). The values for these parameters are inherited from a local watershed with similar characteristics. L367 the reference date is missing for "Aristizabal et al. ..."

We included the year following the reviewer's suggestion.

Eq 6 to 10 equation 6, 7, 8, 9, 10 : In the equation, the index "i" referring to the cell "i" of the catchment is specified in the left side of the equation but not in the right one. Please either specify the "i" index to any or no one of the parameter with cell dependent value.

We edited the equations in order to specify the index according to the revierwer's suggestion

Eq 16 the index "j" should be added to the parameter Fd,i which depends on it.

We added the "j" following the reviewer's suggestion.

L449 please remove the sentence "The model simulation is set up [...]". It refers to the method and therefore should be inserted in the section 3.

We removed the sentence following the reviewer's suggestion.

L470 the observed timing could be indicated in brackets, as a remember.

We followed the reviewer's suggestion.

L482 "particularly in the low end values", do you mean the discharge recession?.

The reviewer is right, the manuscript was not clear in this section, we change the text to: **particularly during the peak flow and early recession.**

L496 those lines correspond to the method description and should be inserted in section 3.2.

We moved the lines to the end of the **Hydrological modeling framework** subsection, which is inside the **Modeling** section.

L516 The first sentence "It is well known [...]" is not relevant and could be remove for a sake of conciseness.

We agree with the reviewer and we removed the phrase: "It is well known [...]".

Figure 2 unit of the slopes is [-] and not [mm. -1].

We corrected the error following the reviewer's suggestion.

Figure 5 plot a) y-axis: please confirm the rainfall unit : [mm / 5 min] ?.

The original units were in fact mm/5min, however that is an odd unit to present rainfall. In the new manuscript we changed the units to mm/hour.

All Within all the manuscript : choose between "sub-model" or "submodel" and maintain the same spelling.

We changed the monuscript to use the word submodel.

**Response to Reviewer 2 (Dr. Eric Gaume)**

May 28, 2019

Manuscript title: Reconstructing the Salgar 2015 Flash Flood Using Radar Retrievals and a Conceptual Modeling Framework: A Basis for a Better Flood Generating Mechanisms Discrimination

Authors: Nicolás Velásquez, Carlos D. Hoyos, Jaime I. Vélez, and Esneider Zapata

**General Concept:**

The manuscript has been substantially modified after the first reviews. This new version deserves nevertheless some major improvements before a publication in HESS can be considered to my point of view. An annotated manuscript is joined to this review. Six major points need especially improvement.

We sincerely thank Dr. Eric Gaume for his reviews and suggestions. We find most of them very useful, however, we disagree in some of the points he raised during this round of reviews. In particular, we would like to stress, as it is mentioned in the paper, that the information available for La Liboriana basin is limited, hence there could not be a perfect validation of the model. The conditions for La Liboriana basin, and the lack of data, are not unique of this region, but we must strive to develop strategies to provide risk management tools for these areas. We believe we do that in our work, based, as much as possible, on a conceptualization of the physical processes leading to the triggering of the flash flood. As we did during the first round of reviews, we have taken in serious considerations all the issues raised by Dr. Gaume, and have responded to his concerns both in the paragraphs below and within the manuscript. We feel the current manuscript is an improvement thanks to the reviewer comments. Below, the reviewer comments are in black and our comments are in "blue". Event though we do not agree 100% with Dr. Gaume, we acknowledge that his comments were constructive. In the cases we do not agree with the reviewer, we try to state it clearly.

1) The manuscript is based on a very limited amount of inaccurate observations: a single peak discharge, estimated at a cross-section located downstream the considered watershed, radar quantitative precipitation estimates from a radar located 90 km far away from the considered mountainous area with no ground raingauge or even amateur measurements confirming the estimated amounts and two aerial photographs for the location of landslides. This is very little to support the implementation, calibration and validation of a physically based distributed model.

This is perhaps the main difference we have with Dr. Gaume. We certainly agree that, in general, the information available to reconstruct La Liboriana flash flood is very limited. However, that is precisely the point. If there was enough information perhaps the point of the study would be very different. The goal of the study, which takes into account the fact that the information is scarce, is to test a low-cost methodology to reconstruct the main features of the Salgar flash flood. As mentioned before, the conditions for La Liboriana basin, and the lack of data, are not unique of this region, but we must strive to develop strategies to provide risk management tools for these areas. We believe we do that in our work, based, as much as possible, on a conceptualization of the physical processes

leading to the triggering of the flash flood, and the observational evidence available. Information from a C-band radar 90 km away is considered, in general, very good information, especially considering the high-resolution scanning strategy used at SIATA. In the original revision, we provided evidence of the quality of the QPE, and we provide more evidence of the spatial scale of the radar information in this revision.

Figure 1: In situ water stage sensor installation after the flash flood.

Additionally, we have included a new analysis of the model based on water stage data, using measurements from an in situ radar sensor (see Figure 1) installed at the same time as the rain gauges used to validate the QPE technique (the rain gauge information was included in the first review, and it is included in the work by Hoyos et al. (2019)). We include the rainfall information in this response, once again:

About rainfall estimates The radar QPE methodology uses rainfall gauges and disdrometers within the radar domain. The results indicate that the rainfall estimation works well within a radius of 120km (La Liboriana is located at 90km). Despite the distance between the radar and the basin, and the mountains between them, there are no blind spots in this region for the radar. Figure 2 shows a comparison between the radar QPE and two rain gauges stations installed 3 days after the event The correlation among the hourly precipitation records is 0.65.

---

## Author Response (AR3)

**Response to Reviewer 1**

August 26, 2019

**Manuscript title:** Reconstructing the Salgar 2015 Flash Flood Using Radar Retrievals and a Conceptual Modeling Framework: A Basis for a Better Flood Generating Mechanisms Discrimination

**Authors:** Nicolás Velásquez, Carlos D. Hoyos, Jaime I. Vélez, and Esneider Zapata

**General Concept:**

I already did the revision of the two first versions of manuscript. I really appreciate the supplementary information added in the manuscript, namely the figure 9, explaining the floodplain iterative assessment; the new discharge data, that comforts the hydrological model results, and the information concerning the stream velocity in Annexe.

However, I'm a bit upset that there are still a lot of annotation issues, and writing mistakes that might exist on the first submission but really not of the third one. Several co-authors comments are also spread within the text. Quick checks of the homogeneity and the consistency of the equations and annotations should be done before submitting and NOT by the reviewer. As example in the hydrological model, you mentioned calibrated parameters that are not in the equations of the model. Also the indexation of the variables/ parameters are still sporadic.

I'm convinced there is a lot of work behind the paper, and that each part of the modeling framework should have been a main topic on its own, making easier the writing, the results analysis and the reading. Nevertheless I respect the choice of the paper to gather the modeling work in one paper. In addition I would have personally chosena slightly different way to highlight the results and to orientate the discussion on several points. But again you're the only ones to choose the direction of your paper.

I divided my comment in three parts. The first one consists in several modifications of the section 3 to make clearer the methodology description, specially the hydrological model and the floodplain submodel description. Those comments and/or suggested modifications are very important and almost unavoidable (from my point of view) to clarify and make the method understandable. In the second part, I made some comments and suggestions on the results and on the discussion that should make the paper insights be highlighted. And finally in the last part, I pointed out the spelling or language mistakes I found.

We sincerely thank again the reviewer. We are truly impressed by her careful, and detailed review. We feel truly lucky having had her as a reviewer of our work. We would also like to apologize for the writing mistakes, typos and for the poor description of the hydrological model. In hindsight, part of the issue is due to the fact that we were able to integrate most of the comments by both reviewers in the new versions, improving the manuscript, but we failed to integrate some of them in a proper way, leading to organization issues in the manuscript. We think we managed to do that in the current version of the manuscript. Also, we have rewritten the model and submodel formulations in detail, following the suggestions by the reviewer.

We understand the point about having considered at the beginning writing this work as two separate manuscripts, one on the modeling framework and other on the Salgar event, but as the reviewer mentioned, we wanted to have the modeling framework highlighted by a real and relevant

application.

We have taken in consideration all the suggestions, and we have modified the manuscript accordingly.

**1 Comments on the sections**

**1.1 Comments on the methodology's description**

**Description of the hydrological model**

- **Page 13, line 303-305:** I don't understand this sentence "Vertical flows are only time dependent, while lateral flows could also depend on the actual state of the tank (kinematic approximation)." The vertical flows also depend on the actual state of the tank, doesn't it? I would suggest to remove this sentence.

  Yes, the reviewer is completely right. The vertical flows also depend on the state of the tank. The sentence was poorly worded. We have removed the sentence.

- **Page 13, lines 306-309:** You mentioned 4 modifications of the hydrological model but I don't agree with the fact to classify two of them as modification: The first one, "the direct use of radar QPE [...]": it is not a modification, but it is a specific choice of rainfall inputs. The 4th modification: "the development of two modules [...]": this refers to the "landslide submodel" and the "HydroFlash submodel". For my understanding, it is not a modification of the hydrological model, but additional modelling elements that use the results of the hydrological model. I'm sorry to be picky, but I think for an easy understanding of the paper, you should strictly follow the subsections's structure and only mention here what is related to the hydrological model.

  The reviewer is right. Allowing the direct use of QPE rainfall is a technical modification and not a hydrological one. Also, the addition of a landslide and a floodplain submodels do not directly modified the equations of the TETIS model so it is not an actual modification of the model but of the modeling framework. We have modified the text following the summary suggestion by the reviewer.

- **Page 13, 311-317:** I prefer when this part was inserted in the results description. The cell classification is a tool to analyze the spatially heterogeneous response of the catchment. I would even suggest to merge the figure 7 and the figure 14, keeping only the 50 classes categorizations of the figure 7 used in the Figure 14. Moreover I don't understand the first sentence and the expression "soil-rainfall-discharge coupling holistically". Another option would be to introduce this in the same section as the virtual tracers.

  We agree with the reviewer and we accept the suggestion. We merged again Figures 7 and 14, including only the example of the 50 groups. Also, in the description we modified the sentence **soil-rainfall-discharge coupling holistically** to:**Additionally, we propose a graphical method to analyze, at the same time, the evolution of multiple hydrological variables in the entire basin**. The description of the graphical method is included in the **Tools for spatial analysis of the results: virtual tracers and catchment cell grouping** subsection, and the figure and its description are included in the **Results** section.

- **Page 14, figure 6:** in the Hydrological modelling panel, the storage tanks are called "Ti", but the "Si" parameters are not mentioned as the legend does.

  The reviewer is right; it was our mistake. The name of the variable in Figure 6 and in the equation is $S_i$ and not $T_i$. We modified Figure 6 in order to match the equation.

- **Page 14, title section 3.1.1:** the modelling modifications are not only related to runoff but to all the lateral flow; I would suggest to call this section "Lateral flow modelling modifications"

We agree with the reviewer and have changed the title of the section to: **Lateral flow modeling modifications**.

- **Page 15 − 16:** the variable $A_i(t)$, called in the text "sectional area of the storage", has actually no unit [-], according to the equation (6). Reading Velez's thesis, it seems rather to be a coefficient. The actual sectional area is $S_i(t) * \Delta x$. This error makes the understanding of the equation really complex, and even makes me doubt about the meaning of the equation.

The description of the model was rewritten considering the reviewers suggestions and concerns. The text of the subsection is now written as follows:

The TETIS model relies on the concept of mass balance where the storage of tank $i$ at the end of the simulation interval $S_i(t)^*$ [mm] is function of the storage at the start of the simulation interval $S_i(t)$ [mm] and the storage outflow $E_i(t)$ [mm] during the interval $t$, as follows:

$$S_i(t)^* = S_i(t) - E_i(t) \tag{1}$$

The storage outflow $E_i$ is estimated by transforming the storage $S_i(t)$ into an equivalent cross sectional area $A_i$ [m²], as follows:

$$A_i(t) = S_i(t)F_c/\Delta x, \tag{2}$$

where $\Delta x$ [m] is the model cell width, and $F_c$ [m³ mm⁻¹] is a units conversion factor that is equal to the area of each cell element $A_e$ [m²] multiplied by 1 m/1000 mm. According to [34], $E_i$ changes as a function of $A_i$, the flow speed $v_i$ [ms⁻¹], and the model time step $\Delta t$ [s], as follows:

$$E_i(t) = A_i(t)^* v_i(t)\Delta t/F_c. \tag{3}$$

The expression for the cross sectional area at the end of the simulation period $A_i(t)^*$ is found by replacing $S_i(t)$ in equation (2) for $S_i(t)^*$, and then resulting expression and equation (3) into equation (1),

$$A_i(t)^* = \frac{S_i(t)F_c}{\Delta x + v_i(t)\Delta t}. \tag{4}$$

Equation (4) is solved coupled with the equation for the speed $v_i$:

$$v_i(t) = \beta A_i(t)^\alpha \tag{5}$$

Equation 5 is the generic formulation for the speed used in this work to represent nonlinearities in the relationship between $v_i$ and $A_i$. In the formulation, both, $\beta$ and $\alpha$ change depending on the type of flow: overland, subsurface, base, and channel flow. The solution for $v_i$ is obtained by using the successive substitution method described by [6]. In the model, we use a 5-minute time step which ensures the stability of the computations. When a solution is reached, $E_i$ is computed using equation (3) and $S_i$ is updated using equation (1).

Nonlinear equations in lateral flows result in a better representation of processes at high resolutions [4, 19]. A nonlinear approximation of runoff is presented in equation 6. This approximation is a modification of Manning's formula for flow in gullies. According to [12], $\varepsilon$ and $e_1$ are a coefficient and an exponent used to translate the Manning channel concept into multiple small channels or gullies. The values of $\varepsilon$ and $e_1$ are 0.5 and 0.64, respectively [12]. $A_{i,2}$ [m²] is the corresponding sectional area obtained from $S_{i,2}$ by using equation (4). In addition, $M_{i,0}$ is the slope of the cell, and $n_i$ is the Manning coefficient.

$$v_{i,2} = C_7 \frac{\varepsilon}{n} M_{i,0}^{1/2} A_{i,2}(t)^{(2/3)e_1} \tag{6}$$

The nonlinear equation 7 corresponds to an adaptation of the [21] formula for subsurface runoff $v_{i,4}$, where $k_{i,s}$ is the saturated hydraulic conductivity of cell $i$, and the exponent $b$ is dependent on the soil type, and it is assumed to be equal to 2. $A_{i,g}$ is the equivalent cross-section area of the maximum gravitational storage ($H_{i,g}$ [mm]). $A_{i,3}$ is the corresponding sectional area for the gravitational storage ($S_{i,3}$) obtained by using equation (4). There is also return flow from tank 3 to tank 2, when $S_{i,3} = H_{i,g}$, which represents runoff generation by saturation. In the case of the base-flow, we assume that the speed $v_{i,4}$ is constant for each cell and depends on the aquifer hydraulic conductivity $k_{i,p}$ (see equation 8).

$$v_{i,3} = C_8 \frac{k_{i,s} M_{i,0}^2}{(b+1) A_{i,g}^b} A_{i,3}(t)^b \tag{7}$$

$$v_{i,4} = C_9 k_{i,p} \tag{8}$$

Finally, the stream flow velocity is calculated by using the geomorphological kinematic wave approximation [34? ], in which $\Lambda$ [km$^2$] represents the upstream area, and $\Omega$ and $\omega_i$, a regional coefficient and regional exponents, respectively

$$v_{i,5} = C_{10} \Omega M_{i,0}^{\omega_1} \Lambda_i^{\omega_2} A_{i,5}^{\omega_3} \tag{9}$$

The streamflow speed expression is a version of equation (5). This considering that the terms $\Omega$, $M_{i,0}^{\omega_1}$, $\Lambda^{\omega_2}$, and the exponent $\omega_3$ are constant with time.

- **Page 15:** definition of vi: I'm not familiar with the dimensionless variable Ai you defined. I used to use relationship between velocity and hydraulic radius or storage water level.

  See response to previous comment. The variable $A_i$ [m$^2$] refers to the equivalent equivalent cross sectional area of the tank $i$, as follows

- **Page 15-16, equations 1- 8:** I would suggest to present the general equations that control all the lateral flow first (eq. 7, 6, 1), before indicating the particularity of each tank lateral flow (eq. 2-5).

  We agree with the reviewer. We have rewritten the section as shown in the responses to the previous two comments.

- **Page 15, line 334:** please call the slope in a different way, it might be confused with the tank levels $Si$.

  We modified the variable following the suggestion by the reviewer. The slope is now referred to as $M_{i,0}$.

- **Page16, equation 5:** how we are suppose to understand the equation? There is more exponents than parameters.. please also simplify, specially if at the end you will use a regional parameter deduce from any catchment.. Here what is important is the fact that $v_5$ is depending not only on $A_5$ but also on the stream bed slope. Only 3 parameters should appear : $v5 = \beta * S_i(t)^\alpha S_0^\lambda$.

  We thank the reviewer. There was a typo in equation. We have now corrected it. The correct form of the equation is: $v_{i,5} = C_{10} \Omega M_{i,0}^{\omega_1} \Lambda_i^{\omega_2} A_{i,5}^{\omega_3}$. In the formulation made by [34], the equation depends the slope $M_{i,0}$, the upstream area $\Lambda_i$ [km$^2$], and the equivalent sectional area $A_{i,5}$ [m$^2$]. The parameters correspond the the coefficient $\Omega$ and the exponents $\omega_1$, $\omega_2$, and $\omega_3$. According to [34], the regional variables are constant for all the watershed. For each channel element, the equation can be rewritten by grouping in a single coefficient the terms $\beta_c$ the

constant parameters $\Omega$, $M_{i,0}^{\omega_1}$, and $\Lambda^{\omega_2}$. This results in an equation that has the same form as the generic equation $v_{i,5} = \beta A_{i,5}(t)^{\omega_3}$. An extended discussion of the regional parameters can be found in [34].

- **Page 16, page 366-367:** $\Delta x$ versus $L$: Actually according to Velez, 2011 page 89; the $\Delta X$ variable used his thesis correspond to the cell width; i.e. the resolution if the flow direction is orthogonal or $\Delta x\sqrt{2}$ if the flow direction is diagonal.

  The reviewer is correct, in the model $L = \Delta X$ if the flow direction is orthogonal, and $L = \Delta x\sqrt{2}$ if the direction is diagonal. The text and the equations have been modified accordingly.

- **To summary the remarks I did from page 12, line 290 to page 16, line 372, I would suggest to reword more or less as follow:**

  We used a physically-based and distributed model developed and described in Vélez (2001) and Frances et al. (2007). The spatial distribution and the hydrological flow path schema is based on the 12.75 m resolution DEM data. In each cell, five tanks represent the hydrological processes including capillary (tank 1), gravitational (tank 2), runoff (tank 3), baseflow (tank 4) and channel storage (tank 5). The state of each tank varies as a function of vertical and lateral flows as shown in the diagram, where the storage is represented by Si [m] and the vertical input to each tank by Di [m], which in turns depends on the vertical flow through tanks Ri [m]. Ei represents the downstream connection between cells, except for tank 1, where E1 represents the evaporation rate.

  The original model fully described in Vélez (2001) and Frances et al. (2007) are modified to improve the representation of the flow processes that occur during flash floods (see section 3.1.1). In addition, two analysis tools of the hydrological modelling results are introduced: virtual tracers tracking precipitation origins as well as water paths over or through the soils; and catchment cell grouping (see section 3.1.2). The tools objective is to allow an analysis of the spatially distributed response of the catchment.

  We thank the reviewer for the careful review of the model equations. We have modified the text and the model formulation, being careful in the equation typewriting, the coherence between the text, the variables and the equations, the completeness of the model description, and considering all the suggestions made by the reviewer.

- **Page 17, line 396:** hydrological and not hydrologic.

  We corrected the word in the entire document.

- **pages 17, line $410-412$:** I would suggest to speak about the calibrated parameters rather than the non calibrated parameters.

  We have change the text to list the parameters that were calibrated.

- **page 18, table 3:** the parameters you mentioned in the table are not in the related equations.

  We have modified Table 3 to explicitly relate all the parameters to the model equations. Also, the caption of the table was modified to describe the parameters not included in the model formulation.

- **page 18, table 3:** Assuming the velocity parameters correspond of the velocity of each flow when the related water storage is equal to 1; I would expect the increasing magnitude order: Subterranean speed, subsurface speed, surface speed, channel speed. How can you explain that the subsurface speed is higher than all the other ones?

  We believe that this might be a misunderstanding caused by the original description and setup of Table 3. In the new Table 3, it can be seen that the mentioned value corresponds to an scalar coefficient that multiplies the physical value of the velocities, rather than the actual physical value. In other words, the constants $C_i$ corresponds to calibration coefficients: $v_i = C_i B_i A_i^\alpha$,

where $C_i$ is the calibrating parameter, $B_i$ the physical factor, $A_i$ the corresponding sectional area, and $\alpha$ the exponent. To avoid misunderstandings, we deleted the **mean value** column.

On a side note, the reviewer is correct, in the model the mean speed of each tank increase from the subterranean storage to the runoff.

- **page 18, line 417:** please remove "above the slip surface Z i,w ".

We appreciate the reviewer comment. We re-phrase the corresponding paragraph to: The landslide submodel coupled to the TETIS model is proposed by [1]. The stability of each cell is calculated through the assessment of the different stresses applied to the soil matrix. The coupling between TETIS and the landslide submodel is required because the stability of the soil decreases with the pore water pressure [17]. The saturated soil depth $Z_{i,w}$ depends on the gravitational storage $S_{i,3}(t)$, the soil wilting point $W_{i,pwp}$, and the soil field capacity $W_{i,fc}$, as follows:

$$Z_{i,w}(t) = \frac{S_{i,3}(t)}{W_{i,cfc} - W_{i,pmp}} \tag{10}$$

- **page 19, line 441:** I suggest the following title "the floodplain submodel Hydroflash".

Following the reviewer comment, we changed the title to: Floodplain submodel (Hydroflash)

- **page 21, line 477:** $A_{i,sed}$ meaning: Is really the flooded area (area along x,y) or the sectional area along the cross profile (area along z,y, x being the stream flow direction)? According to the attributed name, it seems to be the first definition; but according to the figure 9, I would say the second definition. It makes a big difference...

We agree with the reviewer, the text in its original form leads to misunderstandings. We have corrected the document, and, in addition, we now refer to $A_{i,load}$, as suggested by the reviewer.

- **page 19-20:** Hereafter I'll suggest some rewording, introducing ALL the annotations. The equation references has to be added. It is roughly drafted. Please, feel free to integrate or not.

[...]

We appreciate the reviewer's advice about the floodplain submodel description. In the new version of the document, we rewrote following all the comments by the reviewer, as follows:

The HydroFlash submodel is designed to interpret the TETIS simulations as floodplain inundations (Figure 8). For each stream cell and at each time step, the submodel (i) calculates the stream discharge including sediment load (equations 11 - 16, see [33]), and (ii) determines the inundated cells according to the stream cross-profile, the sectional area, and the stream velocities when including the sediment load (equations 15 - 17, [33]). To determine the discharge including sediment load ($Q_{i,load}$), a realistic channel width is calculated according to [22] approach as

$$W_i = 3.26Q_i^{-0.469} \tag{11}$$

where $Q_i$ corresponds to the streamflow estimated based on a long-term water balance.

Assuming an infinite sediment and ruble supply, equations 12, 13, 14 are used to deduce, from the channel width $W_i$, the water level $Y_i$ (equation (12)), the friction velocity $v_{i,fr}$ (equation (12, described in [33]), the sediment concentration $c_i$ (equation (14)), and finally the sediment-loaded stream discharge (equation (16)), as follows:

$$Y_i(t) = \frac{Q_{i,sim}(t)}{v_{i,sim}(t)W_i} \tag{12}$$

$$v_{i,fr}(t) = \frac{v_{i,sim}(t)}{5.75 log\left(\frac{Y_i(t)}{D_{i,50}}\right) + 6.25} \tag{13}$$

$$c_i(t) = C_{max}(0.06Y_i(t))^{\frac{0.2}{v_{i,fr}(t)}} \tag{14}$$

$$r_i(t) = \frac{1}{D_{i,50}} \left[\frac{g}{0.0128}\left(c_i + (1 - c_i)\frac{\gamma_w}{\gamma_{sed}}\right)\right]^{1/2} \cdot \left[\left(\frac{C_{max}}{c_i}\right)^{1/3} - 1\right] \tag{15}$$

$$Q_{i,load}(t) = \frac{Q_{i,sim}(t)}{1 - c_i(t)} \tag{16}$$

where $v_{i,sim}$ and $Q_{i,sim}$ are the simulated velocity and streamflow, respectively. Also, $r_i$ is the constitutive coefficient of the flow, that summarizes the flow dynamics associated with sediments and colliding particles. The above mentioned relationships depend of 2 parameters: the maximum sediment concentration ($C_{max}$ [-]) and the characteristic diameter of the sediments $D_{i,50}$ [m]. Both terms are assumed to be constant and equal to 0.75 [26] and 0.138 [16], respectively.

To determine the inundated cells, the flood depth ($F_{i,d}$) and the sectional area of the stream including sediments ($A_{i,load}$) are iteratively calculated by reducing the difference between $Q_{i,load}$ and $\hat{Q}_{i,load}$. The channel cross-section for cell $i$, $E_{i,bed}$, is defined by the DEM. In each iteration $N$, the model updates $F_{i,d}$ with a $\Delta y = 0.1$ m increase. The cross sectional area $A_{i,load}$ is calculated by taking difference between $F_{i,d}$ and the elevation of each cell $j$ in the cross-section $E_{i,bed}$.

$$\hat{Q}_{i,load}(t) = 0.2r_i(t)(N\Delta y)^{\frac{3}{2}} S_{i,0} A_{i,load}(t) \tag{17}$$

$$F_{d,i}^N = F_{d,i}^{N-1} + \Delta y \tag{18}$$

$$A_{i,load}^N = \Delta x \sum_{j=1}^{N} F_{i,j,d}^N - E_{i,j,bed} with E_{i,j,bed} < F_{i,j,d}^N \tag{19}$$

The resulting flood maps might include the presence of small isolated flood spots and discontinuities where the flow direction changes from orthogonal to diagonal across or vice versa. We included two post-processing steps to correct these issues by (i) using an image processing erosion algorithm [32] to remove the small and isolated flood spots (step 4 in Figure 8), and, to solve the flow direction discontinuities, (ii) for each flooded cell the model seeks to inundate the eight neighboring cells: A neighbor cell is also flooded if the altitude of the original flooded cell, plus the flood depth, is higher than its elevation (step 5 in Figure 8). The image erosion is performed once with a 3 by 3 kernel. An example of the final result for a time step $t$ is shown in the step 6 in Figure 8.

- **Page 19-20** about the HydroFlash model: I'm curious to know about the ratio Qsed/Qsim: what is the range of value of c? Is there a significant change to include the sediments when calculating the floodplain?.

The simulated peak discharge $Q_{sim}$ with the final parameterization used in the study is 220 m³s⁻¹, and the streamflow with the sediment load reached values around 285 m³s⁻¹, for a $Q_{sed}/Q_{sim}$ ratio of 1.3. The extra 30% discharge is certainly a relevant contribution with impacts in the floodplain simulations. We added this comment in the discussion section, as follows.

Similarly, results of the HydroFlash submodel are satisfactory despite the hydraulic over simplifications, and are potentially useful for issuing warnings to the community. From that point

of view, it is important to stress that the low computing cost of HydroFlash, different to that of detailed 2D/3D hydraulic geomorphological models, makes it possible to be executed in real time coupled with rainfall observations, providing valuable information that, while not 100% accurate spatially, helps discriminating to a high degree, for example, which communities need to be evacuated given an extreme event. In addition, the floodplain submodel provides an indirect estimation of the sediment load during extreme events. In the 2015 Salgar simulations, the peak discharge obtained with the hydrological model was 220 m³s⁻¹; the total streamflow considering the sediment load reached values around 285 m³s⁻¹, for a $Q_{sed}/Q_{sim}$ ratio of 1.3. The extra 30% discharge corresponding to the sediment load is certainly a relevant contribution to the total discharge, with impacts in the floodplain determination. Considering the stream network slope, the simulated ratio is comparable with reports in the literature [e.g. 28]. The sediment load is mainly constrained by the maximum sediment concentration $C_{max}$ and the depth of the flow, suggesting that better information about $C_{max}$ could improve the simulation of flood spots.

**1.2 Comments on the results and discussion**

- **page 21-22, section 4.1:** On the one hand, the model simulated a flood peak in the upper range of the discharge peak assessment and the simulated flood peak occurred 20 minutes earlier than the observed one. On the other hand, when doing the sensitivity analysis on the surface speed parameter, decreasing the runoff velocity, the simulated flood peak is diminished and occurs later. Why didn't you calibrate better the surface speed, as the model is sensitive?

  There are a couple of points to consider. First, as mentioned in the manuscript, some reports suggest the peak flow reached the urban area after 2:10 am, and others around 2:40 am. Second, from the point of view of risk management, and having evidence of likely faster speeds in the channel, we wanted to be conservative, and decided to gravitate towards obtaining a discharge in the high end of the interval. And third, after evaluation, the combination of parameters was the best to reproduce not only the 2015 Salgar flooding event, but also the other peak flow events registered during 2015.

- **Page 22, line 506-511:** I think those results are insights of the paper. They should be discussed in the discussion part to confront them to the literature (if there is) and to highlight them.

  We agree with the reviewer. We expanded the discussion as follows:

  [...] The overall evidence suggests that precedent capillary moisture in the basin plays an essential role in modulating river discharge. This behavior could be linked to the temporal occurrence and relative importance and timing of stratiform and convective formations previously described. During the extreme event, when the soils were already wet, the convective rainfall fraction dominated the hydrograph formation. While stratiform rainfall plays an important role moistening the entire basin, convective rainfall generates considerable runoff, leading to flash flooding. Several authors have argued about the role of convective rainfall triggering flash floods [9, 18, 31, 7, 30, 25, 29, 13, 3, 15, 5, 27, 10, 23, 2], however, to our knowledge no other study has tracked convective and stratiform water in a modeling setting to explore their relative role leading to flash flooding.

- **Page 25, figure 13:** the specific flood peaks are interesting. The simulated values are below the envelop $Qpeak = 97A^{-0.4}$ that make the simulated flood peak consistent with the literature on flash flood (Gaume et al. 2009). You should mention it on the discussion to strengthen the flood peaks simulation consistency.

  We thank the reviewer for pointing this out. We included this comment in the discussion section:

The methodology implies changes and additions to the TETIS distributed hydrological model including tracking independently convective and stratiform precipitation within the model, as well as keeping track of the runoff and subsurface portions of the streamflow. TETIS was coupled with a shallow landslide submodel and HydroFlash, a one-dimensional floodplain scheme. The model proposed here indeed allows studying the different hydrological processes relevant to flash flood and landslide occurrence by using different simulation resources, serving as the basis for a better understanding of the overall basin response. **Despite the lack of data, the evidence suggest that the results represents, to a large degree, the magnitude of the disaster; considering also that the simulated peak flow is consistent with the peak flow envelope proposed by [14] for flash floods**. This approach helps to examine the first-order flood-generating mechanisms or causative factors both in time and in space, focusing on the most important physical processes [20, 24], potentially allowing the anticipation of flash flooding events, the issue of warnings, and response by risk management entities.

- **Page 27, line 595-600:** I think my previous comment was misunderstood. I think you were right saying: "In event 2, the convective rainfall and the runoff show a similar evolution, denoting a strong influence of the convective portion (figure 12b)". But I think there were one unmentioned condition to observed similar evolution. The similar evolution comes from the fact that the convective portion is totally controlling the runoff processes AND that there is no effect of the stream network to modify or temporize the runoff advent at the outlet. In other words, it's possible to get strong influence of the convective rainfall runoff without having similar evolution, if the stream network buffers the runoff advent.

We agree with the reviewer. We changed the paragraph to: According to Figure ??b for Event 2, the accumulations of streamflow runoff and convective rainfall become similar with the increase in time. This fact highlights the strong control that, in this case, the convective portion has on the runoff, with almost no effect of the stream network filtering out the convective signal, most likely due to the size and the rapid response of the basin. This description, however, only applies for the runoff portion, since the evolution is different when we consider the total simulated streamflow.

- **Page 27, line 615-617** about the soil depth definition: You justified here the scaling factor by adjusting an underestimated soil depth observation. But then it means that the soil depth definition previously chosen for your hydrological model are also underestimated. I would rather assume that you need to calibrate the model to make the landslide occurring. The scaling factor might explained as to be an artifact of a too simplistic model, and a non calibration of the other parameters.

We partially agree with the reviewer. However, results of the landslide model in other regions show good agreement with observations [1]. In our case we only have a poor and a general description of the watershed soils, that does not allow us to obtain distributed soil maps. This includes soils properties relevant for the model that we did not include. We added a comment about this in the discussion section. See the answer to the last issue raised by the reviewer.

- **Page 29, line 631-632:** This comment should appear in the end of the discussion or in the conclusion, not in the result section.

We agree with the reviewer. We moved the comment to the conclusions section:

Results of the landslide submodel and HydroFlash, while satisfactory, are far from perfect, showing significant differences compared to observations. The evidence suggests, by and large, that most of the observed differences are mainly due to the lack of higher spatial resolution DEM, in the case of HydroFlash, and due to the lack of a detailed soil dataset, in the case of the landslide submodel. However, there is also is considerable room for improvement in both submodels, including a better representation of non-Newtonian hydraulic processes in HydroFlash, and a direct link between landslides and flood spots following, for example, a

similar strategy to the one presented in the STEP-TRAMM model [11]. Notwithstanding the difficulties, the results suggest that the submodel simulations could have been used and should be used in the future for early detection and warning to improve both short- and long-term risk reduction strategies.

- **Page 31, line 669-670:** I would refer to Zocatelli et al (2011) as following "Zocatelli et al (2011) found similar results in . . . (where, and which size of catchement)." As you wrote, it seems that Zocatelli et al (2011) found your own results.

  We modified the paragraph following the suggesting by the reviewer:

  The evolution of the simulation of Events 1 and 2 show evidence of remarkable behavioral differences. During Event 1, both gravitational and capillary tanks are filled along and across the basin as a result of the quasi-homogeneous rainfall spatial distribution. [36] found similar results for watersheds in Europe with areas ranging between 982 and 52 $km^2$. The return flow is low, and most of the runoff occurs within the first 20 groups (40% of the watershed closest to the outlet). In the period between both events, there is a recession in the capillary and gravitational storages in the entire basin. Capillary storage decays considerably slower than gravitational storage. During Event 2, the flash flood triggering event, the first convective core saturates both capillary and gravitational storages in the upper part of the basin and generates both return flow and significant runoff. Due to soil saturation, the second convective core results mainly in surface runoff. During this event, extreme runoff rates are evident in the upper part of the basin, collocated with the steeper slopes. On the other hand, subsurface flow is more important in magnitude than runoff describing Event 1, while runoff is more relevant for Event 2. The precedent storage and the presence of thunderstorm training profoundly condition the streamflow during Event 2. The overall evidence suggests that precedent capillary moisture in the basin plays an essential role in modulating river discharge. This behavior could be linked to the temporal occurrence and relative importance and timing of stratiform and convective formations previously described.

- **Discussion:** As said before, it would be nice if the results of the landslide model and of the floodplain model are discussed. Here some ideas for the landslide model. The facts:

  - your model relates landslides to soil depths, soil water content and topography. the soil depth spatial distribution is roughly done according to the topography.

  - Landslide occurring is therefore only related to soil filling and the combined 'topographical-soil depth properties'.

  - 1) Crossing topographical map and false simulated landslides location, those latter ones seem to appear where there are slope greater than 2. Is 20 cm soil depth on a 2 slope realistic? Or could it explain the false simulated landslides?.

  - 2) The observed landslide is observed where the amount but all the intensities of the rainfall are the highest. Could the rainfall intensity have an impact on landslide and explain why the model is failing (as not taking into account).

    We agree with the reviewer, and include the next paragraph in the document:

    The landslides submodel presents an overall acceptable performance with limitations in certain regions. In particular, there are some false positives in the middle of the basin. These limitations could be associated with the assumptions and approximations inherent to the submodel, including that it only determines unstable cells by slowly filling the soil matrix with water, which, in this case, given the lack of information, depends on the soil depth derived from the topography, and that the model does not consider instability due to intense rainfall events. The lack of detailed soil depth information could explain the false positives landslides. On the other hand, the relation between landslides and high-intensity rainfall must be explored and included in this kind of models. There is also an apparent contradiction regarding the depth of the soils in the basin: While the values

derived from topography appear to work well for the hydrological model, the depth had to be calibrated to obtained a better representation of landslides. There are two possible explanations for the contradiction, (i) that the soils are in fact thicker in the entire basin, but the calibration of the infiltration and percolation rates corrected the hydrological simulations, and (ii) that the landslides submodel is too simplistic, or that no other parameters were calibrated, possibly resulting in over calibration of the soil depth. This is an aspect that needs to be improved further.

The landslide submodel has been used in a nearby watershed with similar characteristics, but with high-quality distributed information [1]. In that case, the model shows a better performance, which highlights the relevance of the quality of the input data. It is also important to consider that, a pinpoint localization of the unstable cells is still considered a hard task, in part due to the small temporal and spatial scale at which landslide processes take place [1, 8, 35].

**2 Technical comments**

- **Page 5, line 120:** As said before, keep the same name to call the different submodels of your modelling framework: I would suggest to use 'hydrological model' and even use it name 'Tetis' (Velez et al, 2002), for the first modelling part; 'landslide submodel' for the second modelling part and 'HydroFlash floodplain submodel' for the third modelling part.

  We accept the suggestion by the reviewer; we now refer by **TETIS** to the hydrological model, and we use the word submodel throughout the document to refer to the **landslide submodel** and to the **HydroFlash floodplain submodel**.

- **Page 5, line 122:** choose to totally insert or remove "assumes infinite sediment suppply".

  We opted for inserting the comment in the sentence without the brackets.

- **Page 5, line 123:** 'hydrological' and not 'hydrologic'.

  We modified the term throughout the entire document.

- **Page 5, line 122-129:** please put the small description in the order it appears in the text: first the hydrological model, second the landslide submodel, third the HydroFlash floodplain submodel.

  We have rewritten paragraph as follows:

  **The methodology followed in this study is based on a modeling framework using the TETIS hydrological model [34? ], modified to include a shallow landslide submodel, and a floodplain submodel termed *HydroFlash*. The TETIS model is a cell-distributed conceptual hydrological model that uses storage tanks and the kinematic wave approximation to simulate the most relevant processes in the basin. The landslides submodel is a stability model that classifies cells into unconditionally-stable, unconditionally-unstable, and conditionally stable depending on geomorphology; conditionally stable cells are further classified as stable or unstable based in their variable water content [1]. HydroFlash corresponds to a low-cost 1D model that assumes infinite sediment supply and estimates the cross-sectional filled area at all time steps based on the liquid discharge and the sediment transport. In addition, the TETIS model was modified to include four virtual tracers to separately explore the role of runoff and subsurface flow, as well as the relative importance of convective and stratiform precipitation in flash flood generation. The assessment of the interactions between runoff, subsurface flow, and convective-stratiform rainfall allows a better understanding of the short-term hydrological mechanisms leading to the flash flood event.**

- **Page 6, line 149-153:** From my point of view, I would remove those sentences from this section. The aim of the section is to describe the catchment, not to come back on the objectives of the study. If you want to emphasize the challenge to work with scarce physiographical information, you should mention within the introduction for example when speaking to ungauge catchment (end of line 112 for example).

  We completely agree with the reviewer. The comment was not in the original versions of the manuscript, and we added it trying to satisfy some comments by the other reviewer, but we did not do it properly. We have removed the mentioned lines and we stress the challenges in different places of the document.

- **Page 6, line 166:** by brackets, I would say ().

  We corrected the issue throughout the entire document.

- **Page 8, figure 3:** please add the zoom number on the first top window.

  We have modified the figure labels and its caption. The caption is now the following: **Aerial overview of La Liboriana basin (source: Department of Antioquia). The top-right panel presents the entire basin, showing the location of key regions detailed in the following panels, in zooms 1 to 5. The stream network is also presented, colored by order, from yellow to deep blue corresponding to orders 1 to 5.**

- **Page 8, line 195** and somewhere else in the manuscript: unity should not be in italic font.

  We made sure to correct all the units in italic to regular font style throughout the document.

- **Page 10, line 241-242:** remove the sentence "the results of the radar [...]" as the same information is done in the sentence line 238-239.

  We have removed the sentence, that was indeed redundant.

- **Page 12, title of the section 3.1:** I would suggest to choose "the hydrological model Tetis" as there is only the description of the model in this section (and not the 2 linked submodels). Or do you consider the that the framework consists in the the model plus the analysing tools (tracers and catchment cells grouping)?.

  We have decided to follow the reviewer suggestion. The title of the subsection is now: **TETIS hydrological model**. In general, we do consider that the framework includes the TETIS model plus the submodels and the analysis tools, but for the methodology section is less confusing to just refer to the model in the title of the subsection.

- **Page 22, line 510:** please remove "On the other hand".

  We have removed it following the suggestion by the reviewer.

- **Page 22, line 517:** 'According to the model simulations, the peak flow occurred at approximately 2.20am LT': Why did you say "approximately" ? You have a solely simulation, that should give exactly one flood peak time.

  The reviewer is correct. We have one model simulation and one value for the flood peak time. We have deleted the word **approximately** from the sentence.

- **Page 22, line 521:** when describing the figure: To make easier the manuscript reading, you should mention the studied parameters in the same order they appear in the figure: parameter of the top panel, parameter of the center panel, parameter of the bottom panel.

  We agree with the comment. We have modified both, the paragraph and the Figure caption as follows:

  Figure 10 shows the results of a sensitivity analysis of the hydrological simulation during the second rainfall event, varying the surface speed, infiltration rate, and the subsurface speed

factors. The aim of the sensitivity analysis is to evaluate the robustness of the overall results, considering the fact that the quality and quantity of some of the watershed information is limited. In the sensitivity analysis, we vary the surface speed factor between 0.01 and 20, the infiltration factor between 0.02 and 20, and the subsurface speed factor between 0.1 and 10. The overall sensitivity results show that the main findings described in the previous paragraphs are, in fact, robust to almost all changes in the mentioned parameters, with the surface runoff associated with convective rainfall controlling the magnitude of the peak discharge during the Event 2. The model's highest sensitivity, and hence the largest uncertainty source, appears to be related to the surface speed parameter (Figure 10a), particularly during the peak flow and the early recession. On the other hand, changes in the infiltration rate factor (Figure 10b) and subsurface velocity factor (Figure 10c) are associated with with a simulation sensitivity smaller than 7 and 20% of the peak flow, respectively.

Figure caption: **Hydrological simulation sensitivity analysis. Similarly as in Figure 9, all panels show the simulated streamflow (purple), and the runoff (green) and subsurface flow (dashed purple) separation. From top to bottom, the panels show the simulation sensitivity to changes in the a) surface speed, b) infiltration rate, and c) subsurface speed factors.**

- **Page 23, line 534:** please define the acronym 'SIATA'.

  The acronym is defined in the **Rainfall information** subsection. SIATA stands for *Sistema de Alerta Temprana de Medellín y el Valle de Aburrá*, which translates to Early Warning System of Medellin and the Aburrá Valley.

- **Page 23, line 545:** write 'skillfully' and 'skillfylly'.

  We have corrected the typo.

- **Page 24, figure 11:** write 'top', 'center' and 'bottom' panels instead of 'left', 'middle' and 'right' panels.

  We modified and corrected the caption of the figure and it now refers to the top, middle and bottom panels.

- **Page 27, line 617 and table 4, page 19:** The scaling parameter for the soil depth is not the same within the text and in the table.

  The reviewer is right, there was an error in Table 4. We have changed the scalar parameter value in Table 4 to 3.5.

- **Page 28, line 622-624:** I would remove those two sentences.

  We removed the sentences as suggested by the reviewer. We modified the paragraph and some of the material was moved to the discussion section.

- **Page 31, line 656:** "abilities" or "capacities" instead of "capabilities".

  We rephrased the sentence to avoid the word "capabilities".

- **Page 32, line 691:** Event 1 and 2 as you already choose the brackets to distinguish between convective (stratiform) events.

  The reviewer is correct; we changed the sentence to: **During Events 1 and 2, convective (stratiform) average accumulations are 28 (23) and 17 (14) mm, respectively**.

[revised manuscript text omitted]

---

## Author Response (AR4)

**Response to Reviewer**

**December 2019**

**Manuscript title:** Reconstructing the 2015 Salgar Flash Flood Using Radar Retrievals and a Conceptual Modeling Framework in an Ungauged Basin

**Authors:** Nicolás Velásquez, Carlos D. Hoyos, Jaime I. Vélez, and Esneider Zapata

**General Concept:**

I already did the revision of the three first versions of manuscript. I really appreciated the clarification of the method as well as the new points supplied into the discussion. As the limitations of the results (related to the applied method and the data availability) are now clarified, the present manuscript gained in reliability. It is now easier to see what has been achieved, what has been limited, and what would be the next step. Also as the method has been now well exposed – and even though I'm not totally convinced by several avenues pursued by the authors – any reader has the possibility to understand the scientific guidelines and to make it own review. For those reasons, I contend that this manuscript could be accepted when answering only one question about the floodplain methodology and looking at some writing revisions.

As we have mentioned before, we sincerely thank the reviewer, and feel honestly in debt taking into account the quality of the comments and the time put into this review. We are truly impressed by the careful and detailed review.

**1 Comments on the floodplain methodology**

Concerning the 1D floodplain methodology, I really don't understand why you are going to such complex method, looking for the loaded sediment while you do not have any data to apply it. Why don't you look directly at the floodplain according to the flood peak discharge without sediment? Moreover, it introduced a big inconsistency within the overall methodology: on the one hand you assessed the flood peak according to mud and flood prints on one river section; on the other hand you use this value to validate your hydrological model while this value must actually content both water + sediment discharge. I agree this is mainly done in hydrological modelling assuming a negligible amount of sediment. But here, as you introduced the sediment discharge in a third step, and above all as you found a non negligible 30 % of loaded sediments, it means that the hydrological model validation on (water) flood peak is not correct anymore. Could you justify that?

We understand the point raised by the reviewer and we partially agree with it. It is usually hard for a hydrological model to estimate the sediment production associated with a flash flood and it would be certainly easier to estimate the floodplain just using the flood peak discharge at each point in the channels. However, observational evidence in mountainous terrain strongly suggests that streamflow increments due to sediments are significant, and could be around 30% of the total volume [e.g. 2, 1, 3]. We decided to evaluate the model considering only the water discharge since the uncertainty in the erosion processes and their representation in the model is still significant, and the fact that we are only considering 1D processes in the channel: The hydrological model relies on water balance. In spite of that, and in order to be in the conservative side for risk management

applications, we consider a potential increase in the total discharge associated with the sediment load. We note this issue in the manuscript (discussion section).

**2 Technical comments**

- Being picky, I just noticed that the slope is called $M_{i,0}$ when being described in the THETIS model, while $\beta_{i,0}$ is used in the landslide submodel. That could be attuned.
  We have unified the terminology using $M_{i,0}$.

- A general comment: the authors often use the word "approximately" even for quantities that shouldn't be approximative (e.g the distance between the radar and the basin is approximatively 90 km, line 222, page 10). I suggest to simply remove that word in those case that doesn't make sense.
  We checked the manuscript and corrected the text in a couple of instances (distance to the radar and DEM resolution)..

**2.1 Abstract**

- **line 17:** changed event that for event. The latter..

- **line 19:** changed modulating for impacting

**2.2 Introduction**

- **line 34:** changed trying to identify for identifying

- **line 34:** changed tried to understand the governing processes of flash floods from the geological formation of the basin for related the flash floods governing processes to the geological properties of the basins

- **line 39:** Removed Due to their rapid nature, flash floods are more likely to occur in small and steep basins (Younis et al., 2008)

- **line 45:** changed vital for significant

- **line 45:** Removed Using a modeling approach

- **line 46:** changed important flash flood dependence on for significant correlation between flash flood magnitude and the...

- **line 50:** changed increases their intrinsic physical and measurement uncertainty of the latter (Wagener et al., 2007), making difficult their for makes difficult their measurements and, consequently, their understanding and their...

- **line 54:** changed )and underlining the need for for The local rainfall storm events related to flash floods require that...

- **line 55:** changed precipitation data (Norbiato et al., 2008). Given the critical role of precipitation, some for to be characterized (Noribato, 2008). Some ...

- **line 60:** changed approximately for about

- **line 63:** changed to examine the structure and evolution of each extreme rain event. The use of radar data to study flash flood-generating storms is vital for understanding and forecasting these events for They ...

- **line 93:** changed approximately for about

- **line 101:** changed with non-existing for without any

- **line 103:** removed certainly not available in real time

- **line 116:** changed issue for one

- **line 119:** changed The methodology followed in this study is based on a modeling framework using for We use the WMF (Watershed Modeling Framework) which includes a variation of the TETIS model.

- **line 122:** changed termed for called

- **line 127:** changed t Aristizábal et al. (2016). HydroFlash corresponds to for (Aristizabal et al, 2016). HydroFlash is...

- **line 128:** removed assumes infinite sediment supply and

- **line 129:** changed based on the  for on the basis of the

- **line 138:** Removed used for the reconstruction of the 2015 La Liboriana flash flood event

**2.3  Study site and data**

- **line 147:** changed , the population of Salgar was estimated at for Salgar counted

- **line 147:** changed persons for inhabitants including

- **line 150:** changed a resolution of approximately for an approximate resolution of

- **line 151:** changed allows estimating the fundamental for allow to estimate the main...

- **line 169:** changed While the elevation differences described in Figure 2 are typical of the region, the for The

- **line 182:** changed , corresponding for corresponds

- **line 204, 207, and 214:** removed approximately

- **line 221:** changed central for used

- **line 226:** removed within the radar domain to obtain spatiotemporal precipitation maps over the basin

- **line 230:** changed located approximately for The radar is

- **line 230:** changed The radar for It

- **line 249:** changed approximately for a moderate avarage of

- **line 249:** changed ; however, over the upper watershed for however

- **line 250:** changed according to the estimated rainfall amounts based on the radar measurements for over the upper watershed

**2.4  Methodology**

- **line 350:** changed allowing the study of the role of flows of different nature during extreme events for enabling to study the different flow paths and water origins

- **line 351:** removed , thereby providing insight about the soil-dependent flow regulation

- **line 363:** removed approximately

[revised manuscript text omitted]